# TSP: A Two-Sided Smoothed Primal-Dual Method for Nonconvex Bilevel Optimization

**Songtao Lu** [1]

## Abstract

Extensive research has shown that a wide range of machine learning problems can be formulated as bilevel optimization, where two levels of learning processes intertwine through distinct sets of optimization variables. However, prevailing approaches often impose stringent assumptions, such as strong convexity of the lower-level loss function or uniqueness of the optimal solution, to enable algorithmic development and convergence analysis. However, these assumptions tend to be overly restrictive in real-world scenarios. In this work, we explore a recently popularized Moreau envelope based reformulation of bilevel optimization problems, accommodating nonconvex objective functions at both levels. We propose a stochastic primal-dual method that incorporates smoothing on both sides, capable of finding Karush-Kuhn-Tucker solutions for this general class of nonconvex bilevel optimization problems. A key feature of our algorithm is its ability to dynamically weigh the lower-level problems, enhancing its performance, particularly in stochastic learning scenarios. Numerical experiments underscore the superiority of our proposed algorithm over existing penalty-based methods in terms of both the convergence rate and the test accuracy.

## 1. Introduction

Bilevel optimization problems have been established as a general formulation for a wide range of machine learning tasks. The two-level structure enables the integration of different learning or optimization processes. This approach ensures that the solution obtained strikes a balance between the two learning objectives. Typical applications include hyperparameter optimization (Franceschi et al., 2017; Shaban et al., 2019), meta-learning (Rajeswaran et al., 2019; Raghu et al., 2020), coreset selection (Zhou et al., 2022; Hao et al., 2024), actor-critic schemes in reinforcement learning (Hong et al., 2023), and many more (Liu et al., 2021a; Lu, 2023). Specifically, it takes the form of:

$$\min_{x, y \in \mathcal{S}(x)} f(x, y), \quad \text{s.t.} \quad \mathcal{S}(x) \triangleq \arg\min_y g(x, y), \quad (1)$$

where $f(x, y)$ and $g(x, y)$ denote the upper-level (UL) and lower-level (LL) objective functions, $\mathcal{S}(x)$ represents the feasible sets that are the optimal solution set that contains all the global optimal solutions of the LL problem with respect to (w.r.t.). the block-$y$. However, solving this class of problems is challenging. Even when both functions $f(x, y)$ and $g(x, y)$ are differentiable and smooth, computing the gradient of the UL loss function with respect to $x$ may involve the computation of high-order derivatives of the LL loss function. For example, when the LL loss function is strongly convex, the closed-form expression of this gradient requires the computation of both the Jacobian and the inverse Hessian matrix w.r.t. the LL loss function (Ghadimi & Wang, 2018). Many works have focused on directly optimizing the UL and LL loss functions by applying iterative numerical methods to perform the Hessian inverse operation, such as reverse-mode iterative differentiation, approximate implicit differentiation techniques (Grazzi et al., 2020; Ji et al., 2021), and the Lanczos method (Gao et al., 2025), achieving good performance. However, a major assumption they cannot avoid is the uniqueness of the LL optimal solution, i.e., the optimal solution set $\mathcal{S}(x)$ must be a singleton. This restrictively constrains the applicability of these algorithms for many machine learning problems where the LL loss function contains multiple optimal solutions. For example, a convex LL objective function is one of the simplest cases where this issue arises.

Targeting this challenge, one of the most straightforward approaches is to penalize the LL optimization problem in the UL, forming a single-level optimization problem that can be tackled with existing constrained optimization methods. Specifically, the original bilevel optimization problem can be written as

$$\min_{x, y} f(x, y), \quad \text{s.t.} \quad g(x, y) - g^\star(x) \leq \delta \quad (2)$$

[1]Department of Computer Science and Engineering and Shun Hing Institute of Advanced Engineering, The Chinese University of Hong Kong, Hong Kong. Correspondence to: Songtao Lu <stlu@cse.cuhk.edu.hk>.

*Proceedings of the 42nd International Conference on Machine Learning*, Vancouver, Canada. PMLR 267, 2025. Copyright 2025 by the author(s).

where $g^\star(x) \triangleq \min_y g(x, y)$ (which is also called value function (Liu et al., 2021b)) is obtained by minimizing $g(x, y)$ over $y$, and $\delta \geq 0$. It can be easily checked that when $\delta = 0$, problem (2) reduces to the original one (1). In this way, optimizing the LL loss function is transformed into enforcing constraint satisfaction, assuming that $g^\star(x)$ can be obtained by some oracles. When $g(x, y)$ w.r.t. $y$ is convex or satisfies certain conditions, such as the Polyak-Łojasiewicz (PŁ) condition, applying gradient descent is sufficient to reach the optimal solution. When $\delta > 0$, there is always a strictly feasible solution for this constraint since $g^\star(x)$ is the minimum value of the LL function. However, finding the optimal solution $g^\star(x)$ generally requires an inner loop algorithm, which weakens the numerical performance of the developed algorithm in practice and complicates the theoretical analysis of the stochastic bilevel algorithm. This is because additional criteria are needed to determine when to stop the inner loop optimization process. This motivates us to explore the following question:

*Can we design a first-order algorithm capable of solving stochastic bilevel optimization problems where both the UL and LL objectives are nonconvex (or weakly convex)?*

## 1.1. Related Work

**Bilevel Optimization without LL Strong Convexity.** There has been a line of work focusing on solving bilevel optimization problems without assuming the strong convexity of the LL objective functions (Liu et al., 2020). For example, the convexity assumption can be replaced by convexity or a certain type of nonconvex property, such as the PŁ condition (Huang, 2024). Aiming at the nonsmoothness issue raised by the multiple LL optimal solutions, a variant of stationary points, i.e., Goldstein stationary points, is used as the metric to quantify the solutions that can be achieved by the zeroth-order switching gradient method with a convergence rate depending on the problem dimension (Chen et al., 2023; Liu et al., 2024b;c). This kind of switching idea has also been adopted in the conditional gradient-based method for solving simple bilevel problems, where one step is searching for the feasible set based on the convex LL loss function while the other one is for minimizing the UL loss function given the obtained solution set. It is shown that this method can achieve $\mathcal{O}(1/\epsilon^2)$ convergence rate to find the $\epsilon$-stationary points under the Frank-Wolfe gap (Jiang et al., 2023; Cao et al., 2024).

Given the problem formulation (2), the penalty method is one of the most standard ways to solve the constrained optimization problem. A min-max reformulation of (2) is considered in (Lu & Mei, 2024), where the minimization is performed with respect to the UL problem while maximization is used for optimizing the LL problem by introducing an auxiliary variable. Besides, if the UL loss function is strongly convex w.r.t. $y$, a single-loop bilevel averaged method of

multipliers (sl-BAMM) (Liu et al., 2023) is still possible to perform Hessian inverse operation, provided with strong convexity property, through dynamically averaging the UL loss function with the convex LL loss function. Further, under the assumption that the LL loss function satisfies the PŁ condition, it has been shown that solving the penalized version of (2) is equivalent to solving the original one in terms of the local and global optimality conditions when the penalty parameter is sufficiently large enough (Shen & Chen, 2023). The authors further develop a double-loop structured value gap-based penalty-based bilevel gradient descent (V-PBGD) algorithm that can find the stationary points of the reformulated penalty-based problem at a rate of $\mathcal{O}(\nu/\epsilon^2)$, where $\nu$ is the penalty parameter. The choice of $\nu$ can be a constant or a dynamically increasing sequence. Another possible way of choosing this parameter is constructing a barrier function that can ensure the decrease of the LL loss function, which is called optimization made easy (BOME) (Liu et al., 2022) algorithm, but the convergence rate of BOME is rather slow. Recent works can sharpen the convergence rate of finding the stationary points of bilevel problems up to $\mathcal{O}(1/\epsilon^2)$ when the LL loss satisfies the PŁ condition, but the convexity assumption and the computation of the Jacobian matrix are further required (Xiao et al., 2024).

**Moreau Envelope Based Methods for Nonconvex Optimization.** Even though penalty methods have achieved great success in solving bilevel optimization problems, their numerical performance often falls short compared to the Lagrangian method. The Lagrange multiplier or dual variable inherent in the Lagrangian method allows for automatic adjustment of constraint violations, leading to faster empirical convergence rates, particularly in scenarios involving multiple constraints (Boob et al., 2023; Jin & Wang, 2022; Li et al., 2024c). However, in cases where the objective function, even in single-level constrained problems, is nonconvex, traditional primal-dual algorithms may fail to converge due to the zero-sum nature of the game involving the increase and decrease of the Lagrangian function. Smoothness serves as an effective strategy to stabilize the convergence of such algorithms. For instance, the smooth Lagrangian method proposed in (Zhang & Luo, 2020; Zeng et al., 2022) has demonstrated efficacy in finding Karush-Kuhn-Tucker (KKT) solutions for nonconvex optimization problems under linear constraints, with further extensions to convex or functional constrained scenarios (Zhang & Luo, 2020; Lu, 2024). Various types of smoothed gradient descent-ascent algorithms have achieved state-of-the-art convergence rates in both concave–concave (Zhao, 2024) and nonconvex minmax optimization problems (Zhang et al., 2020; Zheng et al., 2023; Jiang et al., 2025; Huang & Lin, 2023). This notable performance can be attributed to the equivalence between the smoothed gradient method and the Moreau envelope formulation of nonconvex optimization problems (Nesterov,

*Table 1.* Comparison of representative existing works on nonconvex bilevel optimization and stochastic functionally constrained optimization, where "gradient" indicates the requirements for accessing the first- or second-order derivative of either UL or LL loss functions, "singleton" refers to the uniqueness of the LL optimal solution, "LL" denotes the property that the LL objective function needs to satisfy, and "KKT*" represents the case that constraint satisfaction is achieved while the slackness condition is not verified. Additionally, "cvx" stands for convex, "ncvx" stands for nonconvex or weakly convex, and "scvx" denotes strongly convex.

| Algorithms | Solution | Method | Gradient | Singleton | LL | # of Loops | Rate |
|---|---|---|---|---|---|---|---|
| Inexact-ConEx (Boob et al., 2023) | KKT | primal-dual | stochastic (1st) | n/a | n/a | double | $\mathcal{O}\left(\epsilon^{-6}\right)$ |
| Stoc-iALM (Li et al., 2024c) | KKT | primal-dual | stochastic (1st) | n/a | n/a | double | $\mathcal{O}\left(\epsilon^{-5}\right)$ |
| MA-SOBA (Chen et al., 2024) | stationary | SGD | stochastic (2nd) | ✓ | scvx | single | $\mathcal{O}\left(\epsilon^{-4}\right)$ |
| F$^2$SA (Kwon et al., 2023) | stationary | penalty | stochastic (2nd) | ✓ | scvx | double | $\mathcal{O}\left(\epsilon^{-5}\right)$ |
| BOME (Liu et al., 2022) | KKT* | penalty | deterministic (1st) | ✓ | PŁ | double | $\mathcal{O}\left(\epsilon^{-6}\right)$ $\mathcal{O}\left(\epsilon^{-8}\right)$ |
| SLM (Lu, 2024) | KKT | primal-dual | deterministic (1st) | | PŁ | double | $\mathcal{O}\left(\epsilon^{-7}\right)$ |
| sl-BAMM (Liu et al., 2023) | KKT | penalty | deterministic (2nd) | | cvx | single | $\mathcal{O}\left(\epsilon^{-5}\right)$ |
| penalty method (Lu & Mei, 2024) | KKT | penalty | deterministic (1st) | | cvx | multiple | $\mathcal{O}\left(\epsilon^{-4}\right)$ |
| V-PBGD (Shen & Chen, 2023) | stationary | penalty | deterministic (1st) | | PŁ | double | $\mathcal{O}\left(\nu\epsilon^{-2}\right)$ |
| MEHA (Liu et al., 2024a) | stationary | penalty | deterministic (1st) | | ncvx | single | $\mathcal{O}\left(\nu\epsilon^{-2}\right)$ |
| **TSP** (this work) | KKT | primal-dual | stochastic (1st) | | ncvx | single | $\mathcal{O}\left(\epsilon^{-4}\right)$ |

2005). Recent research has reformulated original bilevel optimization problems using the Moreau envelope, demonstrating that the developed algorithms can identify well-defined KKT points, particularly when the LL loss function is convex (Gao et al., 2023). Moreover, this formulation has been adapted to accommodate additional functional constraints in the LL optimization problem (Yao et al., 2024; 2025). In (Liu et al., 2024a), it was shown that a single-loop Moreau envelope-based Hessian-free algorithm (MEHA) can find stationary points of the reformulated bilevel problem, even in scenarios where both UL and LL loss functions are nonconvex.

***Stochastic Algorithms for Constrained and Bilevel Optimization.*** More attractively, this kind of single-loop structure is more accessible for developing stochastic algorithms. For bilevel optimization with a strongly convex LL objective function, numerous existing stochastic algorithms have been proposed (Kwon et al., 2023; 2024b; Hong et al., 2023; Chen et al., 2022; Shen & Chen, 2022; Yang et al., 2023; Kwon et al., 2024a; Chen et al., 2024). However, due to the constrained nature of bilevel optimization, especially in cases without strong convexity at the LL, demonstrating the convergence of stochastic algorithms is highly challenging. This challenge arises because the dual variable, when utilizing stochastic gradients or functions, can become unbounded, leading to the failure of enforcing the constraint. Existing works on constrained optimization assume the boundedness of the feasible set to enforce the boundedness of the gradient size (Li et al., 2024c; Jin & Wang, 2022; 2024), which is theoretically overly restrictive, or they adopt variance reduction techniques or large batch sizes to mitigate random noise (Alacaoglu & Wright, 2024; Shen & Chen, 2023). These factors collectively con-

tribute to the limited exploration of convergence guarantees for stochastic primal-dual or penalty algorithms in bilevel optimization.

### 1.2. Main Contributions of This Work

In this work, we propose a two-sided smoothed primal-dual method, abbreviated as TSP, for solving nonconvex (stochastic) bilevel optimization problems. Benefiting from the Moreau envelope-based reformulation of the bilevel optimization problem, the TSP algorithm is structured as a single loop, making it easily implementable in a stochastic fashion. By quantifying the descent of our constructed potential function, we demonstrate that the proposed TSP algorithm can find the $\epsilon$-KKT points of the reformulated bilevel problem with a convergence rate of $\mathcal{O}(1/\epsilon^4)$ with high probability. To the best of our knowledge, this is the first result established for quantifying the convergence rate of first-order stochastic methods in finding the $\epsilon$-KKT points for this class of bilevel optimization problems. Our numerical results validate the superior performance of this formulation as well as the quality of the obtained solutions in terms of generalization errors.

The main contributions of this work are highlighted as follows:

▶ The developed TSP algorithm is gradient-based, single-looped, and stochastic, making it easily implementable for solving bilevel machine learning problems in a computationally efficient way.

▶ The theoretical iteration complexity of TSP is $\mathcal{O}(\epsilon^{-4})$ with high probability for finding the $\epsilon$-KKT solutions of the Moreau envelope-reformulated bilevel optimization problem. To the best of our knowledge, this is the first time a stochastic first-order method has successfully

achieved the approximate KKT points of the bilevel optimization problem where the LL objective function is weakly convex.

▶ Numerical results further emphasize the importance of finding the KKT points of this class of problems in comparison with the stationary points in the penalty-based reformulation of bilevel optimization problems in terms of generalization performance.

Due to space constraints, all technical proofs are provided in the supplementary material.

## 2. Primal-Dual Method for Moreau Envelope-Based Bilevel Optimization

In this section, we will introduce a single-loop gradient-based primal-dual method designed to solve the following Moreau envelope based reformulation (Gao et al., 2023; Yao et al., 2024; Liu et al., 2024a) of the following general stochastic bilevel optimization problem.

$$\min_{x,y} \quad f(x,y) \triangleq \mathop{\mathbb{E}}_{\xi \sim \mathcal{D}_{\mathrm{UL}}} F(x,y;\xi) \qquad (3a)$$

$$\text{s.t.} \quad g(x,y) - g_\gamma^\star(x,y) \leq \delta \qquad (3b)$$

where

$$g_\gamma^\star(x,y) \triangleq \arg\min_z \mathop{\mathbb{E}}_{\xi \sim \mathcal{D}_{\mathrm{LL}}} G(x,z;\xi) + \frac{1}{2\gamma}\|z-y\|^2 \qquad (4)$$

denotes the value function of this problem and serves as a lower bound of the original LL loss function, $F(x,y;\xi)$ and $G(x,y;\zeta)$ respectively denote the stochastic UL and LL loss functions, $\mathcal{D}_{\mathrm{LL}}, \mathcal{D}_{\mathrm{UL}}$ respectively denote the UL and LL data distributions at each level, and $\gamma > 0$. Let $g(x,y) \triangleq \mathbb{E}_{\xi \sim \mathcal{D}_{\mathrm{LL}}} G(x,y;\xi)$. When $\delta = 0$, it has been proven in Theorem 1 (Gao et al., 2023) that the problem formulations (1) and (3) are equivalent when $g(x,y)$ is convex and in (Liu et al., 2024a) that the solutions of (3) that satisfy the constraint (3b) are also stationary points of the LL problem (i.e., satisfying $\|\nabla_y g(x,y)\| = 0$) in the original formulation (1), when $g(x,y)$ is weakly convex with respect to $y$.

It is also worth noting that when $\gamma$ is small, the LL loss function becomes strongly convex in $z$, ensuring a uniquely well-defined LL optimal solution. This motivates the development of an algorithm based on this smoothed problem (Bai et al., 2024), particularly in practical stochastic settings. Towards this end, we can construct the Lagrangian function of this bilevel problem as follows:

$$\mathcal{L}(x,y;\lambda) \triangleq f(x,y) + \lambda(g(x,y) - g_\gamma^\star(x,y) - \delta) \qquad (5)$$

where the nonnegative $\lambda$ denotes the Lagrange multiplier or dual variable for the inequality constraint (3b).

After using the Moreau envelope smoothing technique on the LL objective function, we apply the proximal smoothing

terms to the UL objective function, similar to existing works dealing with nonconvex optimization (Zhang & Luo, 2020; Zheng et al., 2023; Lu, 2024), resulting in the following smoothed Lagrangian.

$$K(x,y,\widehat{x},\widehat{y};\lambda) \triangleq f(x,y) + \lambda(g(x,y) - g_\gamma^\star(x,y) - \delta)$$
$$+ \frac{p}{2}\|x-\widehat{x}\|^2 + \frac{p}{2}\|y-\widehat{y}\|^2 \qquad (6)$$

where $\widehat{x},\widehat{y}$ have the same size as $x,y$. It can be easily checked that given $\widehat{x},\widehat{y}$, the smoothed Lagrangian is strongly convex w.r.t. $x$ and $y$ when $p$ is sufficiently large. Next, the algorithm design for finding the equilibrium points of $\min_{x,y,\widehat{x},\widehat{y}} \max_{\lambda \geq 0} K(x,y,\widehat{x},\widehat{y};\lambda)$ is fairly straightforward. We can apply the linearized Lagrangian method or primal-dual method to update the optimization variables using only (stochastic) gradients.

***Dual Update.*** Based on the Moreau envelope-based LL optimization problem (6), we further propose updating the dual variable using a moving average technique, as follows.

$$h^{r+1} = (1-\theta)h^r$$
$$+ \theta\big(\widehat{g}(x^r,y^r) - \widehat{g}(x^r,z^r) - \frac{1}{2\gamma}\|z^r - y^r\|^2 - \delta\big), (7a)$$

$$\lambda_+^r = \mathrm{Proj}_{\geq 0}\left(\lambda^r + \tau h^{r+1}\right), \qquad (7b)$$

$$\lambda^{r+1} = (1-\mu)\lambda^r + \mu\lambda_+^r \qquad (7c)$$

where $r$ stands for the index of the iterations, $\tau$ denotes the step-size for updating the dual variable $\lambda_+^r$, $\mathrm{Proj}_{\geq} 0$ is the nonnegative projection operator that ensures the iterates remain in the nonnegative orthant, and $\mu$ and $\theta$ are smoothing or dampening parameters for the dual variable $\lambda$ and the auxiliary variable $h$, respectively. Here, $\hat{g}(x,y)$ denotes a mini-batch stochastic approximation of $g(x,y)$, computed using a fixed number of i.i.d. samples from the LL data distribution $\mathcal{D}_{\mathrm{LL}}$. The moving average applied to model parameter updates reduces their aggressiveness compared to traditional primal-dual methods, improving the algorithm's robustness to stochastic errors.

***Primal Update.*** After that, we can use stochastic gradient descent (SGD) to get an estimate of $z^\star(x,y)$ given $x$ and $y$.

$$z^{r+1} = z^r - \eta\left(h_z^g(x^r,z^r) + \frac{1}{\gamma}(z^r - y^r)\right) \qquad (8)$$

where $\eta$ is the step-size, and $h_z^g(x,z)$ represents the stochastic gradient estimate of $\nabla_z g(x,z)$ with a constant mini-batch size of independent samples.

It has been established that function $g_\gamma^\star(x,y)$ is differentiable when $\gamma \in (0, 1/(2\rho))$. The rest of the algorithm design involves simply applying SGD with respect to the remaining variables using the function $K(x,y,\widehat{x},\widehat{y};\lambda)$ and replacing the unknown $z^\star$ with its surrogate $z^{r+1}$. Specifically, the updates for $y$ is

$$y^{r+1} = y^r - \beta\big(h_y^f(x^r, y^r)$$
$$+\lambda^{r+1}\left(h_y^g(x^r, y^r) + \frac{z^r - y^r}{\gamma}\right) + p(y^r - \widehat{y}^r)\big), \quad (9)$$

followed by

$$\widehat{y}^{r+1} = \widehat{y}^r + \omega(y^{r+1} - \widehat{y}^r), \quad (10)$$

where $\beta$ denotes the step-size for updating variable $y$, and $0 < \omega < 1$ is the smoothing factor for updating $\widehat{y}$, and $h_y^f(x, y)$ and $h_y^g(x, y)$ represent the stochastic gradient estimates of $\nabla_y f(x, y)$ and $\nabla_y g(x, y)$, respectively.

Similarly for $x$, it is updated by

$$x^{r+1} = x^r - \alpha\Big(h_x^f(x^r, y^{r+1}) + \lambda^{r+1}\big(h_x^g(x^r, y^{r+1})$$
$$- h_x^g(x^r, z^r)\big) + p(x^r - \widehat{x}^r)\Big), \quad (11)$$

followed by

$$\widehat{x}^{r+1} = \widehat{x}^r + \omega(x^{r+1} - \widehat{x}^r), \quad (12)$$

where $\alpha$ denotes the step-size for updating variable $x$, and $0 < \omega < 1$ is the smoothing factor for updating $\widehat{x}$, and $h_x^f(x, y)$ and $h_x^g(x, y)$ represent the stochastic gradient estimates of $\nabla_x f(x, y)$ and $\nabla_x g(x, y)$, respectively. A summary of the implementation of the TSP algorithm is provided in Algorithm 1.

---

**Algorithm 1** Single-loop stochastic **T**wo-sided **S**moothed **P**rimal-dual (TSP) method for bilevel optimization

---
**Initialization:** step-sizes: $\tau, \mu, \theta, \eta, \alpha, \beta, \omega$, variables: $x^1, y^1, z^1, \widehat{x}^1, \widehat{y}^1, \lambda^1, h^1 = 0$
1: **for** $r = 1, 2, \cdots, T$ **do**
2:    Compute $h^{r+1}$ by (7a)
3:    Update $\lambda_+^{r+1}, \lambda^{r+1}$ by (7b) and (7c)
4:    Update $z^{r+1}$ by (8)
5:    Update $y^{r+1}$ and $\widehat{y}^{r+1}$ by (9) and (10)
6:    Update $x^{r+1}$ and $\widehat{x}^{r+1}$ by (11) and (12)
7: **end for**

---

## 3. Theoretical Convergence Results

We first need to make the following blanket assumption to showcase the convergence behavior of the proposed TSP algorithm.

### 3.1. Assumptions
These assumptions are mainly related to the continuity and boundedness of the UL and LL objective functions.

A1. (Smoothness) Assume that functions $f(x, y), g(x, y)$ are differentiable and jointly smooth with constants $L_f, L_g$ w.r.t. both $x, y$.

A2. (Boundedness) Assume that the objective function $f(x, y)$ is lower bounded and denoted as $\underline{f}$.

A3. (Coercivity) The set $\{x, y | f(x, y) \leq R, g(x, y) - g_\gamma^\star(x, y) \leq \delta\}$ is bounded for any $R > 0$.

Assumption A1 implies that $g(x, y)$ is weakly convex in $y$ for any fixed $x$, with weak convexity parameter $\rho$ used throughout the paper.

*Remark 1.* Assumption A3, which requires the objective function to be closed over its open domain, is widely used in optimization theory to ensure bounded level sets and the existence of minimizers (Boyd & Vandenberghe, 2004). In practice, incorporating a small $\ell_2$-penalty into the loss function is a common technique to enforce bounded level sets.

Further, we make the following standard assumptions on the stochastic properties of the gradient estimate. Let us define the gradient estimation noise involved in the primal variable update: $\varepsilon_{g_x}(x, y) \triangleq h_x^g(x, y) - \nabla_x g(x, y), \; \varepsilon_{g_y}(x, y) \triangleq h_y^g(x, y) - \nabla_y g(x, y), \; \varepsilon_{g_z}(x, y) \triangleq h_z^g(x, z) - \nabla_y g(x, z), \; \varepsilon_{f_x} \triangleq h_x^f(x, y) - \nabla_x f(x, y), \; \varepsilon_{f_y} \triangleq h_y^f(x, y) - \nabla_y f(x, y)$, Regarding the dual update, we define the stochastic gradient estimation noise as $\varepsilon_{\widehat{g}_y} \triangleq \widehat{g}(x, y) - g(x, y), \varepsilon_{\widehat{g}_z} \triangleq \widehat{g}(x, z) - g(x, z)$. To ensure theoretical tractability, we make the following assumptions on these quantities.

A4. (Stochasticity of Gradient Estimate in Primal Variable Update) Gradient noise $\mathbb{E}[\varepsilon_{g_\cdot}] = 0$ and $\mathbb{E}[\|\varepsilon_{g_\cdot}\|^2] = \sigma_{g_\cdot}^2$ and $\mathbb{E}[\varepsilon_{f_\cdot}] = 0$ and $\mathbb{E}[\|\varepsilon_{f_\cdot}\|^2] = \sigma_{f_\cdot}^2$, where $\cdot$ represents any of $x, y, z$ with respect to $g$, and $x, y$ with respect to $f$, while $\mathbb{E}$ denotes the expectation conditioned on all past gradient estimates up to the most recent iteration.

A5. (Stochasticity of Function Estimate in Dual Variable Update) Function noise $\mathbb{E}[\varepsilon_{\widehat{g}_\cdot}] = 0$ and $\mathbb{E}[\|\varepsilon_{\widehat{g}_\cdot}\|^2] = \sigma_{\widehat{g}_\cdot}^2$, where $\cdot$ represents anyone of $y, z$.

### 3.2. Iteration Complexity of TSP to the KKT points
Given the above preliminary assumptions, we define $\mathcal{G}(x, y; \lambda)$ as

$$\mathcal{G}(x, y; \lambda) \triangleq \begin{bmatrix} \nabla_x \mathcal{L}(x, y; \lambda) \\ \nabla_y \mathcal{L}(x, y; \lambda) \end{bmatrix}$$

and use $\|\mathcal{G}(x, y; \lambda)\|$ as the stationary gap. Then, an $(\epsilon, \delta)$-approximate KKT point of the constrained problem (3) is naturally defined as follows.

*Definition of $(\epsilon, \delta)$-Approximate KKT Points.* A point $(x^\star, y^\star, \lambda^\star)$ is called an $(\epsilon, \delta)$-approximate KKT point if it satisfies the following three conditions: 1) stationarity condition: $\|\mathcal{G}(x^\star, y^\star; \lambda^\star)\| \leq \epsilon$; 2) constraint violation condition: $|g(x^\star, y^\star) - g_\gamma^\star(x^\star, y^\star) - \delta|_+ \leq \epsilon$; 3) slackness or complementarity condition: $\|g(x^\star, y^\star) - g_\gamma^\star(x^\star, y^\star) - \delta\|\|\lambda^\star\| \leq \epsilon$.

Now, we are ready to show the following theoretical convergence rate of TSP.

**Theorem 1.** (*Convergence Rate of TSP to the $(\epsilon, \epsilon)$-Approximate KKT Points of Problem (3)*). *Suppose that A1–A5 hold. Assume that the iterates $\{x^r, y^r, z^r, \widehat{x}^r, \widehat{y}^r, \lambda^r, \lambda^r_+\}$ are generated by TSP. If the step-sizes $\alpha, \beta, \tau, \omega, \eta, \mu, \theta$ are chosen as $\mathcal{O}(1/\sqrt{T})$ and the parameter $p = \mathcal{O}(\lambda^r)$, then for $T \geq \Theta(\epsilon^{-4})$, the following results hold with high probability:*

$$\frac{1}{T} \sum_{r<T} \|\mathcal{G}(x^r, y^r; \lambda^{r+1})\|^2 \leq \epsilon^2, \tag{13a}$$

$$\frac{1}{T} \sum_{r<T} |g(x^r, y^r) - g^\star_\gamma(x^r, y^r) - \epsilon|^2_+ \leq \epsilon^2, \tag{13b}$$

$$\frac{1}{T} \sum_{r<T} \|g(x^r, y^r) - g^\star_\gamma(x^r, y^r) - \epsilon\|^2 \|\lambda^r\|^2 \leq \epsilon^2, \tag{13c}$$

*where $\epsilon > 0$, $|\cdot|_+$ denotes the positive part, and $T$ is the total number of iterations.*

*Remark 2.* The convergence rate achieved by TSP is optimal, as it matches the lower bound of standard SGD for solving single-level smooth nonconvex problems (Arjevani et al., 2023).

*Remark 3.* The batch size used in TSP is a constant or of size 1; therefore, the sample complexity of TSP is also $\mathcal{O}(\epsilon^{-4})$, which is consistent with SGD.

*Remark 4.* When $g(x, y) = -f(x, y)$, the bilevel problem (1) reduces to the min-max optimization problem $\min_x \max_y f(x, y)$ under the weakly-convex weakly-concave setting. Under assumptions A1–A5, the analysis shows that the iterates generated by TSP remain within a bounded region without requiring additional projection. This implies that the loss values remain bounded over the unconstrained domain, where approximate stationary points of nonconvex-nonconcave min-max problems are shown to always exist and can be found in polynomial time (Daskalakis et al., 2021).

### 3.3. Proof Sketch

The theoretical proofs guiding the algorithm to achieve this iteration and sample complexity mainly consist of three key steps: 1) constructing a potential function $\mathcal{Q}^r$ to track the convergence progress as the algorithm proceeds, 2) bounding the size of the dual variable given the bounded gradients, and 3) deriving the probability that the iterates remain within the bounded region.

Thanks to the smoothing terms introduced in the Moreau envelope reformulation, the function $K(\cdot)$ with respect to the variables $x$ or $y$ exhibits strong convexity in each subproblem. Mathematically, these subproblems can be defined by the following quantities:

$$D(\widehat{x}, \widehat{y}; \lambda) \triangleq \min_{x,y} K(x, y, \widehat{x}, \widehat{y}; \lambda), \tag{14a}$$

$$P(\widehat{x}, \widehat{y}) \triangleq \min_{x, y \in \mathcal{Y}(x)} f(x, y) + \frac{p}{2}\|x - \widehat{x}\|^2 + \frac{p}{2}\|y - \widehat{y}\|^2, \tag{14b}$$

where $\mathcal{Y}(x) \triangleq \{y \mid g(x, y) - g^\star_\gamma(x, y) \leq \delta\}$, and

$$x^\star(\widehat{x}, \widehat{y}; \lambda), y^\star(\widehat{x}, \widehat{y}; \lambda) \triangleq \arg\min_{x,y} K(x, y, \widehat{x}, \widehat{y}; \lambda) \tag{15}$$

denote the optimal solutions of problem (14a) given the reference point $(\widehat{x}, \widehat{y}, \lambda)$. Similarly,

$$\bar{x}^\star(\widehat{x}, \widehat{y}), \ \bar{y}^\star(\widehat{x}, \widehat{y})$$
$$\triangleq \arg\min_{x, y \in \mathcal{Y}(x)} f(x, y) + \frac{p}{2}\|x - \widehat{x}\|^2 + \frac{p}{2}\|y - \widehat{y}\|^2 \tag{16}$$

denote the optimal solutions of (14b) given $(\widehat{x}, \widehat{y})$. Then, we can utilize these quantities, which serve as intermediate anchors for monitoring the optimization process, to derive the descent lemma for TSP.

***Descent Lemma.*** After one round update of variables by TSP (i.e., from $(x^r, y^r, z^r, \widehat{x}^r, \widehat{y}^r, \lambda^r, \lambda^r_+)$ to $(x^{r+1}, y^{r+1}, z^{r+1}, \widehat{x}^{r+1}, \widehat{y}^{r+1}, \lambda^{r+1}, \lambda^{r+1}_+)$), we obtain the following intriguing result.

**Lemma 1.** (*informal*) *Assume that A1-A5 are satisfied. Suppose the sequence $\{x^r, y^r, z^r, \widehat{x}^r, \widehat{y}^r, \lambda^r, \lambda^r_+, \forall r\}$ is generated by TSP, with $p > L$ and $\lambda^r \leq \Lambda$. Additionally, assume that $y^r, \bar{y}^\star(\widehat{x}^r, \widehat{y}^r)$, and $y^\star(\widehat{x}^r, \widehat{y}^r; \lambda^{r+1})$ are bounded. Then, if the step-sizes are chosen appropriately, we have either*

$$\mathcal{Q}^{r+1} - \mathcal{Q}^r$$
$$\leq -\frac{1}{8\alpha} \left\|\mathbb{E}\left[x^{r+1} - x^r\right]\right\|^2 - \frac{1}{8\beta} \left\|\mathbb{E}\left[y^{r+1} - y^r\right]\right\|^2$$
$$- \frac{p}{8\omega}\|\widehat{x}^{r+1} - \widehat{x}^r\|^2 - \frac{p}{8\omega}\|\widehat{y}^{r+1} - \widehat{y}^r\|^2$$
$$- \frac{(1-\varphi)C_z}{4}\|z^r - z^\star(x^r, y^r)\|^2$$
$$- \frac{1}{16\mu\tau}\|\lambda^r_+(\widehat{x}^r, \widehat{y}^r) - \lambda^r\|^2 + n^r_Q, \quad or \tag{17}$$

$$\Big\{\frac{1}{4\alpha}\|\mathbb{E}x^{r+1} - x^r\|^2, \frac{1}{4\beta}\|\mathbb{E}y^{r+1} - y^r\|^2, \frac{p}{4\omega}\|\widehat{x}^{r+1} - \widehat{x}^r\|^2,$$
$$\frac{p}{4\omega}\|\widehat{y}^{r+1} - \widehat{y}^r\|^2, \frac{(1-\varphi)C_z}{2}\|z^r - z^\star(x^r, y^r)\|^2\Big\}$$
$$= \mathcal{O}(\Lambda^2\mu\tau) \text{ and } \|\lambda^r - \lambda^r_+(\widehat{x}^r, \widehat{y}^r)\| = \mathcal{O}(\mu\tau\Lambda) \tag{18}$$

*where $n^r_Q$ is the noise term resulting from the gradient estimate, $0 < \varphi < 1$, the coefficient $C_z = \mathcal{O}(\alpha)$, and $L, \Lambda$ are some positive constants.*

From this lemma, it follows that the potential function is either monotonically decreasing up to some noise ball or the generated iterates have already converged to neighborhoods of the stationary points with a radius of $\mathcal{O}(\Lambda\tau\mu)$. In the first

case, we can easily show that the algorithm will eventually converge to $\epsilon$-stationary points, provided that $Q^r$ is lower bounded by some constant $\underline{Q}$. In the latter case, we need to derive an upper bound for the dual variable by selecting sufficiently small step-sizes for TSP, allowing us to conclude that the iterates have reached the $\epsilon$-KKT points.

***Bounding the Dual Variable***. Given Lemma 1, we can quantify the difference between successive iterates, which plays a crucial role in upper bounding the dual variable.

**Lemma 2.** *Under A1–A5, suppose the sequence $\{x^r, y^r, z^r, \widehat{x}^r, \widehat{y}^r, \lambda^r, \lambda_+^r, \forall r\}$ is generated by TSP. Assume that $y^r$, $h_y^f$, and $h_y^g$ are bounded. When $\delta > 0$, $p = \Theta(1)$, $\delta = \mathcal{O}(\epsilon)$, and the step-sizes are chosen on the order of $T^{-1/2}$, then $\lambda^r$ is upper bounded, i.e., there exists a positive constant $\Lambda$ such that $\lambda^r \leq \Lambda, \forall r$.*

This result indicates that the dual variable $\lambda^r$ remains bounded. However, since the gradient estimate can be unbounded due to stochasticity, we further assess the probability that the gradient estimate remains bounded.

***Bounding the Random Noise***. By leveraging the probabilistic bounds on gradient magnitudes generated by Adam or SGD (Li et al., 2024a;b), we establish a heavy-tailed noise bound for TSP with bounded variance in the proof of Theorem 1. Specifically, we define the following random variables: $t_1 \triangleq \min\{r \mid Q^r - \underline{Q} > Q_{\text{th}}\} \wedge T$, $t_2 \triangleq \min\{r \mid \|\varepsilon_{\max}^r\| > G_{\text{th}}\} \wedge T$, where $a \wedge b$ denotes $\min\{a, b\}$, and $\varepsilon_{\max}^r$ represents the largest magnitude of gradient estimate errors among $\varepsilon_{g.}, \varepsilon_{f.}, \varepsilon_{\widehat{g}.}$. The thresholds $Q_{\text{th}}$ and $G_{\text{th}}$ are predefined thresholds.

The variable $t_2$ quantifies the time at which the iterates become unbounded, while $t_1$ links the term $Q^r$ to measure the gradient magnitude throughout the iterations. Let $t \triangleq \min\{t_1, t_2\}$. Based on the derived descent lemma and concentration inequalities, we show that the probability of $t < T$ is small, implying that the probability of $t = T$ is high. This directly ensures that the gradients remain bounded before $T$.

## 4. Numerical Results

In this section, we evaluate the numerical performance of the proposed TSP algorithm by comparing it with state-of-the-art bilevel optimization algorithms, particularly those based on penalty methods. These methods are closely related to the idea of penalizing the LL loss function using the UL loss function.

**Data Hyper-Cleaning Task**. This problem can be formulated as

$$\min_{x \in \mathbb{R}^m, y \in \mathbb{R}^d} \quad \mathbb{E}_{\xi \sim \mathcal{D}_{\text{val}}} \ell(y, \xi), \tag{19a}$$

$$\text{s.t.} \quad y \in \arg\min_{y' \in \mathbb{R}^d} \ell_{\text{tr}}(x, y') + \bar{\rho} \sum_{i=1}^{d} \frac{y_i'^2}{1 + y_i'^2} \tag{19b}$$

where $\ell(,)$ is the cross entropy loss function, $y_i'$ denotes the $i$th entry of $y'$, and $d$ is the dimension of the LL variable. The LL objective function is defined as $\ell_{\text{tr}}(x, y) \triangleq \sum_{i=1}^{m} \sigma(x_i)\ell(y, \xi_i)$, where $\xi_i \sim \mathcal{D}_{\text{tr}}$, $m$ denotes the total number of training data samples, $x_i$ denotes the $i$th entry of vector $x$ with dimension $m$. Here, $y$ denotes the weights of the neural network, including one hidden layer with parameters of size $10 \times 784$ and corresponding bias, and $\bar{\rho}$ is the nonconvex regularizer parameter. To ensure a fair comparison, we follow the numerical experiment setup from (Shen & Chen, 2023). Specifically, we use the MNIST dataset, splitting it into three parts: $5,000$ training samples, $5,000$ validation samples, and $10,000$ test samples. Additionally, $50\%$ of the training data samples are randomly assigned incorrect labels as polluted data.

**Experiment Setup**. In the experiments, we mainly compare the performance of the proposed TSP algorithm with three closely related algorithms: BOME (Liu et al., 2022), PBGD (Shen & Chen, 2023), and MEHA (Liu et al., 2024a). All these algorithms are variants of penalty methods, with MEHA being designed based on the Moreau envelope-based problem formulation, similar to TSP. Due to the similarity in algorithm structure, we use the same step-sizes and initial points for all tested algorithms. For TSP, we further choose the step-sizes for updating the dual variable $\lambda$ as $0.01$ and set $p = 1$. We also adopt the test accuracy to evaluate the quality of the obtained model parameter $y$ and the F1 score to measure the effectiveness of the hyperparameter $x$.

**Experiment Results and Discussion**. It can be observed from Figure 1(a) that TSP achieves the highest test accuracy among others, which is the major learning objective of this problem. This suggests that the model obtained by TSP generalizes well to the test dataset. Although MEHA also achieves high test accuracy, it is worth noting that the accuracy obtained by MEHA drops rapidly as the algorithm progresses. This is a major issue with penalty-based methods, as the penalty parameter continually increases to enforce the constraint, leading to overfitting of the model to the LL learning problem. Similar issues are evident in the results obtained by BOME and MEHA. However, it is not entirely fair to compare these two algorithms, as they are designed for cases where the LL loss function satisfies the PŁ condition, which is not the case here. Figure 1(b) shows the F1 scores obtained by these methods. Both TSP and MEHA exhibit similar results, further confirming that while the UL solutions may output similar results, the LL solutions can differ significantly as the LL optimization variable is optimized across the two levels. It is implied that the KKT solution, (which is further explained in the appendix), may provide more generalizable results. Figure 1(a) illustrates the convergence behavior of the tested algorithms during the training phase. It can be observed that TSP exhibits a faster convergence rate in terms of iterations compared to the oth-

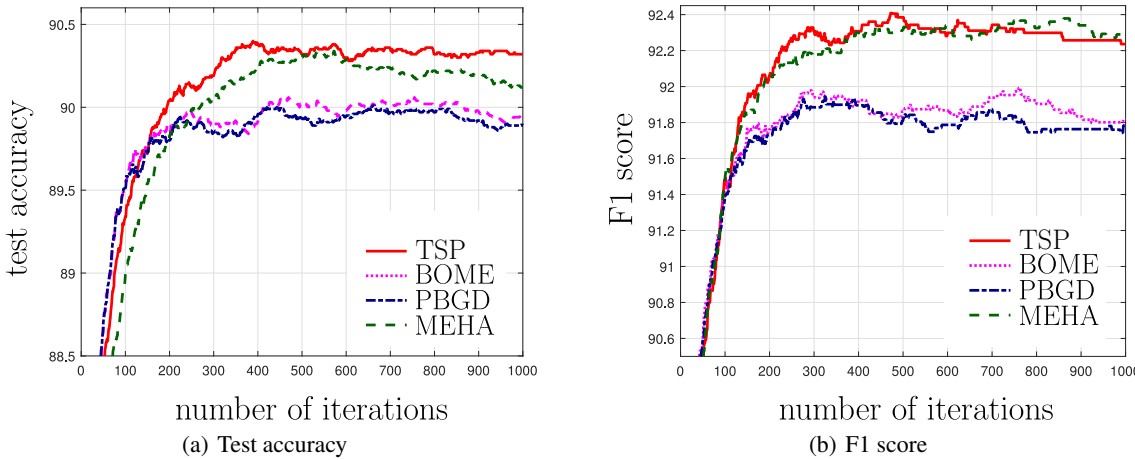

(a) Test accuracy

(b) F1 score

*Figure 1.* Convergence and generalization performance comparison of TSP, BOME (Liu et al., 2022), PBGD (Shen & Chen, 2023), and MEHA (Liu et al., 2024a) in solving the data hyper-cleaning problem.

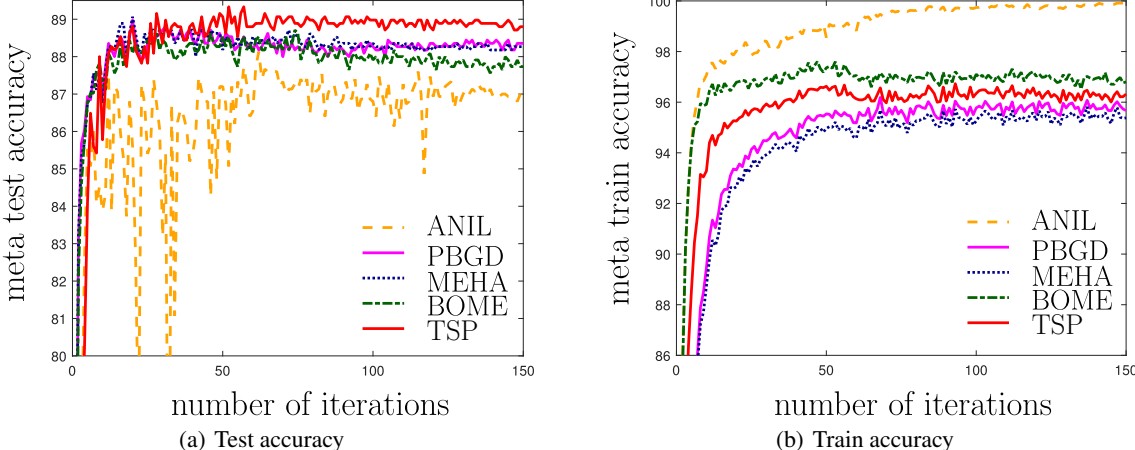

(a) Test accuracy

(b) Train accuracy

*Figure 2.* Convergence and generalization performance comparison of TSP, ANIL (Raghu et al., 2020), BOME (Liu et al., 2022), PBGD (Shen & Chen, 2023), and MEHA (Liu et al., 2024a) in solving the meta-leaning problem.

ers, which is generally attributed to the dual variable update that balances the two levels of the learning process. It is indeed well-known that primal-dual algorithms practically converge faster than penalty methods, although the theoretical analysis of this method is much more challenging than that of penalty-based approaches.

We further check the peak test accuracies obtained by these algorithms, which are as follows. TSP achieves a peak test accuracy of 90.4% with a variance of 0.0429% and a peak F1 score of 92.4% with a variance of 0.100%. In comparison, the peak test accuracy of BOME is 90.06% ± 0.23% with an F1 score of 91.99% ± 0.17%, the peak test accuracy of PBGD is 90.01% ± 0.23% with an F1 score of 91.95% ± 0.17%, and the peak test accuracy of MEHA is 90.34% ± 0.17% with an F1 score of 92.38% ± 0.12%.

**Representation Learning with Multi-Head Architectures**.
We further evaluate these algorithms on meta-learning problems using a multi-head neural network structure. A typical formulation can be written as follows.

$$\min_{x,\{y_{(i)}\}} \quad f(x,\{y_{(i)}\}) \triangleq \mathbb{E}_{\xi \sim \mathcal{D}_{\text{val}}}\left[\frac{1}{K}\sum_{i=1}^{K}\ell(x,y_{(i)};\xi)\right]$$
$$\text{s.t.} \quad y_{(i)} \in \arg\min_{y'_{(i)}} \mathbb{E}_{\xi \sim \mathcal{D}_{\text{tr}}^{(i)}}\ell(x,y'_{(i)};\xi), \quad \text{for } i \in [K].$$

Here, the UL problem involves a shared model parameter layer, denoted by $x$, which typically corresponds to the common feature encoder or backbone network shared across all tasks. The variable $y_{(i)}$ represents the task-specific head parameters, i.e., the final classification layer for task $i$, which is optimized using the task-specific training data $\mathcal{D}_{\text{tr}}^{(i)}$.

**Experiment Setup**. In this experiment, the shared hidden representation has a size of 32 and is followed by eight individual perceptron layers, each corresponding to a specific task. The MNIST dataset is partitioned into eight subsets based on digit labels, with each subset containing 2,500 training samples and 1,500 validation samples. Each task involves recognizing digits in a distinct way, where the data samples contain only one type of digit per task. We use samples labeled with digits 0 through 7: five digits are allocated

for training and validation, while the remaining three are used for both meta-training and meta-testing. Each meta-task corresponds to learning from one digit class, and each LL problem is associated with its own task-specific head and dual variable. We conduct an exhaustive search over initial step-sizes from $\{1, 0.1, 0.01, 0.001\}$ and penalty parameters from $\{1, 0.1, 0.01, 0.001, 0.0001\}$, reducing them in the order of $1/\sqrt{r}$. The gradients used in all implemented algorithms are stochastic, with a batch size of 32.

**Experiment Results and Discussion**. As shown in Figure 2, TSP achieves the best generalization performance, even though it does not attain the highest meta-training accuracy. The key role of TSP in this framework is to adjust the dual variables individually for each task-specific head. This flexibility helps balance the optimization dynamics between the shared and task-specific components, thereby improving the generalization capability of the learned representation across unseen tasks. This numerical example further highlights the advantage of solving bilevel learning problems using the TSP method and underscores the importance of its ability to achieve KKT solutions.

## 5. Concluding Remarks

In this work, we propose a single-loop structured gradient-based Lagrangian method for solving nonconvex stochastic bilevel optimization problems. By leveraging the Moreau envelope reformulation of the LL problem, our proposed method can find KKT points for this class of bilevel problems through a constrained optimization perspective, significantly expanding the scope for solving two-level machine learning problems. Our major contribution lies in establishing the high probability descent lemma with a dual error bound, enabling us to quantify the boundedness of the dual variable and conclude constraint satisfaction. Our theoretical analysis justifies that solving stochastic bilevel optimization problems can be as easy as solving single-level ones, measured by iteration complexity and KKT conditions.

## Acknowledgment

This research is supported in part by project #MMT-8115077 of the Shun Hing Institute of Advanced Engineering, The Chinese University of Hong Kong.

## Impact Statement

The major goal of this work was to develop computationally efficient machine learning algorithms with rigorous theoretical guarantees using optimization techniques, focusing on algorithm design and theoretical analysis in data science applications. Its main impact lies in deepening the understanding of learning algorithm dynamics, with broader implications for developing new theorem-proving methods for optimization algorithms. This research does not significantly involve ethical issues or social consequences.

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

# A. Preliminaries

In this section, we provide some technical preliminaries for the proofs of the lemmas and theorems claimed in the main body of this paper, including parameter definitions and supporting results.

## A.1. Notations

The definitions of the parameters and assumptions are further listed in Table 2.

*Table 2.* Summary of Definitions. ("Lips.": Lipschitz; "grad.": gradient; "const.": constant; "opt.": optimal; "·" represents the gradient is taken w.r.t. either $x$ or $y$.)

| A1 | Definition | Annotation |
|---|---|---|
| $L_f$ | $\|\nabla.f(x,y) - \nabla.f(x',y)\| \leq L_f\|x - x'\|$ | grad. Lips. const. of $f(x,y)$ w.r.t $x$ |
| | $\|\nabla.f(x,y) - \nabla.f(x,y')\| \leq L_f\|y - y'\|$ | grad. Lips. const. of $f(x,y)$ w.r.t $y$ |
| $L_g$ | $\|\nabla_y g(x,y) - \nabla_y g(x,y')\| \leq L_g\|y - y'\|$ | grad. Lips. const. of $g$ w.r.t. $y$ |

*Table 3.* Summary of Definitions. ("Lips.": Lipschitz; "grad.": gradient; "const.": constant; "opt.": optimal; "·" represents the gradient is taken w.r.t. either $x$ or $y$; "$[x;y]$" denotes the concatenation of $x$ and $y$; PEB: primal error bound; DEB: dual error bound.)

| Const. | Definition | Annotation |
|---|---|---|
| $\ell_f$ | $\|f(x,y) - f(x,y')\| \leq \ell_f\|y - y'\|$ | Lips. const. of $f(x,y)$ w.r.t. $y$ |
| $\ell_g$ | $\|g(x,y) - g(x,y')\| \leq \ell_g\|y - y'\|$ | Lips. const. of $g(x,y)$ w.r.t. $y$ |
| | $\|g(x,y) - g(x',y)\| \leq \ell_g\|x - x'\|$ | Lips. const. of $g(x,y)$ w.r.t. $x$ |
| $\ell_\gamma$ | $\|g_\gamma^\star(x,y) - g_\gamma^\star(x,y')\| \leq \ell_\gamma\|y - y'\|$ (cf. (32)) | Lips. const. of $g_\gamma^\star()$ w.r.t. $y$ |
| $L$ | $\|\nabla\mathcal{L}(x,y;\lambda) - \nabla\mathcal{L}(x',y';\lambda)\|$ $\leq L\|(x;y) - (x';y')\|$ (cf. (21)) | grad. Lips. const. of $\mathcal{L}()$ w.r.t. $x, y$ |
| $L_g$ | $\|\nabla.g(x,y) - \nabla.g(x',y)\| \leq L_g\|x - x'\|$ | grad. Lips. const. of $g(x,y)$ |
| $L_\gamma$ | $\|\nabla g_\gamma^\star(x,y) - \nabla g_\gamma^\star(x,y')\| \leq L_\gamma\|y - y'\|$ | Lips. const. of $\nabla g_\gamma^\star(x,y)$ w.r.t. $y$ |
| | $\|\nabla g_\gamma^\star(x,y) - \nabla g_\gamma^\star(x',y)\| \leq L_\gamma\|x - x'\|$ | Lips. const. of $\nabla g_\gamma^\star(x,y)$ w.r.t. $x$ |
| $L_z$ | $\|z^\star(x,y) - z^\star(x',y')\| \leq L_z\|(x,y) - (x',y')\|$ (cf. (30)) | Lips. const. of $z^\star()$ w.r.t. $x, y$ |
| $L_K$ | $L + p$ (cf. (22)) | grad. Lips. const. of $K()$ w.r.t. $x, y$ |
| $\sigma_1$ | $p(p - L)^{-1}$ (cf. (24e)) | const. of PEB w.r.t. $v$ |
| $\sigma_2$ | $(p + L)(p - L)^{-1}$ (cf. (26)) | const. of PEB w.r.t. $\lambda$ |
| $\sigma_3$ | $(p - L)^{-1}$ (cf. (27)) | const. of PEB w.r.t. $x^r$ or $y^r$ |
| $\sigma_\mathrm{w}$ | $(1 + \tau(2\ell_g + \ell_\gamma)\sigma_2)(\tau(p - L))^{-1}$ (cf. (233)) | const. of DEB |
| $v$ | $v \triangleq (\widehat{x}, \widehat{y})$ | abbreviation of $\widehat{x}, \widehat{y}$ |

In Section C.2, we will demonstrate that the iterates (such as $x^r, \bar{x}^\star(\widehat{x}^r, \widehat{y}^r), y^r, \bar{y}^\star(\widehat{x}^r, \widehat{y}^r)$) for which we need to evaluate the gradients or function values of $g(x,y), g_\gamma^\star(x,y)$, and $f(x,y)$ are bounded, which implies that the corresponding Lipschitz continuity holds. To be more precise, $\ell_f, L_g, \ell_g$ are listed in the section of notation in Table 2.

## A.2. Primal Error Bounds (PEBs)

Recall that the definition of $K(x, y, \widehat{x}, \widehat{y}; \lambda)$ is

$$K(x,y,\widehat{x},\widehat{y};\lambda) \triangleq f(x,y) + \lambda(g(x,y) - g_\gamma^\star(x,y) - \delta) + \frac{p}{2}\|x - \widehat{x}\|^2 + \frac{p}{2}\|y - \widehat{y}\|^2. \tag{20}$$

Based on the assumptions listed in Table 2, we have that $f(,), g(,)$ are gradient Lipschitz continuous. For simplicity of the presentation, we assume that $\rho \leq L_g$. Therefore, the Lagrangian $\mathcal{L}(x,y;\lambda)$ is gradient Lipschitz continuous with parameter,

given boundedness of $\lambda$,

$$L \triangleq L_f + \lambda \left( L_g + L_\gamma \right) \tag{21}$$

where $L_\gamma$ denotes the gradient Lipschitz constant of $g_\gamma(x, y)$.

Subsequently, function $K(x, y, \widehat{x}, \widehat{y}; \lambda)$ is strongly convex of $x$ and $y$ with parameter $p - L$ and gradient Lipschitz continuous with parameter

$$L_K \triangleq L + p. \tag{22}$$

The closed-form expressions of the (deterministic) gradient of the smoothed Lagrangian based on $\widehat{z}$ are as follows.

$$\nabla_x \widehat{K}(x, y, z, \widehat{x}, \widehat{y}; \lambda) \triangleq \nabla_x f(x, y) + \lambda \left( \nabla_x g(x, y) - \nabla_x g(x, z) \right) + p(x - \widehat{x}), \tag{23a}$$

$$\nabla_y \widehat{K}(x, y, z, \widehat{x}, \widehat{y}; \lambda) \triangleq \nabla_y f(x, y) + \lambda \left( \nabla_y g(x, y) + \frac{z - y}{\gamma} \right) + p(y - \widehat{y}), \tag{23b}$$

$$\nabla_z \widehat{K}(x, y, z, \widehat{x}, \widehat{y}; \lambda) \triangleq \nabla_y g(x, y) + \frac{z - y}{\gamma}. \tag{23c}$$

Given these properties and the assumptions listed in Table 2, we can obtain the following primal error bounds that have been studied in Lemma 5 in (Zhang et al., 2022), Lemma 3.5 and Lemma 3.10 in (Zhang & Luo, 2020) and Lemma B.2 in (Zhang et al., 2020). To be more specific, from Lemma B.2 in (Zhang et al., 2020) we have

$$\|y^\star(\widehat{x}, \widehat{y}; \lambda) - y^\star(\widehat{x}, \widehat{y}'; \lambda)\| \le \sigma_1 \|\widehat{y} - \widehat{y}'\|, \tag{24a}$$

$$\|x^\star(\widehat{x}, \widehat{y}; \lambda) - x^\star(\widehat{x}', \widehat{y}; \lambda)\| \le \sigma_1 \|\widehat{x} - \widehat{x}'\|, \tag{24b}$$

$$\|\bar{y}^\star(\widehat{x}, \widehat{y}) - \bar{y}^\star(\widehat{x}, \widehat{y}')\| \le \sigma_1 \|\widehat{y} - \widehat{y}'\|, \tag{24c}$$

$$\|\bar{x}^\star(\widehat{x}, \widehat{y}) - \bar{x}^\star(\widehat{x}', \widehat{y})\| \le \sigma_1 \|\widehat{x} - \widehat{x}'\|, \tag{24d}$$

where

$$\sigma_1 \triangleq \frac{p}{p - L}. \tag{24e}$$

Similarly, following Lemma B.2 in (Zhang et al., 2020), we can also have

$$\|y^\star(\widehat{x}, \widehat{y}; \lambda) - y^\star(\widehat{x}, \widehat{y}; \lambda')\| \le \sigma_2 \|\lambda - \lambda'\|, \tag{25a}$$

$$\|x^\star(\widehat{x}, \widehat{y}; \lambda) - x^\star(\widehat{x}, \widehat{y}; \lambda')\| \le \sigma_2 \|\lambda - \lambda'\|, \tag{25b}$$

where

$$\sigma_2 \triangleq \frac{p + L}{p - L}. \tag{26}$$

Let

$$\sigma_3 \triangleq \frac{1}{p - L}. \tag{27}$$

From Lemma 3.10 in (Zhang & Luo, 2020) or Lemma 5 in (Zhang et al., 2022) we can directly get

$$\|y^\star(\widehat{x}^r, \widehat{y}^r; \lambda^{r+1}) - y^r\| \le \frac{\sigma_3}{\beta} \|y^{r+1} - y^r\|, \tag{28a}$$

$$\|x^\star(\widehat{x}^r, \widehat{y}^r; \lambda^{r+1}) - x^r\| \le \frac{\sigma_3}{\alpha} \|x^{r+1} - x^r\|, \tag{28b}$$

$$\|y^\star(\widehat{x}^r, \widehat{y}^r; \lambda^{r+1}) - y^{r+1}\| \le \sigma_4 \|y^{r+1} - y^r\|, \tag{28c}$$

$$\|x^\star(\widehat{x}^r, \widehat{y}^r; \lambda^{r+1}) - x^{r+1}\| \le \sigma_5 \|x^{r+1} - x^r\|, \tag{28d}$$

where the primal error bounds (28b) and (28d) hold as $\widehat{K}$ (which is used for updating $x$) is also $(p - L)$-strongly convex and $(p + L)$-Lipschitz smooth, and

$$\sigma_4 \triangleq \frac{1 + \beta(p - L)}{\beta(p - L)}, \tag{29a}$$

$$\sigma_5 \triangleq \frac{1 + \alpha(p - L)}{\alpha(p - L)}. \tag{29b}$$

### A.3. Lipschitz Continuity

**Lemma 3.** *When $\gamma \in (0, 1/(2\rho))$ and function $g$ is gradient Lipschitz continuous with parameter $L_g$ and weakly convex with parameter $\rho$, $\|z^\star(x, y) - z^\star(x', y')\|$ is Lipschitz continuous, namely, there exist a constant $L_z$ such that*

$$\|z^\star(x, y) - z^\star(x', y')\| \leq L_z\|(x, y) - (x', y')\| \tag{30}$$

*where*

$$L_z \triangleq \frac{\max\left\{L_g, \frac{1}{\gamma}\right\}}{\frac{1}{\gamma} - \rho}. \tag{31}$$

*Further, when function $g$ is Lipschitz continuous with parameters $\ell_g$, function $g_\gamma(x, y)$ is Lipschitz continuous with parameter $\ell_{g_\gamma}$, namely, there exists a constant $\ell_{g_\gamma}$ such that*

$$|g_\gamma^\star(x, y) - g_\gamma^\star(x, y')| \leq \ell_{g_\gamma}\|y - y'\| \tag{32}$$

*where $\ell_{g_\gamma} \triangleq L_z(1 + \gamma L_g \ell_g)$. Also,*

$$|g_\gamma^\star(x, y) - g_\gamma^\star(x', y')| \leq \ell_{g_\gamma}\|(x, y) - (x', y')\| + L_g \ell_g\|x - x'\|. \tag{33}$$

*Proof.* From the optimality condition, we have

$$\nabla g(x, z^\star(x, y)) + \frac{1}{\gamma}(z^\star(x, y) - y) = 0, \tag{34}$$

$$\nabla g(x, z^\star(x, y')) + \frac{1}{\gamma}(z^\star(x, y') - y') = 0. \tag{35}$$

As function $g(x, z) + \frac{1}{2\gamma}\|z - y\|^2$ is strongly convex with parameter $1/(2\gamma) - 1/(2\rho)$ when $\gamma \in (0, 1/(2\rho))$, we can have

$$\left\langle \nabla g(x, z^\star(x, y)) + \frac{z^\star(x, y) - y}{\gamma} - \nabla g(x, z^\star(x', y')) - \frac{z^\star(x', y') - y}{\gamma}, z^\star(x, y) - z^\star(x', y') \right\rangle$$
$$\geq \left(\frac{1}{\gamma} - \rho\right)\|z^\star(x, y) - z^\star(x', y')\|^2, \tag{36}$$

which gives

$$\left(\frac{1}{\gamma} - \rho\right)\|z^\star(x, y) - z^\star(x', y')\|^2 \tag{37}$$
$$\leq \left\langle \nabla g(x', z^\star(x', y')) + \frac{z^\star(x', y') - y'}{\gamma} - \nabla g(x, z^\star(x', y')) - \frac{z^\star(x', y') - y}{\gamma}, z^\star(x, y) - z^\star(x', y') \right\rangle$$
$$\leq \left(L_g\|x - x'\| + \frac{1}{\gamma}\|y - y'\|\right)\|z^\star(x, y) - z^\star(x', y')\|. \tag{38}$$

Therefore, we have

$$\|z^\star(x, y) - z^\star(x', y')\| \leq L_z\|(x, y) - (x', y')\| \tag{39}$$

where $L_z$ is defined in (31).

Under assumption A1, we can have

$$g_\gamma^\star(x, y) - g_\gamma^\star(x, y')$$

$$= g(x, z^\star(x, y)) - g(x, z^\star(x, y')) + \frac{1}{2\gamma}\|z^\star(x, y) - y\| - \frac{1}{2\gamma}\|z^\star(x, y') - y'\|^2 \tag{40}$$

$$\leq |z^\star(x, y) - z^\star(x, y')| + \frac{1}{2\gamma}\langle z^\star(x, y) - y - (z^\star(x, y') - y'), z^\star(x, y) - y + z^\star(x, y') - y'\rangle$$

$$\leq L_z\|y - y'\| + \|\nabla g(x, z^\star(x, y)) - \nabla g(x, z^\star(x, y'))\|\gamma \ell_g \tag{41}$$

$$\leq L_z\|y - y'\| + \gamma L_g \ell_g \|z^\star(x, y) - z^\star(x, y')\| \tag{42}$$

$$\leq L_z(1 + \gamma L_g \ell_g)\|y - y'\|. \tag{43}$$

Similarly,

$$g_\gamma^\star(x, y) - g_\gamma^\star(x', y')$$

$$= g(x, z^\star(x, y)) - g(x', z^\star(x', y')) + \frac{1}{2\gamma}\|z^\star(x, y) - y\| - \frac{1}{2\gamma}\|z^\star(x', y') - y'\|^2 \tag{44}$$

$$\leq |z^\star(x, y) - z^\star(x', y')| + \frac{1}{2\gamma}\langle z^\star(x, y) - y - (z^\star(x', y') - y'), z^\star(x, y) - y + z^\star(x', y') - y'\rangle$$

$$\leq L_z\|(x, y) - (x', y')\| + \|\nabla g(x, z^\star(x, y)) - \nabla g(x', z^\star(x', y'))\|\gamma \ell_g \tag{45}$$

$$\leq L_z\|(x, y) - (x', y')\| + \gamma L_g \ell_g \left(\|z^\star(x, y) - z^\star(x', y')\| + \|x - x'\|\right) \tag{46}$$

$$\leq L_z(1 + \gamma L_g \ell_g)\|(x, y) - (x', y')\| + \gamma L_g \ell_g \|x - x'\|. \tag{47}$$

$\square$

---

**Lemma 4.** *Given $(x, y)$ and $z$ generated by TSP, when $\gamma \in (0, 1/(2\rho))$ and function $g$ is gradient Lipschitz continuous with parameter $L_g$ and weakly convex with parameter $\rho$, namely, there exists a constant $\ell_{g_z}$ such that*

$$\left|g_\gamma^\star(x, y) - g(x, z) - \frac{1}{2\gamma}\|z - y\|^2\right| \leq \ell_{g_z}\|z^\star(x, y) - z\| \tag{48}$$

---

*Proof.* Given any $x, y$, it implies that $z^\star(x, y)$ is bounded due to the strong convexity of the function when $\gamma \in (0, 1/(2\rho))$. From the update rule of $z$, it can be shown in (220) that sequence $z^r$ is also bounded.

$$g_\gamma^\star(x^r, y^r) - g(x^r, z^r) - \frac{1}{2\gamma}\|z^r - y^r\|^2$$

$$\leq |g(x^r, z^\star(x^r, y^r)) - g(x^r, z^r)| + \frac{1}{2\gamma}\left(\|z^\star(x^r, y^r) - y^r\|^2 - \|z^r - y^r\|^2\right) \tag{49}$$

$$\leq |g(x^r, z^\star(x^r, y^r)) - g(x^r, z^r)| + \frac{1}{2\gamma}\langle(z^\star(x^r, y^r) - y^r) - (z^r - y^r), (z^\star(x^r, y^r) - y^r) + (z^r - y^r)\rangle$$

$$\leq |g(x^r, z^\star(x^r, y^r)) - g(x^r, z^r)| + \frac{1}{2\gamma}\|(z^\star(x^r, y^r) - z^r\|\|(z^\star(x^r, y^r) - y^r) + (z^r - y^r)\|$$

$$\leq L_g\|z^\star(x^r, y^r) - z^r\| + \frac{1}{2\gamma}\|(z^\star(x^r, y^r) - z^r\|\|(z^\star(x^r, y^r) - y^r) + (z^r - y^r)\|$$

$$\overset{(a)}{\leq} L_{g_z}\|z^\star(x^r, y^r) - z^r\|$$

where $(a)$ holds due to the boundedness of sequence $\{z^r\}$. $\square$

For notational simplicity, we define $\ell_\gamma := \max\{\ell_{g_z}, \ell_{g_\gamma} + L_g \ell_g\}$.

## A.4. Stochasticity of Gradient Estimate Noise

Without loss of generality, we assume that $\max\{\sigma_{g_x}^2, \sigma_{g_y}^2, \sigma_{f_x}^2, \sigma_{f_y}^2, \sigma_{f_z}^2, \sigma_{\widehat{g}_y}^2, \sigma_{\widehat{g}_z}^2\} \leq \sigma^2$. Based on the assumption regarding stochastic noise in the gradient estimate, we establish the following relation between the iterates generated using the ground truth and their counterparts estimated with a fixed mini-batch of data samples, where the difference is bounded by $\varepsilon$. For example, for the update of the variable $y$, we have

$$
\mathbb{E}\left[y^{r+1} - y^r + \varepsilon_{f_y}^r + \lambda^{r+1}\varepsilon_{g_y}^r\right]
$$
$$
= -\beta\left(\nabla_y f(x^r, y^r) + \varepsilon_{f_g}^r + \lambda^r\left(\nabla_y g_y(x^r, y^r) + \varepsilon_{g_y}^r + \frac{z^{r+1} - y^r}{\gamma}\right) + p(y^r - \widehat{y}^r)\right) \tag{50}
$$
$$
= y^{r+1} - y^r \tag{51}
$$

where the expectation is taken over the underlying data distribution, conditioned on the historical iterates up to the current iteration $r$. In such a way, we can have

$$
y^{r+1} - y^r = \mathbb{E}\left[y^{r+1} - y^r\right] + \beta\varepsilon_y^r \tag{52}
$$

where $\varepsilon_y^r \triangleq \varepsilon_{f_y}^r + \lambda^{r+1}\varepsilon_{g_y}^r$.

Similarly, according to (23a), we also have

$$
x^{r+1} = x^r - \alpha\left(h_x^f(x^r, y^r) + \lambda^{r+1}\left(h_x^g(x^r, y^r) - h_x^g(x^r, z^{r+1})\right) + p(x^r - \widehat{x}^r)\right), \tag{53}
$$

which gives

$$
x^{r+1} - x^r = \mathbb{E}\left[x^{r+1} - x^r\right] + \alpha\varepsilon_x^r \tag{54}
$$

where $\varepsilon_x^r \triangleq \varepsilon_{f_x}^r + \lambda^{r+1}\left(\varepsilon_{g_y}^r - \varepsilon_{g_z}^r\right)$.

# B. Convergence Analysis

We now present the proofs, related results, and technical details that establish the lemmas and theorems of our convergence analysis.

## B.1. Descent Lemmas and Dual Ascent

### B.1.1. PRIMAL DESCENT LEMMA

**Lemma 5.** *(Primal Descent Lemma) Under A1-A5, suppose that the sequence $\{x^r, y^r, z^r, \widehat{x}^r, \widehat{y}^r, \lambda^r, \forall r\}$ is generated by TSP. When*

$$
0 < \alpha, \beta \leq \frac{1}{4L_K}, \quad 0 < \omega \leq 1, \quad and \tag{55}
$$

*then the primal descent inequality holds, namely,*

$$
K(x^{r+1}, y^{r+1}, \widehat{x}^{r+1}, \widehat{y}^{r+1}; \lambda^{r+1}) - K(x^r, y^r, \widehat{x}^r, \widehat{y}^r; \lambda^r)
$$
$$
\leq -\frac{1}{2\alpha}\left\|\mathbb{E}\left[x^{r+1} - x^r\right]\right\|^2 - \frac{1}{2\beta}\left\|\mathbb{E}\left[y^{r+1} - y^r\right]\right\|^2 - \frac{p}{2\omega}\|\widehat{x}^{r+1} - \widehat{x}^r\|^2 - \frac{p}{2\omega}\|\widehat{y}^{r+1} - \widehat{y}^r\|^2
$$
$$
+ \langle g(x^r, y^r) - g_\gamma^\star(x^r, y^r) - \delta, \lambda^{r+1} - \lambda^r\rangle + \frac{\beta}{\gamma^2}\|z^r - z^\star(x^r, y^r)\|^2 + \alpha(\lambda^{r+1}L_g)^2\|z^r - z^\star(x^r, y^{r+1})\|^2
$$
$$
- \alpha\left\langle\nabla_x K(x^r, y^{r+1}, \widehat{x}^r, \widehat{y}^{r+1}; \lambda^{r+1}), \varepsilon_x^r\right\rangle + L_K\alpha^2\|\varepsilon_x^r\|^2
$$
$$
- \beta\langle\nabla_y K(x^r, y^r, \widehat{x}^r, \widehat{y}^r; \lambda^{r+1}), \varepsilon_y^r\rangle + L_K\beta^2\|\varepsilon_y^r\|^2 \tag{56}
$$

*where $\varepsilon_x^r \triangleq \varepsilon_{f_x}^r + \lambda^{r+1}\varepsilon_{g_x}^r$, and $\varepsilon_y^r \triangleq \varepsilon_{f_y}^r + \lambda^{r+1}\varepsilon_{g_y}^r$.*

*Proof.* *y-update:* From (9), one step of gradient step gives

$$K(x^r, y^{r+1}, \widehat{x}^r, \widehat{y}^r; \lambda^{r+1}) - K(x^r, y^r, \widehat{x}^r, \widehat{y}^r; \lambda^{r+1})$$

$$\overset{(a)}{\leq} \left\langle \nabla K(x^r, y^r, \widehat{x}^r, \widehat{y}^r; \lambda^{r+1}), y^{r+1} - y^r \right\rangle + \frac{L_K}{2} \left\| y^{r+1} - y^r \right\|^2 \tag{57}$$

$$\leq \left\langle \nabla K(x^r, y^r, \widehat{x}^r, \widehat{y}^r; \lambda^{r+1}), \mathbb{E}\left[y^{r+1} - y^r\right] \right\rangle + \frac{L_K}{2} \left\| \mathbb{E}\left[y^{r+1} - y^r\right] + \varepsilon_y^r \right\|^2 \tag{58}$$

$$\leq -\frac{1}{2\beta} \left\| \mathbb{E}\left[y^{r+1} - y^r\right] \right\|^2 + \frac{\beta}{\gamma^2} \|z^r - z^\star(x^r, y^r)\|^2$$

$$- \beta \left\langle \nabla_y K(x^r, y^r, \widehat{x}^r, \widehat{y}^r; \lambda^{r+1}), \varepsilon_y^r \right\rangle + L_K \beta^2 \|\varepsilon_y^r\|^2 \tag{59}$$

where $(a)$ holds due to the Lipschitz gradient continuity,

$$\left\langle \nabla_y \widehat{K}(x^r, y^r, z^r, \widehat{x}^r, \widehat{y}^r; \lambda^{r+1}), \mathbb{E}\left[y^{r+1} - y^r\right] \right\rangle$$

$$+ \left\langle \nabla_y K(x^r, y^r, \widehat{x}^r, \widehat{y}^r; \lambda^{r+1}) - \nabla_y \widehat{K}(x^r, y^r, z^r, \widehat{x}^r, \widehat{y}^r; \lambda^{r+1}), \mathbb{E}\left[y^{r+1} - y^r\right] \right\rangle$$

$$\overset{(9)}{\leq} -\frac{1}{\beta} \left\| \mathbb{E}\left[y^{r+1} - y^r\right] \right\|^2$$

$$+ \beta \left\| \nabla_y K(x^r, y^r, \widehat{x}^r, \widehat{y}^r; \lambda^{r+1}) - \nabla_y \widehat{K}(x^r, y^r, z^r, \widehat{x}^r, \widehat{y}^r; \lambda^{r+1}) \right\|^2 + \frac{1}{4\beta} \left\| \mathbb{E}\left[y^{r+1} - y^r\right] \right\|^2 \tag{60}$$

$$\leq -\frac{3}{4\beta} \left\| \mathbb{E}\left[y^{r+1} - y^r\right] \right\|^2 + \beta \left\| \frac{z^r - y^r}{\gamma} - \frac{z^\star(x^r, y^r) - y^r}{\gamma} \right\|^2 \tag{61}$$

$$\leq -\frac{3}{4\beta} \left\| \mathbb{E}\left[y^{r+1} - y^r\right] \right\|^2 + \frac{\beta}{\gamma^2} \|z^r - z^\star(x^r, y^r)\|^2 \tag{62}$$

and $\beta \leq 1/(4L_K)$.

*x-update*: The update of $x$ shown in (11), which is the similar as the $x$-update, gives

$$K(x^{r+1}, y^{r+1}, \widehat{x}^r, \widehat{y}^{r+1}; \lambda^{r+1}) - K(x^r, y^{r+1}, \widehat{x}^r, \widehat{y}^{r+1}; \lambda^{r+1})$$

$$\overset{(a)}{\leq} \left\langle \nabla K(x^r, y^{r+1}, \widehat{x}^r, \widehat{y}^{r+1}; \lambda^{r+1}), \mathbb{E}\left[x^{r+1} - x^r\right] \right\rangle + \frac{L_K}{2} \left\| \mathbb{E}\left[x^{r+1} - x^r\right] \right\|^2 \tag{63}$$

$$\leq \left\langle \nabla \widehat{K}(x^r, y^{r+1}, z^{r+1}, \widehat{x}^r, \widehat{y}^{r+1}; \lambda^{r+1}), \mathbb{E}\left[x^{r+1} - x^r\right] \right\rangle + \frac{L_K}{2} \left\| \left[x^{r+1} - x^r\right] \right\|^2$$

$$+ \left\langle \nabla K(x^r, y^{r+1}, \widehat{x}^r, \widehat{y}^{r+1}; \lambda^{r+1}) - \nabla \widehat{K}(x^r, y^{r+1}, z^{r+1}, \widehat{x}^r, \widehat{y}^{r+1}; \lambda^{r+1}), \mathbb{E}\left[x^{r+1} - x^r\right] \right\rangle$$

$$- \alpha \left\langle \nabla_x K(x^r, y^{r+1}, \widehat{x}^r, \widehat{y}^{r+1}; \lambda^{r+1}), \varepsilon_x^r \right\rangle \tag{64}$$

where $(a)$ follows from the gradient Lipschitz continuity of $K(x, y, \widehat{x}, \widehat{y}; \lambda)$ with Lipschitz constant $L_K$.

From the optimality condition of (11), we have

$$\left\langle \nabla_x \widehat{K}(x^r, y^{r+1}, z^{r+1}, \widehat{x}^r, \widehat{y}^{r+1}; \lambda^{r+1}), \mathbb{E}\left[x^{r+1} - x^r\right] \right\rangle \leq -\frac{1}{\alpha} \left\| \mathbb{E}\left[x^{r+1} - x^r\right] \right\|^2. \tag{65}$$

Regarding the last term at the right-hand side (RHS) of (64), we have

$$\left\langle \nabla K(x^r, y^{r+1}, \widehat{x}^r, \widehat{y}^{r+1}; \lambda^{r+1}) - \nabla \widehat{K}(x^r, y^{r+1}, z^{r+1}, \widehat{x}^r, \widehat{y}^{r+1}; \lambda^{r+1}), \mathbb{E}\left[x^{r+1} - x^r\right] \right\rangle$$

$$\leq \alpha \|\nabla K(x^r, y^{r+1}, \widehat{x}^r, \widehat{y}^{r+1}; \lambda^{r+1}) - \nabla \widehat{K}(x^r, y^{r+1}, z^{r+1}, \widehat{x}^r, \widehat{y}^{r+1}; \lambda^{r+1}\|^2 + \frac{1}{4\alpha} \left\| \mathbb{E}\left[x^{r+1} - x^r\right] \right\|^2$$

where we apply Young's inequality with parameter 2.

For the first term at the RHS of the above inequality, we can further have

$$\|\nabla K(x^r, y^{r+1}, \widehat{x}^r, \widehat{y}^{r+1}; \lambda^{r+1}) - \nabla \widehat{K}(x^r, y^{r+1}, z^r, \widehat{x}^r, \widehat{y}^{r+1}; \lambda^{r+1})\|^2$$

$$\leq (\lambda^{r+1})^2 \|\nabla g(x^r, z^\star(x^r, y^{r+1})) - \nabla g(x^r, z^r)\|^2 \tag{66}$$

$$\leq (\lambda^{r+1} L_g)^2 \|z^r - z^\star(x^r, y^{r+1})\|^2. \tag{67}$$

Substituting (65) and (67) back to (64) gives

$$K(x^{r+1}, y^{r+1}, \widehat{x}^r, \widehat{y}^{r+1}; \lambda^{r+1}) - K(x^r, y^{r+1}, \widehat{x}^r, \widehat{y}^{r+1}; \lambda^{r+1})$$

$$\leq -\frac{1}{2\alpha} \left\| \mathbb{E}\left[ x^{r+1} - x^r \right] \right\|^2 + \alpha(\lambda^{r+1} L_g)^2 \|z^r - z^\star(x^r, y^{r+1})\|^2$$

$$- \alpha \left\langle \nabla_x K(x^r, y^{r+1}, \widehat{x}^r, \widehat{y}^{r+1}; \lambda^{r+1}), \varepsilon_x^r \right\rangle + L_K \alpha^2 \|\varepsilon_x^r\|^2 \tag{68}$$

where we select $\alpha \leq 1/(4L_K)$.

$\widehat{x}$*-update:* From (12), we have

$$K(x^{r+1}, y^{r+1}, \widehat{x}^{r+1}, \widehat{y}^{r+1}; \lambda^{r+1}) - K(x^{r+1}, y^{r+1}, \widehat{x}^r, \widehat{y}^{r+1}; \lambda^{r+1})$$

$$= \frac{p}{2} \left( \|x^{r+1} - \widehat{x}^{r+1}\|^2 - \|x^{r+1} - \widehat{x}^r\|^2 \right) \tag{69}$$

$$= \frac{p}{2} \langle \widehat{x}^r - \widehat{x}^{r+1}, x^{r+1} - \widehat{x}^{r+1} + x^{r+1} - \widehat{x}^r \rangle \tag{70}$$

$$\overset{(a)}{\leq} -\frac{p}{2\omega} \|\widehat{x}^{r+1} - \widehat{x}^r\|^2 \tag{71}$$

where $(a)$ holds due to

$$\langle \widehat{x}^r - \widehat{x}^{r+1}, x^{r+1} - \widehat{x}^{r+1} + x^{r+1} - \widehat{x}^r \rangle$$

$$= \langle \widehat{x}^r - \widehat{x}^{r+1}, x^{r+1} - \widehat{x}^r + \widehat{x}^r - \widehat{x}^{r+1} + x^{r+1} - \widehat{x}^r \rangle = \left( 1 - \frac{2}{\omega} \right) \|\widehat{x}^{r+1} - \widehat{x}^r\|^2 \tag{72}$$

and (12) for $0 < \omega \leq 1$.

$\widehat{y}$*-update:* Similar to the $\widehat{x}$-update. From (10), we have

$$K(x^r, y^{r+1}, \widehat{x}^r, \widehat{y}^{r+1}; \lambda^{r+1}) - K(x^r, y^{r+1}, \widehat{x}^r, \widehat{y}^r; \lambda^{r+1}) \leq -\frac{p}{2\omega} \|\widehat{y}^{r+1} - \widehat{y}^r\|^2.$$

$\lambda$*-update:* After the dual variable is updated, we have

$$K(x^r, y^r, \widehat{x}^r, \widehat{y}^r; \lambda^{r+1}) - K(x^r, y^r, \widehat{x}^r, \widehat{y}^r; \lambda^r) = \langle g(x^r, y^r) - g_\gamma^\star(x^r, y^r) - \delta, \lambda^{r+1} - \lambda^r \rangle. \tag{73}$$

$\square$

### B.1.2. DUAL ASCENT LEMMA

**Lemma 6.** *(Dual Ascent) Under A1-A5, suppose that the sequence $\{x^r, y^r, z^r, \widehat{x}^r, \widehat{y}^r, \lambda^r, \forall r\}$ is generated by TSP. When $p > L$, the dual ascent inequality holds, namely,*

$$D(\widehat{x}^{r+1}, \widehat{y}^{r+1}; \lambda^{r+1}) - D(\widehat{x}^r, \widehat{y}^r; \lambda^r)$$

$$\geq \Big\langle \lambda^{r+1} - \lambda^r, g(x^\star(\widehat{x}^r, \widehat{y}^r; \lambda^{r+1}), y^\star(\widehat{x}^r, \widehat{y}^r; \lambda^{r+1}))$$

$$- g_\gamma^\star(x^\star(\widehat{x}^r, \widehat{y}^r; \lambda^{r+1}), y^\star(\widehat{x}^r, \widehat{y}^r; \lambda^{r+1})) - \delta \Big\rangle$$

$$+ \frac{p}{2} \langle \widehat{y}^{r+1} - \widehat{y}^r, \widehat{y}^{r+1} + \widehat{y}^r - 2y^\star(\widehat{x}^r, \widehat{y}^{r+1}; \lambda^{r+1}) \rangle$$

$$+ \frac{p}{2} \langle \widehat{x}^{r+1} - \widehat{x}^r, \widehat{x}^{r+1} + \widehat{x}^r - 2x^\star(\widehat{x}^{r+1}, \widehat{y}^{r+1}; \lambda^{r+1}) \rangle. \tag{74}$$

*Proof.* Recall

$$K(x, y, \widehat{x}, \widehat{y}; \lambda) \triangleq f(x, y) + \lambda(g(x, y) - g_\gamma^\star(x, y) - \delta) + \frac{p}{2} \|x - \widehat{x}\|^2 + \frac{p}{2} \|y - \widehat{y}\|^2.$$

We have

$$
D(\widehat{x}^r, \widehat{y}^{r+1}; \lambda^{r+1}) - D(\widehat{x}^r, \widehat{y}^r; \lambda^{r+1})
$$
$$
= K(x^\star(\widehat{x}^r, \widehat{y}^{r+1}; \lambda^{r+1}), y^\star(\widehat{x}^r, \widehat{y}^{r+1}; \lambda^{r+1}), \widehat{x}^r, \widehat{y}^{r+1}; \lambda^{r+1})
$$
$$
- K(x^\star(\widehat{x}^r, \widehat{y}^r; \lambda^{r+1}), y^\star(\widehat{x}^r, \widehat{y}^r; \lambda^{r+1}), \widehat{x}^r, \widehat{y}^r; \lambda^{r+1}) \tag{75}
$$
$$
\overset{(15)}{\geq} K(x^\star(\widehat{x}^r, \widehat{y}^{r+1}; \lambda^{r+1}), y^\star(\widehat{x}^r, \widehat{y}^{r+1}; \lambda^{r+1}), \widehat{x}^r, \widehat{y}^{r+1}; \lambda^{r+1})
$$
$$
- K(x^\star(\widehat{x}^r, \widehat{y}^{r+1}; \lambda^{r+1}), y^\star(\widehat{x}^r, \widehat{y}^{r+1}; \lambda^{r+1}), \widehat{x}^r, \widehat{y}^r; \lambda^{r+1}) \tag{76}
$$
$$
\overset{(6)}{=} \frac{p}{2} \left( \|y^\star(\widehat{x}^r, \widehat{y}^{r+1}; \lambda^{r+1}) - \widehat{y}^{r+1}\|^2 - \|y^\star(\widehat{x}^r, \widehat{y}^{r+1}; \lambda^{r+1}) - \widehat{y}^r\|^2 \right) \tag{77}
$$
$$
= \frac{p}{2} \langle \widehat{y}^{r+1} - \widehat{y}^r, \widehat{y}^{r+1} + \widehat{y}^r - 2y^\star(\widehat{x}^r, \widehat{y}^{r+1}; \lambda^{r+1}) \rangle. \tag{78}
$$

Similarly, we can obtain

$$
D(\widehat{x}^{r+1}, \widehat{y}^{r+1}; \lambda^{r+1}) - D(\widehat{x}^r, \widehat{y}^{r+1}; \lambda^{r+1})
$$
$$
= K(x^\star(\widehat{x}^{r+1}, \widehat{y}^{r+1}; \lambda^{r+1}), y^\star(\widehat{x}^{r+1}, \widehat{y}^{r+1}; \lambda^{r+1}), \widehat{x}^{r+1}, \widehat{y}^{r+1}; \lambda^{r+1})
$$
$$
- K(x^\star(\widehat{x}^r, \widehat{y}^{r+1}; \lambda^{r+1}), y^\star(\widehat{x}^r, \widehat{y}^{r+1}; \lambda^{r+1}), \widehat{x}^r, \widehat{y}^{r+1}; \lambda^{r+1}) \tag{79}
$$
$$
\geq K(x^\star(\widehat{x}^{r+1}, \widehat{y}^{r+1}; \lambda^{r+1}), y^\star(\widehat{x}^{r+1}, \widehat{y}^{r+1}; \lambda^{r+1}), \widehat{x}^{r+1}, \widehat{y}^{r+1}; \lambda^{r+1})
$$
$$
- K(x^\star(\widehat{x}^{r+1}, \widehat{y}^{r+1}; \lambda^{r+1}), y^\star(\widehat{x}^{r+1}, \widehat{y}^{r+1}; \lambda^{r+1}), \widehat{x}^r, \widehat{y}^{r+1}; \lambda^{r+1}) \tag{80}
$$
$$
= \frac{p}{2} \left( \|x^\star(\widehat{x}^{r+1}, \widehat{y}^{r+1}; \lambda^{r+1}) - \widehat{x}^{r+1}\|^2 - \|x^\star(\widehat{x}^{r+1}, \widehat{y}^{r+1}; \lambda^{r+1}) - \widehat{x}^r\|^2 \right) \tag{81}
$$
$$
= \frac{p}{2} \langle \widehat{x}^{r+1} - \widehat{x}^r, \widehat{x}^{r+1} + \widehat{x}^r - 2x^\star(\widehat{x}^{r+1}, \widehat{y}^{r+1}; \lambda^{r+1}) \rangle. \tag{82}
$$

Then, we can have

$$
D(\widehat{x}^r, \widehat{y}^r; \lambda^{r+1}) - D(\widehat{x}^r, \widehat{y}^r; \lambda^r)
$$
$$
= K(x^\star(\widehat{x}^r, \widehat{y}^r; \lambda^{r+1}), y^\star(\widehat{x}^r, \widehat{y}^r; \lambda^{r+1}), \widehat{x}^r, \widehat{y}^r; \lambda^{r+1})
$$
$$
- K(x^\star(\widehat{x}^r, \widehat{y}^r; \lambda^r), y^\star(\widehat{x}^r, \widehat{y}^r; \lambda^r), \widehat{x}^r, \widehat{y}^r; \lambda^r) \tag{83}
$$
$$
\overset{(a)}{\geq} K(x^\star(\widehat{x}^r, \widehat{y}^r; \lambda^{r+1}), y^\star(\widehat{x}^r, \widehat{y}^r; \lambda^{r+1}), \widehat{x}^r, \widehat{y}^r; \lambda^{r+1})
$$
$$
- K(x^\star(\widehat{x}^r, \widehat{y}^r; \lambda^{r+1}), y^\star(\widehat{x}^r, \widehat{y}^r; \lambda^{r+1}), \widehat{x}^r, \widehat{y}^r; \lambda^r) \tag{84}
$$
$$
= \langle \lambda^{r+1} - \lambda^r, g(x^\star(\widehat{x}^r, \widehat{y}^r; \lambda^{r+1}), y^\star(\widehat{x}^r, \widehat{y}^r; \lambda^{r+1})) - g_\gamma^\star(x^\star(\widehat{x}^r, \widehat{y}^r; \lambda^{r+1}), y^\star(\widehat{x}^r, \widehat{y}^r; \lambda^{r+1})) - \delta \rangle \tag{85}
$$

where in $(a)$ we use the definition of $y^\star(\widehat{x}^r, \widehat{y}^r; \lambda^{r+1})$ for $p > L$.

Combining all the above gives the desired result. $\qquad \square$

### B.1.3. PROXIMAL DESCENT LEMMA

**Lemma 7.** *(Proximal Descent) Under A1-A5, suppose that the sequence $\{x^r, y^r, z^r, \widehat{x}^r, \widehat{y}^r, \lambda^r, \forall r\}$ is generated by TSP. Assume that $\bar{y}^\star(\widehat{x}^r, \widehat{y}^r)$ is bounded and $p > L$, then the proximal descent inequality holds, namely,*

$$
P(\widehat{x}^{r+1}, \widehat{y}^{r+1}) - P(\widehat{x}^r, \widehat{y}^r)
$$
$$
\leq p(\widehat{y}^{r+1} - \widehat{y}^r)^T(\widehat{y}^r - \bar{y}^\star(\widehat{x}^r, \widehat{y}^r)) + p(\widehat{x}^{r+1} - \widehat{x}^r)^T(\widehat{x}^r - \bar{x}^\star(\widehat{x}^r, \widehat{y}^{r+1}))
$$
$$
+ \frac{p}{2}\left(\frac{p}{p-L} + 1\right)\left(\|\widehat{y}^{r+1} - \widehat{y}^r\|^2 + \|\widehat{x}^{r+1} - \widehat{x}^r\|^2\right). \tag{86}
$$

*Proof.* First, note that $K(x, y, \widehat{x}, \widehat{y}; \lambda)$ is strongly convex w.r.t. $x$ and $y$ jointly with parameter $p - L$. Under A1-A3 and the assumption that $\bar{y}^\star(\widehat{x}^r, \widehat{y}^r)$ is bounded, we can obtain that $\nabla_{\widehat{y}} P(\widehat{x}^r, \widehat{y}^r) = p(\widehat{y}^r - \bar{y}^\star(\widehat{x}^r, \widehat{y}^r))$ by applying the Danskin's theorem in the convex analysis (Tyrrell, 1996; Clarke, 1975). Then, using the primal error bound (24c), we can show that

$\nabla_{\widehat{y}} P(\widehat{x}^r, \widehat{y}^r)$ has a Lipschitz constant, i.e.,

$$\|\nabla P(\widehat{x}^r, \widehat{y}^{r+1}) - \nabla P(\widehat{x}^r, \widehat{y}^r)\| \le p\left(\frac{p}{p-L} + 1\right)\|\widehat{y}^{r+1} - \widehat{y}^r\|. \tag{87}$$

Therefore, it is straightforward that

$$P(\widehat{x}^r, \widehat{y}^{r+1}) - P(\widehat{x}^r, \widehat{y}^r) \le p(\widehat{y}^{r+1} - \widehat{y}^r)^T(\widehat{y}^r - \bar{y}^\star(\widehat{x}^r, \widehat{y}^r)) + \frac{p}{2}\left(\frac{p}{p-L} + 1\right)\|\widehat{y}^{r+1} - \widehat{y}^r\|^2. \tag{88}$$

Similarly, we have

$$\|\nabla_{\widehat{x}} P(\widehat{x}^{r+1}, \widehat{y}^{r+1}) - \nabla_{\widehat{x}} P(\widehat{x}^r, \widehat{y}^{r+1})\| \le p\left(\frac{p}{p-L} + 1\right)\|\widehat{x}^{r+1} - \widehat{x}^r\|, \tag{89}$$

which gives

$$\begin{aligned} &P(\widehat{x}^{r+1}, \widehat{y}^{r+1}) - P(\widehat{x}^r, \widehat{y}^{r+1}) \\ &\le p(\widehat{x}^{r+1} - \widehat{x}^r)^T(\widehat{x}^r - \bar{x}^\star(\widehat{x}^r, \widehat{y}^{r+1})) + \frac{p}{2}\left(\frac{p}{p-L} + 1\right)\|\widehat{x}^{r+1} - \widehat{x}^r\|^2. \end{aligned} \tag{90}$$

$\square$

## B.2. Proof of Potential Function

**Lemma 8.** *Assume that A1-A5 are satisfied. Suppose that the sequence $\{x^r, y^r, z^r, \widehat{x}^r, \widehat{y}^r, \lambda^r, \forall r\}$ is generated by TSP, $p > L$, and $\lambda^r, y^r, \bar{y}^\star(\widehat{x}^r, \widehat{y}^r)$ are bounded. When $\alpha, \beta, \omega$ respectively satisfy (55), then, there exists a constant $\zeta$ such that*

$$\begin{aligned} &Q(x^{r+1}, y^{r+1}, z^{r+1}, \widehat{x}^{r+1}, \widehat{y}^{r+1}; \lambda^{r+1}) - Q(x^r, y^r, z^r, \widehat{x}^r, \widehat{y}^r; \lambda^r) \\ &\le -\left(\frac{1}{2\alpha} - C_{h_x} - 16\mu(\ell_g^2 + L_\gamma^2)\frac{\sigma_3^2}{\alpha^2} - C_z\left(2L_z^2 + \frac{\eta L_z}{2}\right)\right)\|\mathbb{E}[x^{r+1} - x^r]\|^2 - \frac{1}{8\mu\tau}\|\lambda_+^r(\widehat{x}^r, \widehat{y}^r) - \lambda^r\|^2 \\ &\quad -\left(\frac{1}{2\beta} - C_{h_y} - 16\mu(\ell_g^2 + L_\gamma^2)\frac{\sigma_3^2}{\beta^2} - 2\alpha(\lambda^{r+1} L_g)^2 L_z^2 - C_z\left(2L_z^2 + \frac{\eta L_z}{2}\right)\right)\|\mathbb{E}[y^{r+1} - y^r]\|^2 \\ &\quad - p\left(\frac{1}{2\omega} - \left(\frac{1}{\zeta} + \frac{4p}{p-L}\right)\right)\|\widehat{x}^{r+1} - \widehat{x}^r\|^2 - p\left(\frac{1}{2\omega} - \left(\frac{1}{\zeta} + \frac{4p}{p-L}\right) - 6\zeta\sigma_1^2\right)\|\widehat{y}^{r+1} - \widehat{y}^r\|^2 \\ &\quad + 6p\zeta\left(\|x^\star(\widehat{x}^r, \widehat{y}^r; \lambda_+^r(\widehat{x}^r, \widehat{y}^r)) - \bar{x}^\star(\widehat{x}^r, \widehat{y}^r)\|^2 + \|y^\star(\widehat{x}^r, \widehat{y}^r; \lambda_+^r(\widehat{x}^r, \widehat{y}^r)) - \bar{y}^\star(\widehat{x}^r, \widehat{y}^r)\|^2\right) \\ &\quad - (1 - \vartheta)\frac{\tau}{2\mu}\|h^{r+1} - \nabla_\lambda K(x^\star(v^r; \lambda^r), y^\star(v^r; \lambda^r), v^r; \lambda^r)\|^2 - (1 - \varphi)C_z\|z^r - z^\star(x^r, y^r)\|^2 + n_Q^r \end{aligned} \tag{91}$$

*where $C_z$ is defined in (143), potential function*

$$\begin{aligned} Q^r &\triangleq Q(x^r, y^r, z^r, \widehat{x}^r, \widehat{y}^r; \lambda^r) \triangleq K(x^r, y^r, z^r, \widehat{x}^r, \widehat{y}^r; \lambda^r) - 2D(\widehat{x}^r, \widehat{y}^r; \lambda^r) + 2P(\widehat{x}^r, \widehat{y}^r) - \frac{1}{c}M(\lambda^r, h^r) \\ &\quad + \frac{\tau}{2\mu}\|h^{r+1} - \nabla_\lambda K(x^\star(v^r; \lambda^r), y^\star(v^r; \lambda^r), v^r; \lambda^r)\|^2 + C_z\|z^r - z^\star(x^r, y^r)\|^2 - \underline{f} \end{aligned} \tag{92}$$

*and*

$$\begin{aligned} n_Q^r &\triangleq n_Q^r(\alpha, \beta, \tau, \eta, \theta, \varepsilon_{\widehat{g}_y}^r, \varepsilon_{\widehat{g}_z}^r, \varepsilon_x^r, \varepsilon_y^r, \varepsilon_{g_z}^r) \\ &\triangleq -\alpha\left\langle\nabla_x K(x^r, y^{r+1}, \widehat{x}^r, \widehat{y}^{r+1}; \lambda^{r+1}), \varepsilon_x^r\right\rangle + L_K\alpha^2\|\varepsilon_x^r\|^2 - \beta\langle\nabla_y K(x^r, y^r, \widehat{x}^r, \widehat{y}^r; \lambda^{r+1}), \varepsilon_y^r\rangle + L_K\beta^2\|\varepsilon_y^r\|^2 \\ &\quad + (16\sigma_2^2 p\zeta + C_{h_\lambda})\frac{\tau^2}{1-\theta}\max_r\|\varepsilon_{\widehat{g}_y}^r - \varepsilon_{\widehat{g}_z}^r\|^2 + C_z n_{z_2}^r(\eta, \varepsilon_x^r, \varepsilon_y^r, \varepsilon_{g_z}^r) \\ &\quad + \langle\varepsilon_{\widehat{g}_y}^r - \varepsilon_{\widehat{g}_z}^r, \lambda^{r+1} - \lambda^r\rangle + \frac{\tau}{2\mu}\left(\theta(\varepsilon_{\widehat{g}_y}^r - \varepsilon_{\widehat{g}_z}^r)(h_\theta^r + \theta(g(x^r, y^r)) + \theta^2(\varepsilon_{\widehat{g}_y}^r - \varepsilon_{\widehat{g}_z}^r)^2\right). \end{aligned} \tag{93}$$

*Proof.* Recall that

$$E^r = K(x^r, y^r, z^r, \widehat{x}^r, \widehat{y}^r; \lambda^r) - 2D(\widehat{x}^r, \widehat{y}^r; \lambda^r) + 2P(\widehat{x}^r, \widehat{y}^r). \tag{94}$$

Merging (56), (74), and (86) gives

$$
\begin{aligned}
E^{r+1} &- E^r \\
\leq &-\frac{1}{2\alpha} \left\| \mathbb{E}\left[x^{r+1} - x^r\right] \right\|^2 - \frac{1}{2\beta} \left\| \mathbb{E}\left[y^{r+1} - y^r\right] \right\|^2 \\
&- \left(\frac{p}{2\omega} - 2p\frac{p}{p-L}\right) \|\widehat{x}^{r+1} - \widehat{x}^r\|^2 - \left(\frac{p}{2\omega} - 2p\frac{p}{p-L}\right) \|\widehat{y}^{r+1} - \widehat{y}^r\|^2 \\
&+ \langle g(x^r, y^r) - g_\gamma^\star(x^r, y^r) - \delta, \lambda^{r+1} - \lambda^r \rangle \\
&+ \frac{\beta}{\gamma^2} \|z^r - z^\star(x^r, y^r)\|^2 + \alpha(\lambda^{r+1} L_g)^2 \|z^r - z^\star(x^r, y^{r+1})\|^2 \\
&- 2\langle \lambda^{r+1} - \lambda^r, g(x^\star(\widehat{x}^r, \widehat{y}^r; \lambda^{r+1}), y^\star(\widehat{x}^r, \widehat{y}^r; \lambda^{r+1})) - g_\gamma^\star(x^\star(\widehat{x}^r, \widehat{y}^r; \lambda^{r+1}), y^\star(\widehat{x}^r, \widehat{y}^r; \lambda^{r+1})) - \delta \rangle \\
&- \alpha \langle \nabla_x K(x^r, y^{r+1}, \widehat{x}^r, \widehat{y}^{r+1}; \lambda^{r+1}), \varepsilon_x^r \rangle + L_K \alpha^2 \|\varepsilon_x^r\|^2 \\
&- \beta \langle \nabla_y K(x^r, y^r, \widehat{x}^r, \widehat{y}^r; \lambda^{r+1}), \varepsilon_y^r \rangle + L_K \beta^2 \|\varepsilon_y^r\|^2 \\
&- p\langle \widehat{y}^{r+1} - \widehat{y}^r, \widehat{y}^{r+1} + \widehat{y}^r - 2y^\star(\widehat{x}^r, \widehat{y}^{r+1}; \lambda^{r+1}) \rangle - 2p\langle \widehat{y}^{r+1} - \widehat{y}^r, \bar{y}^\star(\widehat{x}^r, \widehat{y}^r) - \widehat{y}^r \rangle \\
&- p\langle \widehat{x}^{r+1} - \widehat{x}^r, \widehat{x}^{r+1} + \widehat{x}^r - 2x^\star(\widehat{x}^{r+1}, \widehat{y}^{r+1}; \lambda^{r+1}) \rangle - 2p\langle \widehat{x}^{r+1} - \widehat{x}^r, \bar{x}^\star(\widehat{x}^r, \widehat{y}^{r+1}) - \widehat{x}^r \rangle
\end{aligned} \tag{95}
$$

where we use the fact that $p/(p-L) > 1$ so that there is a factor of $2p$ in front of terms $\|\widehat{x}^{r+1} - \widehat{x}^r\|^2$ and $\|\widehat{y}^{r+1} - \widehat{y}^r\|^2$.

First, we can get an upper bound for the term in the penultimate line of (95) as follows:

$$
\begin{aligned}
&- p\left\langle \widehat{y}^{r+1} - \widehat{y}^r, \widehat{y}^{r+1} - \widehat{y}^r - 2\left(y^\star(\widehat{x}^r, \widehat{y}^{r+1}; \lambda^{r+1}) - \bar{y}^\star(\widehat{x}^r, \widehat{y}^r)\right)\right\rangle \\
=&- p\left\langle \widehat{y}^{r+1} - \widehat{y}^r, \widehat{y}^{r+1} - \widehat{y}^r - 2\left(y^\star(\widehat{x}^r, \widehat{y}^{r+1}; \lambda^{r+1}) - y^\star(\widehat{x}^r, \widehat{y}^r; \lambda^{r+1})\right)\right\rangle \\
&- p\left\langle \widehat{y}^{r+1} - \widehat{y}^r, y^\star(\widehat{x}^r, \widehat{y}^r; \lambda^{r+1}) - \bar{y}^\star(\widehat{x}^r, \widehat{y}^r)\right\rangle \\
=&- p\|\widehat{y}^{r+1} - \widehat{y}^r\|^2 + 2p\left\langle \widehat{y}^{r+1} - \widehat{y}^r, y^\star(\widehat{x}^r, \widehat{y}^{r+1}; \lambda^{r+1}) - y^\star(\widehat{x}^r, \widehat{y}^r; \lambda^{r+1})\right\rangle \\
&- p\left\langle \widehat{y}^{r+1} - \widehat{y}^r, y^\star(\widehat{x}^r, \widehat{y}^r; \lambda^{r+1}) - \bar{y}^\star(\widehat{x}^r, \widehat{y}^r)\right\rangle.
\end{aligned} \tag{96} \tag{97}
$$

For the last term of the above inequality, we can further have

$$
\begin{aligned}
&p\left\langle \widehat{y}^{r+1} - \widehat{y}^r, y^\star(\widehat{x}^r, \widehat{y}^r; \lambda^{r+1}) - \bar{y}^\star(\widehat{x}^r, \widehat{y}^r)\right\rangle \\
\leq& \frac{p\|\widehat{y}^{r+1} - \widehat{y}^r\|^2}{2\zeta} + \frac{p\zeta}{2}\|y^\star(\widehat{x}^r, \widehat{y}^r; \lambda^{r+1}) - \bar{y}^\star(\widehat{x}^r, \widehat{y}^r)\|^2.
\end{aligned} \tag{98}
$$

For the second term of (97), we can get

$$
\begin{aligned}
&\left\langle \widehat{y}^{r+1} - \widehat{y}^r, y^\star(\widehat{x}^r, \widehat{y}^{r+1}; \lambda^{r+1}) - y^\star(\widehat{x}^r, \widehat{y}^r; \lambda^{r+1})\right\rangle \\
\overset{(a)}{\leq}& \|\widehat{y}^{r+1} - \widehat{y}^r\| \|y^\star(\widehat{x}^r, \widehat{y}^{r+1}; \lambda^{r+1}) - y^\star(\widehat{x}^r, \widehat{y}^r; \lambda^{r+1})\| \\
\overset{(b)}{\leq}& \frac{p}{p-L}\|\widehat{y}^{r+1} - \widehat{y}^r\|^2
\end{aligned} \tag{99} \tag{100}
$$

where $(a)$ is true by applying the Cauchy-Schwarz inequality, in $(b)$ we use the primal error bound (24e).

Similarly, we can obtain an upper bound for the term in the last line of (95) as

$$
\begin{aligned}
&- p\left\langle \widehat{x}^{r+1} - \widehat{x}^r, \widehat{x}^{r+1} - \widehat{x}^r - 2\left(x^\star(\widehat{x}^{r+1}, \widehat{y}^{r+1}; \lambda^{r+1}) - \bar{x}^\star(\widehat{x}^r, \widehat{y}^{r+1})\right)\right\rangle \\
=&- p\left\langle \widehat{x}^{r+1} - \widehat{x}^r, \widehat{x}^{r+1} - \widehat{x}^r - 2\left(x^\star(\widehat{x}^{r+1}, \widehat{y}^{r+1}; \lambda^{r+1}) - x^\star(\widehat{x}^r, \widehat{y}^{r+1}; \lambda^{r+1})\right)\right\rangle \\
&- p\left\langle \widehat{x}^{r+1} - \widehat{x}^r, x^\star(\widehat{x}^r, \widehat{y}^{r+1}; \lambda^{r+1}) - \bar{x}^\star(\widehat{x}^r, \widehat{y}^{r+1})\right\rangle \\
=&- p\|\widehat{x}^{r+1} - \widehat{x}^r\|^2 + 2p\left\langle \widehat{x}^{r+1} - \widehat{x}^r, x^\star(\widehat{x}^{r+1}, \widehat{y}^{r+1}; \lambda^{r+1}) - x^\star(\widehat{x}^r, \widehat{y}^{r+1}; \lambda^{r+1})\right\rangle \\
&- p\left\langle \widehat{x}^{r+1} - \widehat{x}^r, x^\star(\widehat{x}^r, \widehat{y}^{r+1}; \lambda^{r+1}) - \bar{x}^\star(\widehat{x}^r, \widehat{y}^{r+1})\right\rangle.
\end{aligned} \tag{101} \tag{102}
$$

For the last term of the above inequality, we can further have

$$p\left\langle \widehat{x}^{r+1} - \widehat{x}^r, x^\star(\widehat{x}^r, \widehat{y}^{r+1}; \lambda^{r+1}) - \bar{x}^\star(\widehat{x}^r, \widehat{y}^{r+1})\right\rangle$$
$$\leq \frac{p\|\widehat{x}^{r+1} - \widehat{x}^r\|^2}{2\zeta} + \frac{p\zeta}{2}\|x^\star(\widehat{x}^r, \widehat{y}^{r+1}; \lambda^r) - \bar{x}^\star(\widehat{x}^r, \widehat{y}^{r+1})\|^2. \tag{103}$$

Similar as (100), we also have

$$\left\langle \widehat{x}^{r+1} - \widehat{x}^r, x^\star(\widehat{x}^{r+1}, \widehat{y}^{r+1}; \lambda^{r+1}) - y^\star(\widehat{x}^r, \widehat{y}^{r+1}; \lambda^{r+1})\right\rangle$$
$$\overset{(a)}{\leq} \|\widehat{x}^{r+1} - \widehat{x}^r\|\|x^\star(\widehat{x}^{r+1}, \widehat{y}^{r+1}; \lambda^{r+1}) - x^\star(\widehat{x}^r, \widehat{y}^{r+1}; \lambda^{r+1})\| \tag{104}$$
$$\overset{(b)}{\leq} \frac{p}{p-L}\|\widehat{x}^{r+1} - \widehat{x}^r\|^2 \tag{105}$$

where $(a)$ is true by applying the Cauchy-Schwarz inequality, in $(b)$ we use the primal error bound (24b).

Substituting (98), (100), (103), (105) into (95) yields

$$E^{r+1} - E^r$$
$$\leq -\frac{1}{2\alpha}\left\|\mathbb{E}\left[x^{r+1} - x^r\right]\right\|^2 - \frac{1}{2\beta}\left\|\mathbb{E}\left[y^{r+1} - y^r\right]\right\|^2$$
$$- p\left(\frac{1}{2\omega} - \left(\frac{1}{\zeta} + \frac{4p}{p-L}\right)\right)\|\widehat{x}^{r+1} - \widehat{x}^r\|^2 - p\left(\frac{1}{2\omega} - \left(\frac{1}{\zeta} + \frac{4p}{p-L}\right)\right)\|\widehat{y}^{r+1} - \widehat{y}^r\|^2$$
$$- \alpha\left\langle \nabla_x K(x^r, y^{r+1}, \widehat{x}^r, \widehat{y}^{r+1}; \lambda^{r+1}), \varepsilon_x^r\right\rangle + L_K\alpha^2\|\varepsilon_x^r\|^2$$
$$- \beta\langle \nabla_y K(x^r, y^r, \widehat{x}^r, \widehat{y}^r; \lambda^{r+1}), \varepsilon_y^r\rangle + L_K\beta^2\|\varepsilon_y^r\|^2$$
$$+ \frac{\beta}{\gamma^2}\|z^r - z^\star(x^r, y^r)\|^2 + \alpha(\lambda^{r+1}L_g)^2\|z^r - z^\star(x^r, y^{r+1})\|^2$$
$$+ p\zeta\left(\|x^\star(\widehat{x}^r, \widehat{y}^{r+1}; \lambda^{r+1}) - \bar{x}^\star(\widehat{x}^r, \widehat{y}^{r+1})\|^2 + \|y^\star(\widehat{x}^r, \widehat{y}^r; \lambda^{r+1}) - \bar{y}^\star(\widehat{x}^r, \widehat{y}^r)\|^2\right)$$
$$- 2\left\langle \lambda^{r+1} - \lambda^r, g(x^\star(\widehat{x}^r, \widehat{y}^r; \lambda^{r+1}), y^\star(\widehat{x}^r, \widehat{y}^r; \lambda^{r+1})) - g_\gamma^\star(x^\star(\widehat{x}^r, \widehat{y}^r; \lambda^{r+1}), y^\star(\widehat{x}^r, \widehat{y}^r; \lambda^{r+1})) - \delta\right\rangle$$
$$+ \left\langle g(x^r, y^r) - g_\gamma^\star(x^r, y^r) - \delta, \lambda^{r+1} - \lambda^r\right\rangle. \tag{106}$$

Second, we will give an upper bound of the last two terms of (106) as follows.

*Step 1.)*

For any given $h$, the Moreau envelope of the dual update can be written as

$$M(\lambda, h) = \min_{\lambda' \geq 0}\langle -h, \lambda' - \lambda\rangle + \frac{1}{2\tau}\|\lambda' - \lambda\|^2, \tag{107}$$

which directly yields

$$\lambda_+ = \text{Proj}_{\geq 0}(\lambda + \tau h). \tag{108}$$

It is obvious that the quadratic function $M(\lambda, h)$ is smooth. Let $L_M$ denote the gradient Lipschitz parameter. From the optimality condition, we can have

$$\left\langle -h + \frac{1}{\tau}(\lambda_+^r - \lambda^r), \lambda^r - \lambda_+^r\right\rangle \geq 0. \tag{109}$$

Let $\theta = c\mu$ and $w^r \triangleq \widehat{g}(x^r, y^r) - \widehat{g}(x^r, z^r) - \frac{1}{2\gamma}\|z^r - y^r\|^2 - \delta$.

$$M(\lambda^{r+1}, h^{r+1}) - M(\lambda^r, h^r)$$

$$\overset{(a)}{\geq} \langle h^{r+1} - \frac{1}{\tau}(\lambda_+^r - \lambda^r), \lambda^{r+1} - \lambda^r \rangle + \langle -(\lambda_+^r - \lambda^r), h^{r+1} - h^r \rangle$$

$$+ \left(\frac{1}{2\gamma} - \frac{L_M}{2}\right)\left(\|\lambda^{r+1} - \lambda^r\|^2 + \|h^{r+1} - h^r\|^2\right) \tag{110}$$

$$\overset{(b)}{\geq} \mu\langle h^{r+1}, \lambda_+^r - \lambda^r \rangle - \frac{\mu}{\tau}\|\lambda_+^r - \lambda^r\|^2 - \theta\langle \lambda_+^r - \lambda^r, w^r \rangle + \theta\langle \lambda_+^r - \lambda^r, h^r \rangle$$

$$+ \left(\frac{1}{2\gamma} - \frac{L_M}{2}\right)\left(\|\lambda^{r+1} - \lambda^r\|^2 + \|h^{r+1} - h^r\|^2\right) \tag{111}$$

$$\overset{(c)}{\geq} \frac{\theta}{2\tau}\|\lambda_+^r - \lambda^r\|^2 - \theta\langle \lambda_+^r - \lambda^r, w^r \rangle + \left(\frac{1}{2\gamma} - \frac{L_M + \theta}{2}\right)\left(\|\lambda^{r+1} - \lambda^r\|^2 + \|h^{r+1} - h^r\|^2\right) \tag{112}$$

where $(a)$ is true due to the strong convexity when $\tau$ is small, $(b)$ update rule of $\lambda$ and $h^{r+1}$, in $(c)$ we apply the optimality condition (109), i.e, $\langle h, \lambda_+^r - \lambda^r \rangle \geq \tau^{-1}\|\lambda_+^r - \lambda^r\|$ and $\langle \lambda_+^r - \lambda^r, h^r - h^{r+1} \rangle \leq \|\lambda_+^r - \lambda^r\|^2/2 + \|h^r - h^{r+1}\|^2/2$. Therefore, we can obtain

$$M(\lambda^r, h^r) - M(\lambda^{r+1}, h^{r+1})$$

$$\leq -\frac{\theta}{2\tau}\|\lambda_+^r - \lambda^r\|^2 + \theta\left\langle \widehat{g}(x^r, y^r) - \widehat{g}(x^r, z^r) - \frac{1}{2\gamma}\|z^r - y^r\|^2 - \delta, \lambda_+^r - \lambda^r \right\rangle$$

$$+ \left(\frac{L_M + \theta}{2} - \frac{1}{2\gamma}\right)\left(\|\lambda^{r+1} - \lambda^r\|^2 + \|h^{r+1} - h^r\|^2\right). \tag{113}$$

Divide $c$ on both sides gives

$$\frac{1}{c}\left(M(\lambda^r, h^r) - M(\lambda^{r+1}, h^{r+1})\right)$$

$$\leq -\frac{\mu}{2\tau}\|\lambda_+^r - \lambda^r\|^2 + \mu\left\langle \widehat{g}(x^r, y^r) - \widehat{g}(x^r, z^r) - \frac{1}{2\gamma}\|z^r - y^r\|^2 - \delta, \lambda_+^r - \lambda^r \right\rangle$$

$$+ \left(\frac{L_M + \theta}{2c} - \frac{1}{2c\gamma}\right)\left(\|\lambda^{r+1} - \lambda^r\|^2 + \|h^{r+1} - h^r\|^2\right). \tag{114}$$

Note that term $\langle g(x^r, y^r) - g_\gamma^\star(x^r, y^r) - \delta, \lambda^{r+1} - \lambda^r \rangle$ can be decomposed as follows.

$$\langle g(x^r, y^r) - g_\gamma^\star(x^r, y^r) - \delta, \lambda^{r+1} - \lambda^r \rangle$$

$$= \langle g(x^r, y^r) - g_\gamma^\star(x^r, y^r) - \delta, \lambda^{r+1} - \lambda^r \rangle$$

$$+ \left\langle g(x^r, y^r) - g(x^r, z^r) - \frac{1}{2\gamma}\|z^r - y^r\|^2 - \delta, \lambda^{r+1} - \lambda^r \right\rangle - \left\langle g(x^r, y^r) - g(x^r, z^r) - \frac{1}{2\gamma}\|z^r - y^r\|^2 - \delta, \lambda^{r+1} - \lambda^r \right\rangle$$

$$+ \left\langle \widehat{g}(x^r, y^r) - \widehat{g}(x^r, z^r) - \frac{1}{2\gamma}\|z^r - y^r\|^2 - \delta, \lambda^{r+1} - \lambda^r \right\rangle - \left\langle \widehat{g}(x^r, y^r) - \widehat{g}(x^r, z^r) - \frac{1}{2\gamma}\|z^r - y^r\|^2 - \delta, \lambda^{r+1} - \lambda^r \right\rangle$$

$$\leq 2\left\langle g(x^r, y^r) - g_\gamma^\star(x^r, y^r) - \delta, \lambda^{r+1} - \lambda^r \right\rangle$$

$$+ \left\langle g_\gamma^\star(x^r, y^r) - g(x^r, z^r) - \frac{1}{2\gamma}\|z^r - y^r\|^2, \lambda^{r+1} - \lambda^r \right\rangle$$

$$+ \langle \varepsilon_{\widehat{g}_y}^r - \varepsilon_{\widehat{g}_z}^r, \lambda^{r+1} - \lambda^r \rangle$$

$$- \left\langle \widehat{g}(x^r, y^r) - \widehat{g}(x^r, z^r) - \frac{1}{2\gamma}\|z^r - y^r\|^2 - \delta, \lambda^{r+1} - \lambda^r \right\rangle. \tag{115}$$

Subsequently, we can derive an upper bound for the sum of the last two terms in (106) as follows.

$$
2\langle g(x^r, y^r) - g_\gamma^\star(x^r, y^r), \lambda^{r+1} - \lambda^r \rangle
$$

$$
- 2\Big\langle g(x^\star(\widehat{x}^r, \widehat{y}^r; \lambda^{r+1}), y^\star(\widehat{x}^r, \widehat{y}^r; \lambda^{r+1})) - g_\gamma^\star(x^\star(\widehat{x}^r, \widehat{y}^r; \lambda^{r+1}), y^\star(\widehat{x}^r, \widehat{y}^r; \lambda^{r+1})), \lambda^{r+1} - \lambda^r \Big\rangle
$$

$$
+ \Big\langle g_\gamma^\star(x^r, y^r) - g(x^r, z^r) - \frac{1}{2\gamma}\|z^r - y^r\|^2, \lambda^{r+1} - \lambda^r \Big\rangle + \langle \varepsilon_{\widehat{g}_y}^r - \varepsilon_{\widehat{g}_z}^r, \lambda^{r+1} - \lambda^r \rangle
$$

$$
- \Big\langle \widehat{g}(x^r, y^r) - \widehat{g}(x^r, z^r) - \frac{1}{2\gamma}\|z^r - y^r\|^2 - \delta, \lambda^{r+1} - \lambda^r \Big\rangle
$$

$$
\overset{(a)}{\leq} 8\mu\Big( \Big\| g(x^r, y^r) - g_\gamma^\star(x^r, y^r)
$$

$$
- \big(g\left(x^\star(\widehat{x}^r, \widehat{y}^r; \lambda^{r+1}), y^\star(\widehat{x}^r, \widehat{y}^r; \lambda^{r+1})\right) - g_\gamma^\star\left(x^\star(\widehat{x}^r, \widehat{y}^r; \lambda^{r+1}), y^\star(\widehat{x}^r, \widehat{y}^r; \lambda^{r+1})\right)\big) \Big\|^2 \Big)
$$

$$
+ 8\mu\ell_\gamma^2\|z^r - z^\star(x^r, y^r)\|^2 + \frac{1}{2\mu}\|\lambda^{r+1} - \lambda^r\|^2
$$

$$
+ \langle \varepsilon_{\widehat{g}_y}^r - \varepsilon_{\widehat{g}_z}^r, \lambda^{r+1} - \lambda^r \rangle - \Big\langle \widehat{g}(x^r, y^r) - \widehat{g}(x^r, z^r) - \frac{1}{2\gamma}\|z^r - y^r\|^2 - \delta, \lambda^{r+1} - \lambda^r \Big\rangle
$$

$$
\overset{(b)}{\leq} 16\mu(\ell_g^2 + L_\gamma^2)\left(\|x^\star(\widehat{x}^r, \widehat{y}^r; \lambda^{r+1}) - x^r\|^2 + \|y^\star(\widehat{x}^r, \widehat{y}^r; \lambda^{r+1}) - y^r\|^2\right)
$$

$$
+ 8\mu\ell_\gamma^2\|z^r - z^\star(x^r, y^r)\|^2 + \frac{1}{2\mu}\|\lambda^{r+1} - \lambda^r\|^2
$$

$$
+ \langle \varepsilon_{\widehat{g}_y}^r - \varepsilon_{\widehat{g}_z}^r, \lambda^{r+1} - \lambda^r \rangle - \Big\langle \widehat{g}(x^r, y^r) - \widehat{g}(x^r, z^r) - \frac{1}{2\gamma}\|z^r - y^r\|^2 - \delta, \lambda^{r+1} - \lambda^r \Big\rangle
$$

$$
\overset{(c)}{\leq} 16\mu(\ell_g^2 + L_\gamma^2)\sigma_3^2\left(\frac{1}{\alpha^2}\|x^{r+1} - x^r\|^2 + \frac{1}{\beta^2}\|y^{r+1} - y^r\|^2\right) + 8\mu\ell_\gamma^2\|z^r - z^\star(x^r, y^r)\|^2 + \frac{1}{2\mu}\|\lambda^{r+1} - \lambda^r\|^2
$$

$$
+ \langle \varepsilon_{\widehat{g}_y}^r - \varepsilon_{\widehat{g}_z}^r, \lambda^{r+1} - \lambda^r \rangle - \Big\langle \widehat{g}(x^r, y^r) - \widehat{g}(x^r, z^r) - \frac{1}{2\gamma}\|z^r - y^r\|^2 - \delta, \lambda^{r+1} - \lambda^r \Big\rangle
$$

where in $(a)$ we use the Cauchy-Schwarz inequality, $(b)$ is true due to the Lipschitz continuity, i.e.,

$$
\begin{aligned}
&|g(x^r, y^r) - g(x^\star(\widehat{x}^r, \widehat{y}^r; \lambda^{r+1}), y^\star(\widehat{x}^r, \widehat{y}^r; \lambda^{r+1}))|^2 \\
&\leq 2\ell_g^2\left(\|x^r - x^\star(\widehat{x}^r, \widehat{y}^r; \lambda^{r+1})\|^2 + \|y^r - y^\star(\widehat{x}^r, \widehat{y}^r; \lambda^{r+1})\|^2\right),
\end{aligned}
\tag{116}
$$

and

$$
\begin{aligned}
&|g_\gamma^\star(x^r, y^r) - g_\gamma^\star(x^\star(\widehat{x}^r, \widehat{y}^r; \lambda^{r+1}), y^\star(\widehat{x}^r, \widehat{y}^r; \lambda^{r+1}))|^2 \\
&\leq 2\ell_\gamma^2\left(\|x^r - x^\star(\widehat{x}^r, \widehat{y}^r; \lambda^{r+1})\|^2 + \|y^r - y^\star(\widehat{x}^r, \widehat{y}^r; \lambda^{r+1})\|^2\right),
\end{aligned}
\tag{117}
$$

in $(c)$ we apply the primal error bounds (27) and (28b).

Let

$$
F^r = K(x^r, y^r, z^r, \widehat{x}^r, \widehat{y}^r; \lambda^r) - 2D(\widehat{x}^r, \widehat{y}^r; \lambda^r) + 2P(\widehat{x}^r, \widehat{y}^r) - \frac{1}{c}M(\lambda^r, h^r).
\tag{118}
$$

*Step 2.)* Substituting and back to (106) gives

$$
\begin{aligned}
&F^{r+1} - F^r \\
&\leq -\left(\frac{1}{2\alpha} - 16\mu(\ell_g^2 + L_\gamma^2)\frac{\sigma_3^2}{\alpha^2}\right)\|\mathbb{E}\left[x^{r+1} - x^r\right]\|^2 - \left(\frac{1}{2\beta} - 16\mu(\ell_g^2 + L_\gamma^2)\frac{\sigma_3^2}{\beta^2}\right)\|\mathbb{E}\left[y^{r+1} - y^r\right]\|^2 \\
&\quad - p\left(\frac{1}{2\omega} - \left(\frac{1}{\zeta} + \frac{4p}{p-L}\right)\right)\|\widehat{x}^{r+1} - \widehat{x}^r\|^2 - p\left(\frac{1}{2\omega} - \left(\frac{1}{\zeta} + \frac{4p}{p-L}\right)\right)\|\widehat{y}^{r+1} - \widehat{y}^r\|^2 \\
&\quad + \left(\frac{\beta}{\gamma^2} + 8\mu\ell_\gamma^2\right)\|z^r - z^\star(x^r, y^r)\|^2 + \alpha(\lambda^{r+1}L_g)^2\|z^r - z^\star(x^r, y^{r+1})\|^2 \\
&\quad + p\zeta\left(\|x^\star(\widehat{x}^r, \widehat{y}^{r+1}; \lambda^{r+1}) - \bar{x}^\star(\widehat{x}^r, \widehat{y}^{r+1})\|^2 + \|y^\star(\widehat{x}^r, \widehat{y}^r; \lambda^{r+1}) - \bar{y}^\star(\widehat{x}^r, \widehat{y}^r)\|^2\right) \\
&\quad - \alpha\left\langle\nabla_x K(x^r, y^{r+1}, \widehat{x}^r, \widehat{y}^{r+1}; \lambda^{r+1}), \varepsilon_x^r\right\rangle + L_K\alpha^2\|\varepsilon_x^r\|^2 - \beta\langle\nabla_y K(x^r, y^r, \widehat{x}^r, \widehat{y}^r; \lambda^{r+1}), \varepsilon_y^r\rangle + L_K\beta^2\|\varepsilon_y^r\|^2 \\
&\quad - \frac{\mu}{2\tau}\|\lambda_+^r - \lambda^r\|^2 + \left(\frac{L_M + \theta}{2c} - \frac{1}{2c\gamma} + \frac{1}{2\mu}\right)\|\lambda^{r+1} - \lambda^r\|^2 \\
&\quad + \left(\frac{L_M + \theta}{2c} - \frac{1}{2c\gamma}\right)\|h^{r+1} - h^r\|^2 + \langle\varepsilon_{\widehat{g}_y}^r - \varepsilon_{\widehat{g}_z}^r, \lambda^{r+1} - \lambda^r\rangle.
\end{aligned}
\tag{119}
$$

When constant $\gamma$ is small, i.e., $\frac{1}{2\gamma} \geq \frac{L_M}{2}$, we have

$$
-\frac{\mu}{4\tau}\|\lambda_+^r - \lambda^r\|^2 + \left(\frac{L_M + \theta}{2c} - \frac{1}{2c\gamma} + \frac{1}{2\mu}\right)\|\lambda^{r+1} - \lambda^r\|^2 \leq -\frac{\mu}{4\tau}\|\lambda_+^r - \lambda^r\|^2
\tag{120}
$$

where we require $\tau < 1/2$.

*Step 3.)* Applying the reverse triangle inequality, we can get

$$
\|\lambda^{r+1} - \lambda^r\|^2 = \|\lambda^{r+1} - \lambda_+^r(\widehat{x}^r, \widehat{y}^r) + \lambda_+^r(\widehat{x}^r, \widehat{y}^r) - \lambda^r\|^2 \geq \frac{\|\lambda_+^r(\widehat{x}^r, \widehat{y}^r) - \lambda^r\|^2}{2} - \|\lambda^{r+1} - \lambda_+^r(\widehat{x}^r, \widehat{y}^r)\|^2.
\tag{121}
$$

Combining (121), (123) gives

$$
\begin{aligned}
&\|\lambda^{r+1} - \lambda^r\|^2 \\
&\geq \frac{\|\lambda_+^r(\widehat{x}^r, \widehat{y}^r) - \lambda^r\|^2}{2} - \tau^2\|h^r - \nabla_\lambda K(x^\star(v^r; \lambda^r), y^\star(v^r; \lambda^r), v^r; \lambda^r)\|^2.
\end{aligned}
$$

Applying the primal error bounds (27) and (28b), we can obtain

$$
\begin{aligned}
-\frac{\mu}{2\tau}\|\lambda_+^r - \lambda^r\|^2 &= -\frac{1}{2\mu\tau}\|\lambda^{r+1} - \lambda^r\|^2 \\
&\leq -\frac{\|\lambda_+^r(\widehat{x}^r, \widehat{y}^r) - \lambda^r\|^2}{4\mu\tau} + \frac{\tau}{2\mu}\|h^r - \nabla_\lambda K(x^\star(v^r; \lambda^r), y^\star(v^r; \lambda^r), v^r; \lambda^r)\|^2.
\end{aligned}
\tag{122}
$$

According to the definition of $\lambda_+^r(\widehat{x}^r, \widehat{y}^r)$ (cf. (234a)), we have

$$
\begin{aligned}
\|\lambda^{r+1} - \lambda_+^r(\widehat{x}^r, \widehat{y}^r)\|^2 &\overset{(a)}{\leq} \left\|\lambda^r + \tau h^{r+1} - [\lambda^r + \tau\nabla_\lambda K(x^\star(v^r; \lambda^r), y^\star(v^r; \lambda^r), v^r; \lambda^r)]\right\|^2 \\
&\leq \tau^2\|h^{r+1} - \nabla_\lambda K(x^\star(v^r; \lambda^r), y^\star(v^r; \lambda^r), v^r; \lambda^r)\|^2.
\end{aligned}
\tag{123}
$$

Next, we need to derive the recursion for term $\|h^{r+1} - \nabla_\lambda K(x^\star(v^r; \lambda^r), y^\star(v^r; \lambda^r), v^r; \lambda^{r+1})\|^2$. First, we decompose it

from the noise terms as follows.

$$
\|h^{r+2} - \nabla_\lambda K(x^\star(v^{r+1}; \lambda^{r+1}), y^\star(v^{r+1}; \lambda^{r+1}), v^{r+1}; \lambda^{r+1})\|^2
$$

$$
\leq \left\| (1-\theta)h^{r+1} + \theta \left( \widehat{g}(x^r, y^r) - \widehat{g}(x^r, z^r) - \frac{1}{2\gamma}\|z^r - y^r\|^2 - \delta \right) \right.
$$

$$
\left. - \nabla_\lambda K(x^\star(v^{r+1}; \lambda^{r+1}), y^\star(v^{r+1}; \lambda^{r+1}), v^{r+1}; \lambda^{r+1}) \right\|^2 \tag{124}
$$

$$
= \|h_\theta^r + \theta(g(x^r, y^r) - g(x^r, z^r))\|^2 + \theta(\varepsilon_{\widehat{g}_y}^r - \varepsilon_{\widehat{g}_z}^r)(h_\theta^r + \theta(g(x^r, y^r))) + \theta^2(\varepsilon_{\widehat{g}_y}^r - \varepsilon_{\widehat{g}_z}^r)^2
$$

where

$$
h_\theta^r \triangleq (1-\theta)h^{r+1} + \theta(-(2\gamma)^{-1}\|z^r - y^r\|^2 - \delta) - \nabla_\lambda K(x^\star(v^{r+1}; \lambda^{r+1}), y^\star(v^{r+1}; \lambda^{r+1}), v^{r+1}; \lambda^{r+1}). \tag{125}
$$

Then, using the convexity of $\|\|^2$, we can obtain

$$
\left\| (1-\theta)h^{r+1} + \theta \left( \widehat{g}(x^r, y^r) - \widehat{g}(x^r, z^r) - \frac{1}{2\gamma}\|z^r - y^r\|^2 - \delta \right) \right.
$$

$$
\left. - \nabla_\lambda K(x^\star(v^{r+1}; \lambda^{r+1}), y^\star(v^{r+1}; \lambda^{r+1}), v^{r+1}; \lambda^{r+1}) \right\|^2
$$

$$
\leq (1-\theta)\|h^{r+1} - \nabla_\lambda K(x^\star(v^r; \lambda^r), y^\star(v^r; \lambda^r), v^r; \lambda^r)\|^2
$$

$$
+ 2\theta\|\nabla_\lambda K(x^\star(v^{r+1}; \lambda^{r+1}), y^\star(v^{r+1}; \lambda^{r+1}), v^{r+1}; \lambda^{r+1}) - \nabla_\lambda K(x^\star(v^r; \lambda^r), y^\star(v^r; \lambda^r), v^r; \lambda^r)\|^2
$$

$$
+ 2\theta\left\| g(x^r, y^r) - g(x^r, z^r) - \frac{1}{2\gamma}\|z^r - y^r\|^2 - \delta - \nabla_\lambda K(x^\star(v^r; \lambda^r), y^\star(v^r; \lambda^r), v^r; \lambda^r) \right\|^2
$$

$$
\overset{(a)}{\leq} (1-\theta)\|h^{r+1} - \nabla_\lambda K(x^\star(v^r; \lambda^r), y^\star(v^r; \lambda^r), v^r; \lambda^r)\|^2
$$

$$
+ 2 \cdot 24\theta(\ell_g^2 + \ell_\gamma^2)\left( \sigma_1^2\|\mathbb{E}x^{r+1} - x^r\|^2 + \sigma_1^2\|\mathbb{E}y^{r+1} - y^r\|^2 + \sigma_2^2\|\mathbb{E}\lambda^{r+1} - \lambda^r\|^2 \right)
$$

$$
+ 2 \cdot 3\theta(\ell_g + \ell_\gamma)^2\left( \frac{\sigma_3^2}{\alpha^2}\|\mathbb{E}x^{r+1} - x^r\|^2 + \frac{\sigma_3^2}{\beta^2}\|\mathbb{E}y^{r+1} - y^r\|^2 + 4\sigma_2^2\|\mathbb{E}\lambda^{r+1} - \lambda^r\|^2 \right) \tag{126}
$$

$$
\leq (1-\theta)\|h^{r+1} - \nabla_\lambda K(x^\star(v^r; \lambda^r), y^\star(v^r; \lambda^r), v^r; \lambda^r)\|^2 + 6(\ell_g^2 + \ell_\gamma^2)\left( 4\sigma_1^2(1-\theta) + \frac{\sigma_3^2\theta}{\alpha^2} \right)\|\mathbb{E}x^{r+1} - x^r\|^2
$$

$$
+ 6(\ell_g^2 + \ell_\gamma^2)\left( 4\sigma_1^2(1-\theta) + \frac{\sigma_3^2\theta}{\beta^2} \right)\|\mathbb{E}y^{r+1} - y^r\|^2 + 6(\ell_g^2 + \ell_\gamma^2)6\sigma_2^2\|\mathbb{E}\lambda^{r+1} - \lambda^r\|^2
$$

where in $(a)$ we apply

$$
\left\| g(x^\star(x^{r+1}, y^{r+1}; \lambda^{r+1}), y^\star(x^{r+1}, y^{r+1}; \lambda^{r+1})) - g(x^\star(x^r, y^r; \lambda^r), y^\star(x^r, y^r; \lambda^r)) \right.
$$

$$
\left. + g_\gamma^\star(x^\star(x^{r+1}, y^{r+1}; \lambda^{r+1}), y^\star(x^{r+1}, y^{r+1}; \lambda^{r+1})) - g_\gamma^\star(x^\star(x^r, y^r; \lambda^r), y^\star(x^r, y^r; \lambda^r)) \right\|^2
$$

$$
\leq 4(\ell_g^2 + \ell_\gamma^2)\left( \|x^\star(x^{r+1}, y^{r+1}; \lambda^{r+1}) - x^\star(x^r, y^r; \lambda^r)\|^2 + \|y^\star(x^{r+1}, y^{r+1}; \lambda^{r+1}) - y^\star(x^r, y^r; \lambda^r)\|^2 \right) \tag{127}
$$

$$
\leq 24(\ell_g^2 + \ell_\gamma^2)\left( \sigma_1^2\|\mathbb{E}x^{r+1} - x^r\|^2 + \sigma_1^2\|\mathbb{E}y^{r+1} - y^r\|^2 + \sigma_2^2\|\mathbb{E}\lambda^{r+1} - \lambda^r\|^2 \right), \tag{128}
$$

and

$$
\left\| g(x^r, y^r) - (g(x^r, z^r) + \frac{1}{2\gamma}\|z^r - y^r\|^2) - g(x^\star(x^r, y^r; \lambda^r), y^\star(x^r, y^r; \lambda^r)) + g_\gamma^\star(x^\star(x^r, y^r; \lambda^r), y^\star(x^r, y^r; \lambda^r)) \right\|
$$

$$
\leq (\ell_g + \ell_\gamma)\left( \|x^r - x^\star(x^r, y^r; \lambda^r)\| + \|y^r - y^\star(x^r, y^r; \lambda^r)\| \right) \tag{129}
$$

$$
\leq (\ell_g + \ell_\gamma)\left( \frac{\sigma_3}{\alpha}\|\mathbb{E}x^{r+1} - x^r\| + \frac{\sigma_3}{\beta}\|\mathbb{E}y^{r+1} - y^r\| + 2\sigma_2\|\mathbb{E}\lambda^{r+1} - \lambda^r\| \right). \tag{130}
$$

Let

$$C_{h_x} = 6\frac{\tau}{2\mu}(\ell_g^2 + \ell_\gamma^2)\left(4\sigma_1^2(1-\theta) + \frac{\sigma_3^2\theta}{\alpha^2}\right), \tag{131a}$$

$$C_{h_y} = 6\frac{\tau}{2\mu}(\ell_g^2 + \ell_\gamma^2)\left(4\sigma_1^2(1-\theta) + \frac{\sigma_3^2\theta}{\beta^2}\right), \tag{131b}$$

$$C_{h_\lambda} = 6\frac{\tau}{2\mu}(\ell_g^2 + \ell_\gamma^2)6\sigma_2^2. \tag{131c}$$

Finally, by observing the dual error bound, we need to further quantify $\|y^\star(\widehat{x}^r, \widehat{y}^r; \lambda^{r+1}) - \bar{y}^\star(\widehat{x}^r, \widehat{y}^r)\|^2 + \|x^\star(\widehat{x}^r, \widehat{y}^{r+1}; \lambda^{r+1}) - \bar{x}^\star(\widehat{x}^r, \widehat{y}^{r+1})\|^2$ as follows. Note that

$$\|y^\star(\widehat{x}^r, \widehat{y}^r; \lambda^{r+1}) - \bar{y}^\star(\widehat{x}^r, \widehat{y}^r)\|^2 + \|x^\star(\widehat{x}^r, \widehat{y}^{r+1}; \lambda^{r+1}) - \bar{x}^\star(\widehat{x}^r, \widehat{y}^{r+1})\|^2$$
$$\leq \|y^\star(\widehat{x}^r, \widehat{y}^r; \lambda^{r+1}) - \bar{y}^\star(\widehat{x}^r, \widehat{y}^r)\|^2 + 3\|x^\star(\widehat{x}^r, \widehat{y}^r; \lambda^{r+1}) - \bar{x}^\star(\widehat{x}^r, \widehat{y}^r)\|^2$$
$$+ 3\|x^\star(\widehat{x}^r, \widehat{y}^{r+1}; \lambda^{r+1}) - x^\star(\widehat{x}^r, \widehat{y}^r; \lambda^{r+1})\|^2 + 3\|\bar{x}^\star(\widehat{x}^r, \widehat{y}^{r+1}) - \bar{x}^\star(\widehat{x}^r, \widehat{y}^r)\|^2 \tag{132}$$
$$\overset{(a)}{\leq} 2\|y^\star(\widehat{x}^r, \widehat{y}^r; \lambda_+^r(\widehat{x}^r, \widehat{y}^r)) - \bar{y}^\star(\widehat{x}^r, \widehat{y}^r)\|^2 + 6\|x^\star(\widehat{x}^r, \widehat{y}^r; \lambda_+^r(\widehat{x}^r, \widehat{y}^r)) - \bar{x}^\star(\widehat{x}^r, \widehat{y}^r)\|^2$$
$$+ 2\|y^\star(\widehat{x}^r, \widehat{y}^r; \lambda^{r+1}) - y^\star(\widehat{x}^r, \widehat{y}^r; \lambda_+^r(\widehat{x}^r, \widehat{y}^r))\|^2$$
$$+ 6\|x^\star(\widehat{x}^r, \widehat{y}^r; \lambda^{r+1}) - x^\star(\widehat{x}^r, \widehat{y}^r; \lambda_+^r(\widehat{x}^r, \widehat{y}^r))\|^2 + 6\sigma_1^2\|\widehat{y}^{r+1} - \widehat{y}^r\|^2 \tag{133}$$
$$\overset{(b)}{\leq} 2\|y^\star(\widehat{x}^r, \widehat{y}^r; \lambda_+^r(\widehat{x}^r, \widehat{y}^r)) - \bar{y}^\star(\widehat{x}^r, \widehat{y}^r)\|^2 + 6\|x^\star(\widehat{x}^r, \widehat{y}^r; \lambda_+^r(\widehat{x}^r, \widehat{y}^r)) - \bar{x}^\star(\widehat{x}^r, \widehat{y}^r)\|^2$$
$$+ 8\sigma_2^2\left\|\lambda_+^r(\widehat{x}^r, \widehat{y}^r) - \lambda^{r+1}\right\|^2 + 6\sigma_1^2\|\widehat{y}^{r+1} - \widehat{y}^r\|^2 \tag{134}$$

where in $(a)$ we use the primal error bounds (24e) and (24c), $(b)$ holds as we first apply the primal error bounds (26) and (25b).

Let

$$G^r = F^r + \frac{\tau}{2\mu}\|h^{r+1} - \nabla_\lambda K(x^\star(v^r; \lambda^r), y^\star(v^r; \lambda^r), v^r; \lambda^r)\|^2. \tag{135}$$

As a result, we can get

$$G^{r+1} - G^r$$
$$\leq -\left(\frac{1}{2\alpha} - C_{h_x} - 16\mu(\ell_g^2 + L_\gamma^2)\frac{\sigma_3^2}{\alpha^2}\right)\left\|\mathbb{E}\left[x^{r+1} - x^r\right]\right\|^2 - \left(\frac{1}{2\beta} - C_{h_y} - 16\mu(\ell_g^2 + L_\gamma^2)\frac{\sigma_3^2}{\beta^2}\right)\left\|\mathbb{E}\left[y^{r+1} - y^r\right]\right\|^2$$
$$- p\left(\frac{1}{2\omega} - \left(\frac{1}{\zeta} + \frac{4p}{p-L}\right)\right)\|\widehat{x}^{r+1} - \widehat{x}^r\|^2 - p\left(\frac{1}{2\omega} - \left(\frac{1}{\zeta} + \frac{4p}{p-L}\right) - 6\zeta\sigma_1^2\right)\|\widehat{y}^{r+1} - \widehat{y}^r\|^2$$
$$+ \left(\frac{\beta}{\gamma^2} + 8\mu\ell_\gamma^2\right)\|z^r - z^\star(x^r, y^r)\|^2 + \alpha(\lambda^{r+1}L_g)^2\|z^r - z^\star(x^r, y^{r+1})\|^2$$
$$+ 6p\zeta\left(\|x^\star(\widehat{x}^r, \widehat{y}^r; \lambda_+^r(\widehat{x}^r, \widehat{y}^r)) - \bar{x}^\star(\widehat{x}^r, \widehat{y}^r)\|^2 + \|y^\star(\widehat{x}^r, \widehat{y}^r; \lambda_+^r(\widehat{x}^r, \widehat{y}^r)) - \bar{y}^\star(\widehat{x}^r, \widehat{y}^r)\|^2\right)$$
$$- (1-\vartheta)\frac{\tau}{2\mu}\|h^{r+1} - \nabla_\lambda K(x^\star(v^r; \lambda^r), y^\star(v^r; \lambda^r), v^r; \lambda^r)\|^2$$
$$- \alpha\left\langle\nabla_x K(x^r, y^{r+1}, \widehat{x}^r, \widehat{y}^{r+1}; \lambda^{r+1}), \varepsilon_x^r\right\rangle + L_K\alpha^2\|\varepsilon_x^r\|^2$$
$$- \beta\langle\nabla_y K(x^r, y^r, \widehat{x}^r, \widehat{y}^r; \lambda^{r+1}), \varepsilon_y^r\rangle + L_K\beta^2\|\varepsilon_y^r\|^2$$
$$- \frac{\mu}{2\tau}\|\lambda_+^r - \lambda^r\|^2 - \frac{1}{4\mu\tau}\|\lambda_+^r(\widehat{x}^r, \widehat{y}^r) - \lambda^r\|^2 + 8\sigma_2^2 p\zeta\left\|\lambda_+^r(\widehat{x}^r, \widehat{y}^r) - \mathbb{E}\lambda^{r+1}\right\|^2 + C_{h_\lambda}\mu^2\|\lambda^r - \mathbb{E}\lambda_+^r\|^2$$
$$+ \langle\varepsilon_{\widehat{g}_y}^r - \varepsilon_{\widehat{g}_z}^r, \lambda^{r+1} - \lambda^r\rangle + \frac{\tau}{\mu}\left(\theta(\varepsilon_{\widehat{g}_y}^r - \varepsilon_{\widehat{g}_z}^r)(h_\theta^r + \theta(g(x^r, y^r)) + \theta^2(\varepsilon_{\widehat{g}_y}^r - \varepsilon_{\widehat{g}_z}^r)^2)\right). \tag{136}$$

Note that
$$\left\|\lambda_+^r(\widehat{x}^r, \widehat{y}^r) - \lambda^{r+1}\right\|^2 = 2\|\lambda_+^r(\widehat{x}^r, \widehat{y}^r) - \lambda^r\|^2 + 2\|\lambda^r - \lambda^{r+1}\|^2, \tag{137}$$

and

$$\|\lambda^r - \lambda^{r+1}\|^2 = \mu^2 \|\lambda^r - \lambda_+^r\|^2. \tag{138}$$

Then, we can get

$$
-\frac{\mu}{2\tau}\|\lambda_+^r - \lambda^r\|^2 - \frac{1}{4\mu\tau}\|\lambda_+^r(\widehat{x}^r, \widehat{y}^r) - \lambda^r\|^2 + 8\sigma_2^2 p\zeta \left\|\lambda_+^r(\widehat{x}^r, \widehat{y}^r) - \mathbb{E}\lambda^{r+1}\right\|^2 + C_{h_\lambda}\|\mathbb{E}\lambda^{r+1} - \lambda^r\|^2
$$

$$
\leq \left(-\frac{1}{8\mu\tau} + 16\sigma_2^2 p\zeta\right)\|\lambda_+^r(\widehat{x}^r, \widehat{y}^r) - \lambda^r\|^2 + \left(-\frac{1}{8\tau\mu} + 16\sigma_2^2 p\zeta + C_{h_\lambda}\right)\|\lambda^{r+1} - \lambda^r\|^2
$$

$$
+ (16\sigma_2^2 p\zeta + C_{h_\lambda})\frac{\tau^2}{1-\theta}\max_r\|\varepsilon_{\widehat{g}_y}^r - \varepsilon_{\widehat{g}_z}^r\|^2 \tag{139}
$$

where we use the fact that

$$
\|\mathbb{E}\lambda^{r+1} - \lambda^{r+1}\|^2 \leq \tau^2 \|\mathbb{E}h^{r+1} - h^{r+1}\| \leq \frac{\tau^2}{1-\theta}\max_r\|\varepsilon_{\widehat{g}_y}^r - \varepsilon_{\widehat{g}_z}^r\|^2. \tag{140}
$$

Further, we have

$$
\|z^r - z^\star(x^r, y^{r+1})\|^2 \leq 2\left(\|z^r - z^\star(x^r, y^r)\|^2 + \|z^\star(x^r, y^r) - z^\star(x^r, y^{r+1})\|^2\right) \tag{141}
$$

$$
\leq 2\left(\|z^r - z^\star(x^r, y^r)\|^2 + L_z^2\|\mathbb{E}y^{r+1} - y^r\|^2\right). \tag{142}
$$

Let

$$
C_z \triangleq \frac{\beta}{\gamma^2} + 8\mu\ell_\gamma^2 + 2\alpha(\lambda^{r+1}L_g)^2. \tag{143}
$$

Let define the final potential function as

$$
Q^r = G^r + C_z\|z^r - z^\star(x^r, y^r)\|^2 - \underline{f}. \tag{144}
$$

Substituting (220), (143) to (136) gives the desired result.

$$
Q^{r+1} - Q^r
$$

$$
\leq -\left(\frac{1}{2\alpha} - C_{h_x} - 16\mu(\ell_g^2 + L_\gamma^2)\frac{\sigma_3^2}{\alpha^2} - C_z\left(2L_z^2 + \frac{\eta L_z}{2}\right)\right)\left\|\mathbb{E}\left[x^{r+1} - x^r\right]\right\|^2 - \frac{1}{8\mu\tau}\|\lambda_+^r(\widehat{x}^r, \widehat{y}^r) - \lambda^r\|^2
$$

$$
- \left(\frac{1}{2\beta} - C_{h_y} - 16\mu(\ell_g^2 + L_\gamma^2)\frac{\sigma_3^2}{\beta^2} - 2\alpha(\lambda^{r+1}L_g)^2 L_z^2 - C_z\left(2L_z^2 + \frac{\eta L_z}{2}\right)\right)\left\|\mathbb{E}\left[y^{r+1} - y^r\right]\right\|^2
$$

$$
- p\left(\frac{1}{2\omega} - \left(\frac{1}{\zeta} + \frac{4p}{p-L}\right)\right)\|\widehat{x}^{r+1} - \widehat{x}^r\|^2 - p\left(\frac{1}{2\omega} - \left(\frac{1}{\zeta} + \frac{4p}{p-L}\right) - 6\zeta\sigma_1^2\right)\|\widehat{y}^{r+1} - \widehat{y}^r\|^2
$$

$$
+ 6p\zeta\left(\|x^\star(\widehat{x}^r, \widehat{y}^r; \lambda_+^r(\widehat{x}^r, \widehat{y}^r)) - \bar{x}^\star(\widehat{x}^r, \widehat{y}^r)\|^2 + \|y^\star(\widehat{x}^r, \widehat{y}^r; \lambda_+^r(\widehat{x}^r, \widehat{y}^r)) - \bar{y}^\star(\widehat{x}^r, \widehat{y}^r)\|^2\right)
$$

$$
- (1-\vartheta)\frac{\tau}{2\mu}\|h^{r+1} - \nabla_\lambda K(x^\star(v^r; \lambda^r), y^\star(v^r; \lambda^r), v^r; \lambda^r)\|^2 - (1-\varphi)C_z\|z^r - z^\star(x^r, y^r)\|^2
$$

$$
- \alpha\left\langle\nabla_x K(x^r, y^{r+1}, \widehat{x}^r, \widehat{y}^{r+1}; \lambda^{r+1}), \varepsilon_x^r\right\rangle + L_K\alpha^2\|\varepsilon_x^r\|^2 - \beta\left\langle\nabla_y K(x^r, y^r, \widehat{x}^r, \widehat{y}^r; \lambda^{r+1}), \varepsilon_y^r\right\rangle + L_K\beta^2\|\varepsilon_y^r\|^2
$$

$$
+ (16\sigma_2^2 p\zeta + C_{h_\lambda})\frac{\tau^2}{1-\theta}\max_r\|\varepsilon_{\widehat{g}_y}^r - \varepsilon_{\widehat{g}_z}^r\|^2 + C_z n_{z_2}^r(\eta, \varepsilon_x^r, \varepsilon_y^r, \varepsilon_{g_z}^r)
$$

$$
+ \langle\varepsilon_{\widehat{g}_y}^r - \varepsilon_{\widehat{g}_z}^r, \lambda^{r+1} - \lambda^r\rangle + \frac{\tau}{2\mu}\left(\theta(\varepsilon_{\widehat{g}_y}^r - \varepsilon_{\widehat{g}_z}^r)(h_\theta^r + \theta(g(x^r, y^r)) + \theta^2(\varepsilon_{\widehat{g}_y}^r - \varepsilon_{\widehat{g}_z}^r)^2\right) \tag{145}
$$

where the terms in the last three lines are noise terms and defined as $n_Q^r \triangleq n_Q^r(\alpha, \beta, \tau, \eta, \theta, \varepsilon_{\widehat{g}_y}^r, \varepsilon_{\widehat{g}_z}^r, \varepsilon_x^r, \varepsilon_y^r, \varepsilon_{g_z}^r)$.

$\square$

## B.3. Descent of Potential Function

**Lemma 1** (Formal). *Assume that A1-A5 are satisfied. Suppose that the sequence $\{x^r, y^r, z^r, \widehat{x}^r, \widehat{y}^r, \lambda^r, \forall r\}$ is generated by TSP, $p > L$, and $\lambda^r \leq \Lambda$, $x^r, y^r, \bar{y}^\star(\widehat{x}^r, \widehat{y}^r), y^\star(\widehat{x}^r, \widehat{y}^r; \lambda^{r+1})$ are bounded. When the step-sizes are chosen such that* (149), (150), (151), *hold, then, we have either*

$$
\begin{aligned}
&\mathcal{Q}^{r+1} - \mathcal{Q}^r \\
&\leq -\frac{1}{8\alpha}\left\|\mathbb{E}\left[x^{r+1} - x^r\right]\right\|^2 - \frac{1}{8\beta}\left\|\mathbb{E}\left[y^{r+1} - y^r\right]\right\|^2 - \frac{p}{8\omega}\|\widehat{x}^{r+1} - \widehat{x}^r\|^2 - \frac{p}{8\omega}\|\widehat{y}^{r+1} - \widehat{y}^r\|^2 \\
&\quad - \frac{(1-\varphi)C_z}{8}\|z^r - z^\star(x^r, y^r)\|^2 - \frac{1}{16\mu\tau}\|\lambda_+^r(\widehat{x}^r, \widehat{y}^r) - \lambda^r\|^2 + n_\mathcal{Q}^r
\end{aligned}
\tag{146}
$$

*or*

$$
\begin{aligned}
&\left\{\frac{1}{4\alpha}\left\|\mathbb{E}\left[x^{r+1} - x^r\right]\right\|^2, \frac{1}{4\beta}\left\|\mathbb{E}\left[y^{r+1} - y^r\right]\right\|^2, \frac{p}{4\omega}\|\widehat{x}^{r+1} - \widehat{x}^r\|^2, \frac{(1-\varphi)C_z}{8}\|z^r - z^\star(x^r, y^r)\|^2\right\} \\
&\leq C_{\mathrm{w}}^2 p^2 \zeta^2 \sigma_{\mathrm{w}}^2 \Lambda^2 \mu\tau
\end{aligned}
\tag{147}
$$

*where $C_{\mathrm{w}}^2 \triangleq 2 \cdot 8 \cdot 6^2 \cdot 2^2$, $C_z$ is defined in* (143), *and*

$$
\|\lambda^r - \lambda_+^r(\widehat{x}^r, \widehat{y}^r)\| \leq 8\mu\tau \cdot 2 \cdot 6p\zeta\sigma_{\mathrm{w}}\Lambda.
\tag{148}
$$

*Proof.* From (91), it is clear that if we can select the step-sizes properly so that the coefficients in front of $\|\mathbb{E}x^{r+1} - x^r\|^2$, $\|\mathbb{E}y^{r+1} - y^r\|^2$, $\|\widehat{x}^{r+1} - \widehat{x}^r\|^2$, $\|\widehat{y}^{r+1} - \widehat{y}^r\|^2$ are strictly negative, then the potential function $Q^r$ will be decreasing. To be more specific, the step-sizes are chosen as follows.

1) Selection of $\alpha$. Given the condition of (55), we request

$$
\frac{1}{2\alpha} - 18\frac{\tau}{2\mu}(\ell_g^2 + \ell_\gamma^2)\left(4\sigma_1^2(1-\theta) + \frac{\sigma_3^2\theta}{\alpha^2}\right) - 16\mu(\ell_g^2 + L_\gamma^2)\frac{\sigma_3^2}{\alpha^2} - C_z\left(2L_z^2 + \frac{\eta L_z}{2}\right) > \frac{1}{4\alpha} > 0.
\tag{149}
$$

2) Selection of $\beta$.

$$
\frac{1}{2\beta} - 18\frac{\tau}{2\mu}(\ell_g^2 + \ell_\gamma^2)\left(4\sigma_1^2(1-\theta) + \frac{\sigma_3^2\theta}{\beta^2}\right) - 16\mu(\ell_g^2 + L_\gamma^2)\frac{\sigma_3^2}{\beta^2} - 2\alpha(\lambda^{r+1}L_g)^2 L_z^2 - C_z\left(2L_z^2 + \frac{\eta L_z}{2}\right) > \frac{1}{4\beta} > 0.
\tag{150}
$$

3) Selection of $\omega$.

$$
\frac{1}{2\omega} - \left(\frac{1}{\zeta} + \frac{4p}{p-L}\right) - 6\zeta\sigma_1^2 > \frac{1}{4\omega} > 0,
\tag{151}
$$

i.e.,

$$
\omega < \frac{1}{4}\frac{1}{\frac{1}{\zeta} + \frac{4p}{p-L} + 6\zeta\sigma_1^2}.
\tag{152}
$$

Then, consider the following two cases:

*Case 1.*

$$
\begin{aligned}
&\frac{1}{2}\max\left\{\frac{1}{4\alpha}\left\|\mathbb{E}\left[x^{r+1} - x^r\right]\right\|^2, \frac{1}{4\beta}\left\|\mathbb{E}\left[y^{r+1} - y^r\right]\right\|^2, \frac{p}{4\omega}\|\widehat{x}^{r+1} - \widehat{x}^r\|^2,\right. \\
&\qquad\left.\frac{p}{4\omega}\|\widehat{y}^{r+1} - \widehat{y}^r\|^2, \frac{(1-\varphi)C_z}{2}\|z^r - z^\star(x^r, y^r)\|^2, \frac{1}{8\mu\tau}\|\lambda_+^r(\widehat{x}^r, \widehat{y}^r) - \lambda^r\|^2\right\} \\
&> 6p\zeta\left(\|y^\star(\widehat{x}^r, \widehat{y}^r; \lambda_+^r(\widehat{x}^r, \widehat{y}^r)) - \bar{y}^\star(\widehat{x}^r, \widehat{y}^r)\|^2 + \|x^\star(\widehat{x}^r, \widehat{y}^r; \lambda_+^r(\widehat{x}^r, \widehat{y}^r)) - \bar{x}^\star(\widehat{x}^r, \widehat{y}^r)\|^2\right).
\end{aligned}
$$

In this case, we can have

$$
\begin{aligned}
&\mathcal{Q}^{r+1} - \mathcal{Q}^r \\
&\leq -\frac{1}{8\alpha} \left\| \mathbb{E}\left[x^{r+1} - x^r\right] \right\|^2 - \frac{1}{8\beta} \left\| \mathbb{E}\left[y^{r+1} - y^r\right] \right\|^2 - \frac{p}{8\omega} \|\widehat{x}^{r+1} - \widehat{x}^r\|^2 - \frac{p}{8\omega} \|\widehat{y}^{r+1} - \widehat{y}^r\|^2 \\
&\quad - \frac{(1-\varphi)C_z}{4} \|z^r - z^\star(x^r, y^r)\|^2 - \frac{1}{16\mu\tau} \|\lambda_+^r(\widehat{x}^r, \widehat{y}^r) - \lambda^r\|^2 + n_{\mathcal{Q}}^r,
\end{aligned}
\tag{153}
$$

meaning that $\mathcal{Q}^r$ is decreasing at each step.

*Case 2.*

$$
\begin{aligned}
&\frac{1}{2} \max\Bigg\{ \frac{1}{4\alpha} \left\| \mathbb{E}\left[x^{r+1} - x^r\right] \right\|^2, \frac{1}{4\beta} \left\| \mathbb{E}\left[y^{r+1} - y^r\right] \right\|^2, \frac{p}{4\omega} \|\widehat{x}^{r+1} - \widehat{x}^r\|^2, \frac{p}{4\omega} \|\widehat{y}^{r+1} - \widehat{y}^r\|^2, \\
&\qquad \frac{(1-\varphi)C_z}{2} \|z^r - z^\star(x^r, y^r)\|^2, \frac{1}{8\mu\tau} \|\lambda_+^r(\widehat{x}^r, \widehat{y}^r) - \lambda^r\|^2 \Bigg\} \\
&\leq 6p\zeta \big( \|y^\star(\widehat{x}^r, \widehat{y}^r; \lambda_+^r(\widehat{x}^r, \widehat{y}^r)) - \bar{y}^\star(\widehat{x}^r, \widehat{y}^r)\|^2 + \|x^\star(\widehat{x}^r, \widehat{y}^r; \lambda_+^r(\widehat{x}^r, \widehat{y}^r)) - \bar{x}^\star(\widehat{x}^r, \widehat{y}^r)\|^2 \big).
\end{aligned}
$$

Recall the weak error bound

$$
\begin{aligned}
&\|y^\star(\widehat{x}, \widehat{y}; \lambda_+(\widehat{x}, \widehat{y})) - \bar{y}^\star(\widehat{x}, \widehat{y})\|^2 + \|x^\star(\widehat{x}, \widehat{y}; \lambda_+(\widehat{x}, \widehat{y})) - \bar{x}^\star(\widehat{x}, \widehat{y})\|^2 \\
&\leq \sigma_{\mathrm{w}} \|\lambda - \lambda_+(\widehat{x}, \widehat{y})\| \|\lambda(\widehat{x}, \widehat{y}) - \lambda_+(\widehat{x}, \widehat{y})\|
\end{aligned}
\tag{154}
$$

where $\lambda(v) \in \arg\max_{\lambda \geq 0} K(\bar{x}^\star(v), \bar{y}^\star(v), v; \lambda)$.

We can get

$$
\begin{aligned}
&\|y^\star(\widehat{x}^r, \widehat{y}^r; \lambda_+^{r+1}(\widehat{x}^r, \widehat{y}^r)) - \bar{y}^\star(\widehat{x}^r, \widehat{y}^r)\|^2 + \|x^\star(\widehat{x}^r, \widehat{y}^r; \lambda_+^{r+1}(\widehat{x}^r, \widehat{y}^r)) - \bar{x}^\star(\widehat{x}^r, \widehat{y}^r)\|^2 \\
&\leq 2\sigma_{\mathrm{w}} \Lambda \|\lambda^r - \lambda_+^r(\widehat{x}^r, \widehat{y}^r)\|,
\end{aligned}
\tag{155}
$$

which gives

$$
\|\lambda^r - \lambda_+^r(\widehat{x}^r, \widehat{y}^r)\| \leq 8\mu\tau \cdot 2 \cdot 6p\zeta\sigma_{\mathrm{w}}\Lambda.
\tag{156}
$$

Then, we can have

$$
\begin{aligned}
&\frac{p}{4\omega} \|\widehat{y}^{r+1} - \widehat{y}^r\|^2 \\
&\leq 2 \cdot 6p\zeta \big( \|y^\star(\widehat{x}^r, \widehat{y}^r; \lambda_+^r(\widehat{x}^r, \widehat{y}^r)) - \bar{y}^\star(\widehat{x}^r, \widehat{y}^r)\|^2 + \|x^\star(\widehat{x}^r, \widehat{y}^r; \lambda_+^r(\widehat{x}^r, \widehat{y}^r)) - \bar{x}^\star(\widehat{x}^r, \widehat{y}^r)\|^2 \big) \\
&\leq 2 \cdot 6p\zeta 2\sigma_{\mathrm{w}}\Lambda \|\lambda^r - \lambda_+^r(\widehat{x}^r, \widehat{y}^r)\| \\
&\leq 2 \cdot 8 \cdot 6^2 \cdot 2^2 p^2 \zeta^2 \sigma_{\mathrm{w}}^2 \Lambda^2 \mu\tau.
\end{aligned}
\tag{157}
\tag{158}
$$

Similarly,

$$
\begin{aligned}
&\left\{ \frac{1}{4\alpha} \left\| \mathbb{E}\left[x^{r+1} - x^r\right] \right\|^2, \frac{1}{4\beta} \left\| \mathbb{E}\left[y^{r+1} - y^r\right] \right\|^2, \frac{p}{4\omega} \|\widehat{x}^{r+1} - \widehat{x}^r\|^2, \frac{(1-\varphi)C_z}{2} \|z^r - z^\star(x^r, y^r)\|^2 \right\} \\
&\leq \underbrace{2 \cdot 8 \cdot 6^2 \cdot 2^2}_{\triangleq C_{\mathrm{w}}^2} p^2 \zeta^2 \sigma_{\mathrm{w}}^2 \Lambda^2 \mu\tau.
\end{aligned}
\tag{159}
$$

These results imply that the iterates generated by TSP will converge to some point within a ball with a radius of $\mathcal{O}(\Lambda^2 \mu\tau)$.

$\square$

## C. Boundedness of Dual Variable, LL Variables, and Potential Function

### C.1. Boundedness of Dual Variable

**Lemma 2** (Formal) *Under A1-A5, suppose that the sequence $\{x^r, y^r, z^r, \widehat{x}^r, \widehat{y}^r, \lambda^r, \forall r\}$ is generated by TSP. Assume that $y^r$ and $h_y^f$ and $h_y^g$ are bounded. When $\delta > 0$, $p = \Theta(\Lambda)$, $\gamma = \mathcal{O}(1)$, $\delta = \omega = \eta = \zeta = \beta = \tau = \mathcal{O}(T^{-1/2})$, such that $p > L$ and $\alpha, \beta, \omega$ satisfy (149), (150), (151), then, the sequence $\{\lambda^r\}$ is upper bounded, i.e., $\lambda^r \leq \Lambda$ for all $r$, given a sufficiently large $T$, where $\Lambda$ is a constant.*

*Proof.* From the update rule of variable $y$, we can obtain

$$\lambda^{r+1}\left(h_y^g(x^r, y^r) + \frac{z^r - y^r}{\gamma}\right) = -\left(h_y^f(x^r, z^r) + p(y^r - \widehat{y}^r) + \frac{1}{\beta}(y^{r+1} - y^r)\right)$$

Multiplying $y^r - z^r$ on both sides gives

$$\left\langle \lambda^{r+1}h_y^g(x^r, y^r), y^r - z^r \right\rangle$$
$$\leq -\left\langle h_y^f(x^r, y^r) + p(y^r - z^r) + \frac{1}{\beta}(y^{r+1} - y^r), y^r - z^r \right\rangle + \frac{\lambda^{r+1}}{\gamma}\|y^r - z^r\|^2. \tag{160}$$

Note that when $\gamma < 1/\rho$, the following function

$$\varphi(x, y, z) \triangleq g(x, z) + \frac{1}{2\gamma}\|z - y\|^2 \tag{161}$$

is strongly convex w.r.t. $z$, i.e.,

$$\varphi(x, y, z) \geq \varphi(x, y, y) + \langle \nabla_z \varphi(x, y, y), z - y \rangle. \tag{162}$$

Therefore, we have

$$\widehat{g}(x^r, z^r) + \frac{1}{2\gamma}\|z^r - y^r\|^2 \geq \widehat{g}(x^r, y^r) + \langle h_y^g(x^r, y^r), z^r - y^r \rangle, \tag{163}$$

which is equivalent to

$$\langle h_y^g(x^r, y^r), y^r - z^r \rangle \geq \widehat{g}(x^r, y^r) - \widehat{g}(x^r, z^r) - \frac{1}{2\gamma}\|z^r - y^r\|^2 \tag{164}$$

by some simple algebra manipulations. Subsequently, we can get

$$\lambda^{r+1}\left(\widehat{g}(x^r, y^r) - \widehat{g}(x^r, z^r) - \frac{1}{2\gamma}\|z^r - y^r\|^2\right)$$
$$\leq -\left\langle h_y^f(x^r, y^r) + p(y^r - z^r) + \frac{1}{\beta}(y^{r+1} - y^r), y^r - z^r \right\rangle + \frac{\lambda^{r+1}}{\gamma}\|y^r - z^r\|^2. \tag{165}$$

We assume that $\lambda^r$ and $h^r$ are bounded. If $w^r = \widehat{g}(x^r, y^r) - \widehat{g}(x^r, z^r) - \frac{1}{2\gamma}\|z^r - y^r\|^2 - \delta < 0$, it implies that $h^{r+1} \leq \max\{(1 - \theta)h^r + \theta w^r, 0\}$ is bounded automatically, giving the boundedness of $\lambda_+^r$ and $\lambda^{r+1}$. Otherwise, note that from (7a) we have

$$\frac{1}{\theta}(h^{r+1} - (1 - \theta)h^r) + \delta = \widehat{g}(x^r, y^r) - \widehat{g}(x^r, z^r) - \frac{1}{2\gamma}\|z^r - y^r\|^2.$$

Substituting it back to (165) yields

$$\lambda^{r+1} \leq -\frac{1}{\delta}\left(\left\langle h_y^f(x^r, y^r) + p(y^r - z^r) + \frac{1}{\beta}(y^{r+1} - y^r), y^r - z^r \right\rangle + \frac{\lambda^{r+1}}{\gamma}\|y^r - z^r\|^2\right)$$
$$\leq \frac{1}{\delta}\|h_y^f(x^r, y^r)\|\|y^r - z^r\| + \frac{p}{\delta}\|y^r - z^r\|\|y^r - z^r\| + \frac{1}{\beta\delta}\|y^{r+1} - y^r\|\|y^r - z^r\| + \frac{\lambda^{r+1}}{\gamma\delta}\|y^r - z^r\|^2. \tag{166}$$

From (8), we know that

$$z^{r+1} = z^r - \eta \left( h_y^g(x^r, z^r) + \frac{1}{\gamma}(z^r - y^r) \right), \tag{167}$$

which gives

$$y^r - z^r = \gamma(z^{r+1} - z^r) + \eta\gamma h_y^g(x^r, z^r), \tag{168}$$

or equivalently

$$y^r - z^r = (\gamma - 1)(z^{r+1} - z^\star(x^r, y^r) + z^\star(x^r, y^r) - z^r) + \eta\gamma h_y^g(x^r, z^r). \tag{169}$$

Applying the triangle inequality gives

$$\|y^r - z^r\|$$
$$\leq |\gamma - 1|\|z^{r+1} - z^\star(x^r, y^r) + z^\star(x^r, y^r) - z^r\| + \eta\gamma\|h_y^g(x^r, z^r)\| \tag{170}$$
$$\leq |\gamma - 1|\|\varrho + 1\|\|z^r - z^\star(x^r, y^r)\| + \eta\gamma\|h_y^g(x^r, z^r)\| + |\gamma - 1||n_{z'}^r(\eta, \varepsilon_{g_z}^r)| \tag{171}$$
$$\overset{(a)}{\leq} |\gamma - 1||\varrho + 1|2p\zeta C_w \sigma_w \Lambda \frac{\sqrt{\mu\tau}}{\sqrt{C_z}} + \eta\gamma\|h_y^g(x^r, z^r)\| + |\gamma - 1||n_{z'}^r(\eta, \varepsilon_{g_z}^r)| \tag{172}$$

where in $(a)$ we apply (159) that serves as an upper bound for the size of different iterates.

Substituting (172) back to (166) yields

$$\lambda^{r+1} \leq \frac{1}{\delta}\|h_y^f(x^r, y^r)\|\|y^r - z^r\| + \frac{p}{\delta}\|y^r - z^r\|\|y^r - z^r\|$$
$$+ \frac{1}{\beta\delta}\|y^{r+1} - y^r\|\|y^r - z^r\| + \frac{\lambda^{r+1}}{\gamma\delta}\|y^r - z^r\|^2 \tag{173}$$

$$\leq \frac{1}{\delta}\|h_y^f(x^r, y^r)\|\|y^r - z^r\| + \frac{p}{\delta}\|y^r - z^r\|\|y^r - z^r\| + \frac{\lambda^{r+1}}{\gamma\delta}\|y^r - z^r\|^2$$
$$+ \frac{1}{\beta\delta}\left(\|\mathbb{E}y^{r+1} - y^r\| + \beta\|\varepsilon_y^r\|\right)\|y^r - z^r\| \tag{174}$$

$$\leq \|h_y^f(x^r, y^r)\| \left( |\gamma - 1|\,|\varrho + 1|\,2p\zeta C_w \sigma_w \Lambda \frac{\sqrt{\mu\tau}}{\delta\sqrt{C_z}} + \frac{\eta\gamma}{\delta}\|h_y^g(x^r, z^r)\| + |\gamma - 1||n_{z'}^r(\eta, \varepsilon_{g_z}^r)| \right)$$
$$+ \left( \frac{p}{\delta} + \frac{\Lambda}{\gamma\delta} \right) \left( 2p\zeta C_w \sigma_w \Lambda \frac{\sqrt{\mu\tau}}{\sqrt{C_z}} + \eta\gamma\|h_y^g(x^r, z^r)\| + |\gamma - 1||n_{z'}^r(\eta, \varepsilon_{g_z}^r)| \right)^2$$
$$+ \left( \frac{2C_w p\zeta\sigma_w\Lambda\sqrt{\tau}}{\sqrt{\beta}\delta} + \frac{\|\varepsilon_y^r\|}{\delta} \right) \left( 2p\zeta C_w \sigma_w \Lambda \frac{\sqrt{\mu\tau}}{\sqrt{C_z}} + \eta\gamma\|h_y^g(x^r, z^r)\| + |\gamma - 1||n_{z'}^r(\eta, \varepsilon_{g_z}^r)| \right). \tag{175}$$

We choose

$$p = \Theta(\Lambda),\ \gamma = \mathcal{O}(1),\ \delta = \mathcal{O}(\epsilon),\ \alpha = \eta = \zeta = \beta = \tau = \mathcal{O}(T^{-1/2}) \quad \text{or} \quad \text{of the same small order}, \tag{176}$$

then, it can be easily checked that the three terms in (175) can be upper bounded by

$$\lambda^{r+1} \leq \mathcal{O}\left(\frac{\sqrt{\mu\tau}}{\delta}\right) + \mathcal{O}\left(\frac{\Lambda\mu^2}{\delta}\right) + \mathcal{O}\left(\frac{\Lambda\sqrt{\tau}}{\sqrt{\beta}\delta}\mu\right) = \mathcal{O}(\Lambda). \tag{177}$$

Thus, we have $\lambda^{r+1} < \Lambda = \mathcal{O}(1)$ when the step-sizes are sufficiently small. In turn, this implies the upper boundedness of $h^{r+1}$ immediately.

$\square$

**C.2. Boundedness of Variables** $(y^r, \bar{y}^\star(\hat{x}^r, \hat{y}^r), y^\star(\hat{x}^r, \hat{y}^r; \lambda^{r+1})$ **and** $x^r, \bar{x}^\star(\hat{x}^r, \hat{y}^r), x^\star(\hat{x}^r, \hat{y}^r; \lambda^{r+1}))$

> **Lemma 9.** *Under A1-A5, suppose that the sequence* $\{x^r, y^r, z^r, \hat{x}^r, \hat{y}^r, \lambda^r, \lambda^r_+, \forall r\}$ *is generated by TSP. Assume that* $y^r, \bar{y}^\star(\hat{x}^r, \hat{y}^r)$ *are bounded and boundedness of the gradient estimate. Then, we have* $\bar{y}^\star(\hat{x}^{r+1}, \hat{y}^{r+1}), y^\star(\hat{x}^r, \hat{y}^r; \lambda^{r+1}), y^{r+1}$ *are also bounded.*

*Proof.* We prove these results by induction. First, we assume that $y^r, \bar{y}^\star(\hat{x}^r, \hat{y}^r)$ are bounded, which gives the (gradient) Lipschitz continuity of $g(x, \cdot)$ at these points.

Recall the bounded level set assumption that let

$$\psi(x, y) = f(x, y), y \in \mathcal{Y}(x) \triangleq \{y | g(x, y) - g^\star_\gamma(x, y) \le \delta\}. \tag{178}$$

Under the assumption that $y^r, \bar{y}^\star(w^r, z^r)$ are bounded and $\alpha = \beta = \tau = \mathcal{O}(T^{-1/2})$, we can have either the monotonic decrease of the potential function up to a small error or convergence of the iterates. By the fact that $\psi(\bar{x}^\star(\hat{x}^{r+1}, \hat{y}^{r+1}), \bar{y}^\star(\hat{x}^{r+1}, \hat{y}^{r+1})) \le P(\hat{x}^{r+1}, \hat{y}^{r+1})$, for any $(x^1, y^1, \hat{x}^1, \hat{y}^1; \lambda^1)$, there exists a constant $R$ such that

$$\{\bar{x}^\star(\hat{x}^{r+1}, \hat{y}^{r+1}), \bar{y}^\star(\hat{x}^{r+1}, \hat{y}^{r+1}) | P(\hat{x}^{r+1}, \hat{y}^{r+1}) \le \mathcal{Q}^{r+1}\} \subseteq \mathcal{B}(R(x^1, y^1, \hat{x}^1, \hat{y}^1; \lambda^1)), \tag{179}$$

which gives that $\bar{x}^\star(\hat{x}^{r+1}, \hat{y}^{r+1})$ and $\bar{y}^\star(\hat{x}^{r+1}, \hat{y}^{r+1})$ are bounded.

Applying the weak error bound result gives

$$\|y^\star(\hat{x}^r, \hat{y}^r; \lambda^r_+(\hat{x}^r, \hat{y}^r)) - \bar{y}^\star(\hat{x}^r, \hat{y}^r)\| \le \sqrt{\sigma_w \|\lambda^r - \lambda^r_+(\hat{x}^r, \hat{y}^r)\| \|\mathbb{E}\lambda(\hat{x}^r, \hat{y}^r) - \lambda^r_+(\hat{x}^r, \hat{y}^r)\|} \overset{(a)}{=} \mathcal{O}(1)$$

where $(a)$ holds due to the facts (234a), (140) and the boundedness of the dual variable. So, we can have that $y^\star(\hat{x}^r, \hat{y}^r; \lambda^r_+(\hat{x}^r, \hat{y}^r)) = \mathcal{O}(1)$ is bounded. Additionally, it can be checked that

$$\|y^\star(\hat{x}^r, \hat{y}^r; \lambda^{r+1}) - y^\star(\hat{x}^r, \hat{y}^r; \lambda^r_+(\hat{x}^r, \hat{y}^r))\| \overset{(26)}{\le} \frac{p+L}{p-L} \|\mathbb{E}\lambda^{r+1} - \lambda^{r+1}_+(\hat{x}^r, \hat{y}^r)\| \overset{(a)}{=} \mathcal{O}(1)$$

where $(a)$ holds due to the boundedness of the dual variable.

Note that $K(x, \cdot, z, \hat{x}^r, \hat{y}^r; \lambda^{r+1})$ is strongly convex with modulus $p - L$ and gradient Lipschitz continuous with parameter $p + L$. From (Hardt & Simchowitz, 2018), we have

$$\|y^{r+1} - y^\star(\hat{x}^r, \hat{y}^r; \lambda^{r+1})\| \le \left(1 - \frac{p-L}{p+L}\right) \|y^r - y^\star(\hat{x}^r, \hat{y}^r; \lambda^{r+1})\| + n_y(\beta \varepsilon^r_y), \tag{180}$$

where $n_y$ denotes the random noise term. Given that the gradient estimates are bounded and $\beta = \mathcal{O}(T^{-1/2})$, we can conclude the boundedness of $y^{r+1}$. Similarly, the above boundedness properties are also true for $x^{r+1}, x^\star(\hat{x}^r, \hat{y}^r; \lambda^{r+1}), x^\star(\hat{x}^r, \hat{y}^r; \lambda^r_+(\hat{x}^r, \hat{y}^r)) = \mathcal{O}(1)$. $\square$

**C.3. Lower Boundedness of** $Q^r$

From (92), we know that

$$Q(x^r, y^r, z^r, \hat{x}^r, \hat{y}^r; \lambda^r)$$

$$\ge K(x^r, y^r, z^r, \hat{x}^r, \hat{y}^r; \lambda^r) - 2D(\hat{x}^r, \hat{y}^r; \lambda^r) + 2P(\hat{x}^r, \hat{y}^r) - \frac{1}{c}M(\lambda^r, h^r) - \underline{f} \tag{181}$$

$$= P(\hat{x}^r, \hat{y}^r) + K(x^r, y^r, z^r, \hat{x}^r, \hat{y}^r; \lambda^r) - D(\hat{x}^r, \hat{y}^r; \lambda^r) + (P(\hat{x}^r, \hat{y}^r) - D(\hat{x}^r, \hat{y}^r; \lambda^r) - \frac{1}{c}M(\lambda^r, h^r) - \underline{f}$$

$$\overset{(a)}{\ge} P(\hat{x}^r, \hat{y}^r) - \frac{1}{c}M(\lambda^r, h^r) - \underline{f} \overset{(b)}{\ge} \underline{Q} \tag{182}$$

where $(a)$ holds due to 1) $K(x^r, y^r, z^r, \hat{x}^r, \hat{y}^r; \lambda^r) - D(\hat{x}^r, \hat{y}^r; \lambda^r) \ge 0$ based on the definition of $D(\hat{x}^r, \hat{y}^r; \lambda^r)$ and 2) note that $P(\hat{x}, \hat{y}) = \min_{x,y} \max_{\lambda \ge 0} f(x, y) + \lambda(g(x, y) - g^\star_\gamma(x, y) - \delta) + \frac{p}{2}\|x - \hat{x}\|^2 + \frac{p}{2}\|y - \hat{y}\|^2$ and $P(\hat{x}^r, \hat{y}^r) - D(\hat{x}^r, \hat{y}^r; \lambda^r) \ge 0$, which is true because the minimax equality theorem (Kakutani, 1941; Bertsekas et al., 2003) holds when $K(x, y, z, \hat{x}, \hat{y}; \lambda)$ is strongly convex in $x, y$ and linear (concave) in $\lambda$ when variables are within compact set. Also, as $h^1 = 0$, we have $Q^1 \triangleq Q(x^1, y^1, z^1, \hat{x}^1, \hat{y}^1; \lambda^1) \ge 0$, and $(b)$ holds due to the definition of $P(\hat{x}^r, \hat{y}^r)$ and the lower boundedness of function $g()$, where $\underline{Q}$ denotes the lower bound of $Q^r$.

# D. Theoretical Convergence Results

### D.1. Proof of Theorem 1

*Proof.* **Stationarity**. Recall

$$\mathcal{G}(x^r, y^r; \lambda^{r+1}) = \begin{bmatrix} \nabla_x \mathcal{L}(x^r, y^r; \lambda^{r+1}) \\ \nabla_y \mathcal{L}(x^r, y^r; \lambda^{r+1}) \end{bmatrix}. \tag{183}$$

For the block-$x$, we have

$$\left\| \nabla_x \mathcal{L}(x^r, y^r; \lambda^{r+1}) \right\|$$
$$\leq \left\| \nabla_x f(x^r, y^r) + \lambda^{r+1} (\nabla_x g(x^r, y^r) - \nabla_x g(x^r, z^\star(x^r, y^r))) \right\|$$
$$\overset{(a)}{\leq} \frac{1}{\alpha} \left\| \mathbb{E} \left[ x^{r+1} - x^r \right] \right\| + \Lambda \| \nabla_x g(x^r, z^\star(x^r, y^r)) - \nabla_x g(x^r, z^r) \| + p \left\| \mathbb{E} \left[ x^r - \widehat{x}^r \right] \right\|$$
$$\overset{(b)}{\leq} \left( \frac{1}{\alpha} + p \right) \left\| \mathbb{E} \left[ x^{r+1} - x^r \right] \right\| + \frac{p}{\omega} \left\| \widehat{x}^{r+1} - \widehat{x}^r \right\| + \Lambda L_g \| z^r - z^\star(x^r, y^r) \| + p\alpha \| \varepsilon_x^r \|$$

where in $(a)$ we apply the following optimality condition of the $x$-subproblem

$$\mathbb{E} x^{r+1} = \mathbb{E} x^r - \alpha \left( \nabla_x f(x^r, y^r) + \lambda^{r+1} \left( \nabla_x g(x^r, y^r) - \nabla_x g(x^r, z^r) \right) + p\mathbb{E}(x^r - \widehat{x}^r) \right),$$

and $(b)$ results from the fact that $\| \widehat{x}^{r+1} - \widehat{x}^r - [\mathbb{E} \widehat{x}^{r+1} - \widehat{x}^r] \| \leq \omega \| x^{r+1} - \mathbb{E} x^{r+1} \| \leq \alpha \omega \| \varepsilon_x^r \|$.

For the block-$y$, we have

$$\left\| \nabla_y \mathcal{L}(x^r, y^r; \lambda^{r+1}) \right\|$$
$$\leq \left\| \nabla_y f(x^r, y^r) + \lambda^{r+1} \left( \nabla_y g(x^r, y^r) + \frac{1}{\gamma} (z^\star(x^r, y^r) - y^r) \right) \right\| \tag{184}$$
$$\overset{(a)}{\leq} \frac{1}{\beta} \left\| \mathbb{E} \left[ y^{r+1} - y^r \right] \right\| + \frac{\Lambda}{\gamma} \| \mathbb{E} z^r - z^\star(x^r, y^r) \| + p \| \mathbb{E} y^r - \widehat{y}^r \| \tag{185}$$
$$\overset{(10)}{\leq} \left( \frac{1}{\beta} + p \right) \left\| \mathbb{E} \left[ y^{r+1} - y^r \right] \right\| + \frac{\Lambda}{\gamma} \| z^r - z^\star(x^r, y^r) \| + \frac{p}{\omega} \left\| \mathbb{E} \left[ \widehat{y}^{r+1} - \widehat{y}^r \right] \right\| + p\beta \| \varepsilon_y^r \| \tag{186}$$

where in $(a)$ we apply the following optimality condition of the $y$-subproblem

$$\nabla_y f(x^r, y^r) + \lambda^{r+1} \nabla_y g(x^r, y^r) = \frac{\mathbb{E} y^r - y^{r+1}}{\beta} - \lambda^{r+1} \frac{z^r - y^r}{\gamma} - p\mathbb{E}(y^r - \widehat{y}^r). \tag{187}$$

Therefore, the primal optimality gap can be quantified as follows:

$$\| \mathcal{G}(x^r, y^r; \lambda^{r+1}) \|^2$$
$$\leq 4 \left( \frac{1}{\alpha} + p \right)^2 \left\| \mathbb{E} \left[ x^{r+1} - x^r \right] \right\|^2 + 4 \left( \frac{1}{\beta} + p \right)^2 \left\| \mathbb{E} \left[ y^{r+1} - y^r \right] \right\|^2 + \frac{4p^2}{\omega^2} \left\| \mathbb{E} \left[ \widehat{x}^{r+1} - \widehat{x}^r \right] \right\|^2$$
$$+ \frac{4p^2}{\omega^2} \left\| \mathbb{E} \left[ \widehat{y}^{r+1} - \widehat{y}^r \right] \right\|^2 + 4 \left( L_g^2 + \frac{1}{\gamma^2} \right) \Lambda^2 \| z^r - z^\star(x^r, y^r) \|^2 + 4p^2 (\alpha^2 \| \varepsilon_x^r \|^2 + \beta^2 \| \varepsilon_y^r \|^2). \tag{188}$$

Note that we choose

$$\gamma = \mathcal{O}(1), \quad p = \mathcal{O}(\Lambda) = \mathcal{O}(1), \quad \alpha = \beta = \eta = \theta = \tau = \mathcal{O} \left( \frac{1}{\sqrt{T}} \right) \tag{189}$$

and (176). It can be easily verified that these choices of parameters also satisfy (149), (150), and (151). If case 2 shown in (18) appears, then it is directly implied that $\| \mathcal{G}(x^r, y^r; \lambda^{r+1}) \| \to \epsilon$. Otherwise, we need to analyze the noise term more carefully. From (17), we have the following inequalities.

The first one is

$$\min\left\{\frac{1}{8\alpha}, \frac{1}{8\beta}, \frac{p}{8\omega}\right\} \left(\left\|\mathbb{E}\left[x^{r+1} - x^r\right]\right\|^2 + \left\|\mathbb{E}\left[y^{r+1} - y^r\right]\right\|^2 + \left\|\widehat{x}^{r+1} - \widehat{x}^r\right\|^2 + \left\|\widehat{y}^{r+1} - \widehat{y}^r\right\|^2\right)$$
$$\leq Q^r - Q^{r+1} + n_Q^r. \tag{190}$$

The second one is

$$\frac{(1-\varphi)C_z}{8}\left(\|z^r - z^\star(x^r, y^r)\|^2\right) \leq Q^r - Q^{r+1} + n_Q^r. \tag{191}$$

Then, we let

$$\rho_1 \triangleq \min\left\{\frac{1}{8\alpha}, \frac{1}{8\beta}, \frac{p}{8\omega}\right\}, \tag{192a}$$

$$\rho_2 \triangleq \max\left\{3\left(\frac{1}{\alpha} + p\right)^2, 3\left(\frac{1}{\beta} + p\right)^2, \frac{3p^2}{\omega^2}\right\}. \tag{192b}$$

Plugging (190), (191) into (188) along with (192a) and (192b), we can have

$$\|\mathcal{G}(x^r, y^r; \lambda^{r+1})\|^2 \leq \left(\frac{\rho_2}{\rho_1} + \frac{32\left(L_g^2 + \gamma^{-1}\right)^2 \Lambda^2}{(1-\varphi)C_z}\right)\left(Q^r - Q^{r+1} + n_Q^r\right). \tag{193}$$

From the above analysis, it can be seen that the boundedness of the gradient estimate is essential for ensuring the boundedness of iterates, especially for the dual variable. Equivalently, the gradient estimate error is bounded. Let $\varepsilon$ denote any noise term, e.g., $\varepsilon_x$. From the noise terms shown in (93), it is apparent that each noise term is either in a linear form or a quadratic form, coupled with the corresponding step-sizes, meaning that it takes the form $\upsilon\langle\varepsilon, \phi\rangle$ or $\upsilon^2\|\varepsilon\|^2$, where $\upsilon$ represents the step-size and $\phi$ denotes the coefficient vector, which is either the iterates or the gradients of the loss functions. There are a total of 10 linear terms and 6 quadratic terms. Let $\varepsilon_{\max}^r$ denote the noise term with the largest magnitude among all noise terms at the $r$th iteration, and let $\vartheta$ be the corresponding step-size and $G$ be the largest magnitude among all $\phi$ (note that we have shown that the iterates generated by TSP are bounded and that all loss functions are smooth).

Define
$$t_1 \triangleq \min\left\{r | Q^r - \underline{Q} > \bar{Q}\right\} \wedge T, t_2 \triangleq \min\left\{r | \|\varepsilon_{\max}^r\| > \frac{G}{50\vartheta}\right\} \wedge T, t \triangleq \min\{t_1, t_2\} \tag{194}$$

where $a \wedge b$ denotes $\min\{a, b\}$ for any $a, b \in \mathbb{R}$, and the threshold $\bar{Q} = G^2/2$.

For the quadratic term of the noise w.r.t. $\varepsilon^r$, we can have

$$\mathbb{E}\left[\sum_{r<t}\|\varepsilon^r\|^2\right] \leq \mathbb{E}\left[\sum_{r<t}\|\varepsilon^r\|^2\right] \leq \sigma^2 T, \tag{195}$$

due to A4 and A5, after removing the constant factors.

For the cross term, not that $\mathbb{E}_{r-1}\langle\phi^r, \varepsilon^r\rangle = 0$. So, this term is the sum of a martingale difference sequence. Since $t$ is a stopping time, we can apply the optimal stopping theorem and obtain

$$\mathbb{E}\left[\sum_{r\leq t}\langle\phi^r, \varepsilon^r\rangle\right] = 0, \tag{196}$$

which gives

$$-\mathbb{E}\left[\sum_{r<t}\langle\phi^r, \varepsilon^r\rangle\right] \stackrel{(196)}{=} \mathbb{E}\left[\langle\phi^r, \varepsilon^t\rangle\right] \stackrel{(a)}{\leq} G\mathbb{E}\|\varepsilon^t\| \leq G\sqrt{\mathbb{E}\|\varepsilon^t\|^2} \leq G\sqrt{\mathbb{E}\sum_{r\leq T}\|\varepsilon^r\|^2} \leq \sigma G\sqrt{T+1} \leq \sigma G\sqrt{2T}$$

where $(a)$ holds due to the definition of $t$.

Applying the telescoping sum over $r = 1, \ldots, T$ yields

$$\mathbb{E} \sum_{r<t} \|\mathcal{G}(x^r, y^r; \lambda^{r+1})\|^2$$

$$\leq \left( \frac{\rho_2}{\rho_1} + \frac{24 \left( L_g^2 + \gamma^{-1} \right)^2 \Lambda^2}{(1-\varphi)C_z} \right) \mathbb{E} \sum_{r<t} \left( Q^r - Q^{r+1} + n_Q^r \right)$$

$$\overset{(195),(197)}{\leq} \left( \frac{\rho_2}{\rho_1} + \frac{24 \left( L_g^2 + \gamma^{-1} \right)^2 \Lambda^2}{(1-\varphi)C_z} \right) \left( Q^1 - \underline{Q} + \underline{Q} - Q^r + 10\vartheta G \sqrt{2T} + 10\vartheta^2 \sigma^2 T \right),$$

which gives

$$\mathbb{E} Q^t - \underline{Q} + \sum_{r<t} \frac{1}{\tilde{\rho}} \|\mathcal{G}(x^r, y^r; \lambda^{r+1})\|^2 \leq Q^1 - \underline{Q} + 10\vartheta G \sqrt{2T} + 10\vartheta^2 \sigma^2 T, \tag{197}$$

where

$$\frac{1}{\tilde{\rho}} \triangleq \frac{\rho_2}{\rho_1} + \frac{24 \left( L_g^2 + \gamma^{-1} \right)^2 \Lambda^2}{(1-\varphi)C_z} = \mathcal{O}(\vartheta) = \mathcal{O}(T^{-1/2}). \tag{198}$$

The first bound $\mathbb{P}(t_2 < T)$ is

$$\mathbb{P}(t_2 < T) = \mathbb{P} \left( \bigcup_{r<T} \|\varepsilon_{\max}^r\| > \frac{G}{50\vartheta} \right) \overset{(a)}{\leq} \sum_{r<T} \mathbb{P} \left( \|\varepsilon_{\max}^r\| > \frac{G}{50\vartheta} \right) \overset{(b)}{\leq} \frac{2500\vartheta^2 \sigma^2 T}{G^2} \overset{(c)}{\leq} \frac{\varsigma}{4} \tag{199}$$

where in $(a)$, we apply the union bound, in $(b)$ we use Chebyshev's inequality, and $(c)$ holds because we can choose $\vartheta = \mathcal{O}(T^{-1/2})$ such that $0 < \varsigma < 1$.

The second bound $\mathbb{P}(t_1 < T, t_2 = T)$ can be obtained as follows. In this case, we have $Q^{r+1} - \underline{Q} > \bar{Q}$ and $\|\varepsilon^r\| \leq G/(50\vartheta)$. Note that as we always have $Q^r - \underline{Q} \leq \bar{Q}$, which implies that we have bounded gradient. From (193), we have

$$Q^{r+1} - Q^r \leq 10 \left( \vartheta G \|\varepsilon_{\max}^r\| + \vartheta^2 \|\varepsilon_{\max}^r\|^2 \right) \overset{(a)}{\leq} 10 \left( \frac{\vartheta G^2}{50} + \vartheta^2 \frac{G^2}{2500\vartheta^2} \right) \overset{(b)}{\leq} \frac{\bar{Q}}{2} \tag{200}$$

where $(a)$ is true because we choose $\vartheta = \mathcal{O}(T^{-1/2})$ for sufficiently large $T$, and $(b)$ holds due to $G^2 \triangleq 2\bar{Q}$.

Consequently, under the event $\{t_1 < T, t_2 = T\}$, we have

$$Q^t - \underline{Q} = Q^t - Q^{t+1} + Q^{t+1} - \underline{Q} > \frac{\bar{Q}}{2}, \tag{201}$$

which gives

$$\mathbb{P}(t_1 < T, t_2 = T) \leq \mathbb{P} \left( Q^t - \underline{Q} > \frac{\bar{Q}}{2} \right) \overset{(a)}{\leq} \frac{\mathbb{E} Q^t - \underline{Q}}{\bar{Q}/2} \leq \frac{2(Q^1 - \underline{Q} + \sqrt{2} + \sigma^2/40)}{\bar{Q}} \overset{(197)}{\leq} \frac{\varsigma}{4} \tag{202}$$

where in $(a)$ we use Markov's inequality, $(b)$ we choose $\vartheta \leq 1/(10G\sqrt{T})$ in (197), and $(c)$ holds because we choose $\bar{Q} = 8(Q^1 - \underline{Q} + \sqrt{2} + \sigma^2/40)/\varsigma$. Therefore, we have

$$\mathbb{P}(t < T) \leq \mathbb{P}(t_2 < T) + \mathbb{P}(t_1 < T, t_2 = T) \leq \frac{\varsigma}{2}, \tag{203}$$

which gives that $\mathbb{P}(t = T) \geq 1 - \varsigma/2$. Then, from (197) we have

$$\tilde{\rho} \left( Q^1 - \underline{Q} + \sqrt{2} + \sigma^2/40 \right) \geq \mathbb{E} \sum_{r<t} \|\mathcal{G}(x^r, y^r; \lambda^{r+1})\|^2$$

$$\geq \mathbb{P}(t = T) \mathbb{E} \left[ \sum_{r<T} \|\mathcal{G}(x^r, y^r; \lambda^{r+1})\|^2 | t = T \right] \geq \frac{1}{2} \mathbb{E} \left[ \sum_{r<T} \|\mathcal{G}(x^r, y^r; \lambda^{r+1})\|^2 | t = T \right]. \tag{204}$$

Therefore, we can obtain

$$\mathbb{E}\left[\frac{1}{T}\sum_{r<T}\|\mathcal{G}(x^r,y^r;\lambda^{r+1})\|^2|t=T\right] \leq \frac{2\tilde{\rho}\left(Q^1-\underline{Q}+\sqrt{2}+\sigma^2/40\right)}{T} \stackrel{(a)}{=} \frac{\varsigma\tilde{\rho}\bar{Q}}{4T} \stackrel{(b)}{\leq} \frac{\varsigma\epsilon^2}{4} \tag{205}$$

where $(a)$ holds due to the definition of $\bar{Q}$ and $(b)$ is true since $T \geq \bar{Q}/(\vartheta\epsilon^2)$. Let $\mathcal{E} \triangleq \{T^{-1}\sum_{r<T}\|\mathcal{G}(x^r,y^r;\lambda^{r+1})\|^2 > \epsilon\}$ denote the event that the generated iterate does not converge to an $\epsilon$-stationary point. Then, according to Markov's inequality, we have $\mathbb{P}(\mathcal{E}) \leq \varsigma/2$, which gives $\mathbb{P}(t < T \cup \mathcal{E}) \leq \varsigma$.

**Constraint Violation**.

From (130), we can get

$$\||g(x^r,y^r)-g_\gamma^\star(x^r,y^r)-\delta|_+\|^2$$
$$\leq 2\left\|g(x^r,y^r)-g(x^r,z^r)-\frac{1}{2\gamma}\|z^r-y^r\|^2-\delta-\nabla_\lambda K(x^\star(v^r;\lambda^r),y^\star(v^r;\lambda^r),v^r;\lambda^r)\right\|^2$$
$$\qquad +2\||\nabla_\lambda K(x^\star(v^r;\lambda^r),y^\star(v^r;\lambda^r),v^r;\lambda^r)|_+\|^2 \tag{206}$$
$$\stackrel{(234a)}{\leq} 6(\ell_g+\ell_\gamma)^2\left(\frac{\sigma_3^2}{\alpha^2}\|x^{r+1}-x^r\|^2+\frac{\sigma_3^2}{\beta^2}\|y^{r+1}-y^r\|^2+4\sigma_2^2\|\lambda^{r+1}-\lambda^r\|^2\right)$$
$$\qquad +\frac{2}{\tau^2}\|\lambda^r-\lambda_+^r(\widehat{x}^r,\widehat{y}^r)\|^2 \tag{207}$$
$$\leq 6(\ell_g+\ell_\gamma)^2\left(\frac{\sigma_3^2}{\alpha^2}\|x^{r+1}-x^r\|^2+\frac{\sigma_3^2}{\beta^2}\|y^{r+1}-y^r\|^2+4\sigma_2^2\|\lambda^{r+1}-\lambda^r\|^2\right)$$
$$\qquad +\frac{2}{\tau^2}\|\lambda^r-\lambda_+^r(\widehat{x}^r,\widehat{y}^r)\|^2. \tag{208}$$

Note that

$$\|\lambda^{r+1}-\lambda^r\|^2 = \|\lambda^{r+1}-\lambda_+^r(\widehat{x}^r,\widehat{y}^r)+\lambda_+^r(\widehat{x}^r,\widehat{y}^r)-\lambda^r\|^2$$
$$\leq 2\|\lambda_+^r(\widehat{x}^r,\widehat{y}^r)-\lambda^r\|^2+2\|\lambda^{r+1}-\lambda_+^r(\widehat{x}^r,\widehat{y}^r)\|^2 \tag{209}$$
$$\leq 2\|\lambda_+^r(\widehat{x}^r,\widehat{y}^r)-\lambda^r\|^2+2\tau^2\|h^{r+1}-\nabla_\lambda K(x^\star(v^r;\lambda^r),y^\star(v^r;\lambda^r),v^r;\lambda^r)\|^2. \tag{210}$$

Then, we can get

$$\||g(x^r,y^r)-g_\gamma^\star(x^r,y^r)-\delta|_+\|^2$$
$$\leq 6(\ell_g+\ell_\gamma)^2\left(\frac{\sigma_3^2}{\alpha^2}\|x^{r+1}-x^r\|^2+\frac{\sigma_3^2}{\beta^2}\|y^{r+1}-y^r\|^2\right)+\left(6(\ell_g+\ell_\gamma)^28\sigma_2^2+\frac{2}{\tau^2}\right)\|\lambda^r-\lambda_+^r(\widehat{x}^r,\widehat{y}^r)\|^2$$
$$\qquad +12\tau^2(\ell_g+\ell_\gamma)^2\|h^{r+1}-\nabla_\lambda K(x^\star(v^r;\lambda^r),y^\star(v^r;\lambda^r),v^r;\lambda^r)\|^2. \tag{211}$$

From (190) and (191), we have

$$\left(\frac{\sigma_3^2}{\alpha^2}\|x^{r+1}-x^r\|^2+\frac{\sigma_3^2}{\beta^2}\|y^{r+1}-y^r\|^2\right),\frac{2}{\tau^2}\|\lambda^r-\lambda_+^r(\widehat{x}^r,\widehat{y}^r)\|^2 = \mathcal{O}\left(\max\{\alpha,\beta\}\left(Q^r-Q^{r+1}+n_Q^r\right)\right). \tag{212}$$

Similarly, we can get $\tau^2\|h^{r+1}-\nabla_\lambda K(x^\star(v^r;\lambda^r),y^\star(v^r;\lambda^r),v^r;\lambda^r)\|^2 = \mu\tau(Q^r-Q^{r+1}+n_Q^r)$ according to (145).

Applying the telescoping and the same argument as (205), we can get

$$\mathbb{E}\left[\frac{1}{T}\sum_{r<T}|g(x^r,y^r)-g_\gamma^\star(x^r,y^r)-\epsilon|_+^2|t=T\right] \stackrel{(204)}{\leq} \mathcal{O}\left(\frac{\varsigma\bar{Q}}{\sqrt{T}}\right) = \mathcal{O}\left(\frac{1}{\sqrt{T}}\right). \tag{213}$$

**Slackness**. If $\lambda^r = 0$, then it is trivial that $|g(x^r,y^r)-g_\gamma^\star(x^r,y^r)-\delta|\lambda^r$ is zero. So, we only need to consider the case where $\lambda^r > 0$ as follows.

Note that

$$|g(x^r, y^r) - g_\gamma^\star(x^r, y^r) - \delta|^2$$

$$\leq 3|g(x^r, y^r) - g(x^\star(\widehat{x}^r, \widehat{y}^r; \lambda^{r+1}), y^\star(\widehat{x}^r, \widehat{y}^r; \lambda^{r+1}))|^2$$

$$+ 3|g(x^\star(\widehat{x}^r, \widehat{y}^r; \lambda^{r+1}), y^\star(\widehat{x}^r, \widehat{y}^r; \lambda^{r+1})) - g_\gamma^\star(x^\star(\widehat{x}^r, \widehat{y}^r; \lambda^{r+1}), y^\star(\widehat{x}^r, \widehat{y}^r; \lambda^{r+1})) - \delta|^2$$

$$+ 3|g_\gamma^\star(x^r, y^r) - g_\gamma^\star(x^\star(\widehat{x}^r, \widehat{y}^r; \lambda^{r+1}), y^\star(\widehat{x}^r, \widehat{y}^r; \lambda^{r+1}))|^2 \tag{214}$$

$$\overset{(a)}{\leq} 6(\ell_g^2 + \ell_\gamma^2)\left(\|y^r - y^\star(\widehat{x}^r, \widehat{y}^r; \lambda^{r+1})\|^2 + \|x^r - x^\star(\widehat{x}^r, \widehat{y}^r; \lambda^{r+1})\|^2\right)$$

$$+ 3\max\left\{\frac{1}{\tau^2}\|\lambda^r - \lambda_+^r(\widehat{x}^r, \widehat{y}^r)\|^2, \|\delta\|^2\right\} \tag{215}$$

$$\leq \frac{6(\ell_g^2 + \ell_\gamma^2)}{\beta^2(p-L)^2}\|y^{r+1} - y^r\|^2 + \frac{6(\ell_g^2 + \ell_\gamma^2)}{\alpha^2(p-L)^2}\|x^{r+1} - x^r\|^2$$

$$+ 3\max\left\{\frac{1}{\tau^2}\|\lambda^r - \lambda_+^r(\widehat{x}^r, \widehat{y}^r)\|^2, \|\delta\|^2\right\} \tag{216}$$

$$\leq \frac{6(\ell_g^2 + \ell_\gamma^2)}{(p-L)^2}\max\left\{\frac{1}{\alpha^2}, \frac{1}{\beta^2}\right\}\left(\|x^{r+1} - x^r\|^2 + \|y^{r+1} - y^r\|^2\right)$$

$$+ 3\max\left\{\frac{1}{\tau^2}\|\lambda^r - \lambda_+^r(\widehat{x}^r, \widehat{y}^r)\|^2, \|\delta\|^2\right\} \tag{217}$$

where in $(a)$ there are two cases: 1) $g(x^\star(\widehat{x}^r, \widehat{y}^r; \lambda^{r+1}), y^\star(\widehat{x}^r, \widehat{y}^r; \lambda^{r+1})) - g_\gamma^\star(x^\star(\widehat{x}^r, \widehat{y}^r; \lambda^{r+1}), y^\star(\widehat{x}^r, \widehat{y}^r; \lambda^{r+1})) \geq 0$ and we apply (234a) or $g(x^\star(\widehat{x}^r, \widehat{y}^r; \lambda^{r+1}), y^\star(\widehat{x}^r, \widehat{y}^r; \lambda^{r+1})) - g_\gamma^\star(x^\star(\widehat{x}^r, \widehat{y}^r; \lambda^{r+1}), y^\star(\widehat{x}^r, \widehat{y}^r; \lambda^{r+1})) < 0$, which gives the upper bound of $\|\delta\|^2$.

Applying (212) and the same argument as (205), we can get

$$\sum_{r<t} \|g(x^r, y^r) - g_\gamma^\star(x^r, y^r) - \delta\|^2 \|\lambda^r\|^2 \leq \left(\max\{\alpha, \beta\} + \frac{1}{\tau}\right)\Lambda(Q^1 - \underline{Q} + 10\vartheta G\sqrt{2T} + 10\vartheta^2\sigma^2 T)$$

where $(a)$ we choose the same parameters as before. Therefore, we can obtain

$$\mathbb{E}\left[\frac{1}{T}\sum_{r<T}\|g(x^r, y^r) - g_\gamma^\star(x^r, y^r) - \epsilon\|^2\|\lambda^r\|^2 \Big| t = T\right] = \mathcal{O}(\epsilon^2) = \mathcal{O}\left(\frac{1}{\sqrt{T}}\right). \tag{218}$$

$\square$

# E. Additional Proofs

## E.1. Proof of Contraction of LL Sequence

**Lemma 10.** *Under A1-A5, suppose that the sequence $\{x^r, y^r, z^r, \widehat{x}^r, \widehat{y}^r, \lambda^r, \forall r\}$ is generated by TSP and $\gamma \in (0, 1/(2\rho))$. The difference between the iterates and their corresponding optimal solution satisfies the following stochastic contraction property:*

$$\|z^{r+1} - z^\star(x^r, y^r)\|^2 \leq \varrho\|z^r - z^\star(x^r, y^r)\|^2 + n_{z'}^r(\eta, \varepsilon_{g_z}^r), \tag{219}$$

$$\|z^{r+1} - z^\star(x^{r+1}, y^{r+1})\|^2 \leq \varphi\|z^r - z^\star(x^r, y^r)\|^2 + \left(2L_z^2 + \frac{\eta L_z}{2}\right)\|x^{r+1} - x^r\|^2$$

$$+ \left(2L_z^2 + \frac{\eta L_z}{2}\right)\|y^{r+1} - y^r\|^2 + n_z^r(\eta, \varepsilon_x^r, \varepsilon_y^r, \varepsilon_{g_z}^r) \tag{220}$$

*where the contraction constants and the random errors are*

$$\varrho \triangleq \left(1 - \frac{\eta}{2}\left(\frac{1}{\gamma} - \rho\right)\right) < 1, \tag{221a}$$

$$\varphi \triangleq (1 + \eta L_z)\left(1 - \frac{\eta}{2}\left(\frac{1}{\gamma} - \rho\right)\right) < 1, \tag{221b}$$

$$n_{z'}^r(\eta, \varepsilon_{g_z}^r) \triangleq \eta\langle\varepsilon_{g_z}^r, z^r - z^\star(x^r, y^r)\rangle + 2\eta^2\|\varepsilon_{g_z}^r\|^2,$$

$$n_z^r(\eta, \varepsilon_x^r, \varepsilon_y^r, \varepsilon_{g_z}^r) \triangleq \eta(1 + \eta L_z)\langle\varepsilon_{g_z}^r, z^r - z^\star(x^r, y^r)\rangle + \langle z^{r+1} - z^\star(x^r, y^r), \eta\nabla_x z^\star(\tilde{x}^r, \tilde{y}^r)\varepsilon_x^r\rangle \tag{221c}$$

$$+ \langle z^{r+1} - z^\star(x^r, y^r), \eta\nabla_y z^\star(\tilde{x}^r, \tilde{y}^r)\varepsilon_y^r\rangle + 2\eta^2(1 + \eta L_z)\|\varepsilon_{g_z}^r\|^2. \tag{221d}$$

*Proof.*

$$\|z^{r+1} - z^\star(x^r, y^r)\|^2$$
$$= \|z^{r+1} - z^r + z^r - z^\star(x^r, y^r)\|^2$$
$$= \|z^{r+1} - z^r\|^2 + 2\eta\left\langle z^{r+1} - z^r, z^r - z^\star(x^r, y^r)\right\rangle + \|z^r - z^\star(x^r, y^r)\|^2 \tag{222}$$
$$\overset{(a)}{\leq} \left(1 - \frac{\eta}{2}\left(\frac{1}{\gamma} - \rho\right)\right)\|z^r - z^\star(x^r, y^r)\|^2 + \eta\langle\varepsilon_{g_z}^r, z^r - z^\star(x^r, y^r)\rangle + 2\eta^2\|\varepsilon_{g_z}^r\|^2 \tag{223}$$

where $(a)$ is true due to the strong convexity of the loss function when $\gamma \in (0, 1/(2\rho))$.

$$\|z^{r+1} - z^\star(x^{r+1}, y^{r+1})\|^2$$
$$\leq \|z^{r+1} - z^\star(x^r, y^r) + z^\star(x^r, y^r) - z^\star(x^{r+1}, y^{r+1})\|^2 \tag{224}$$
$$\leq \|z^{r+1} - z^\star(x^r, y^r)\|^2 + \|z^\star(x^r, y^r) - z^\star(x^{r+1}, y^{r+1})\|^2 + 2\langle z^{r+1} - z^\star(x^r, y^r), z^\star(x^r, y^r) - z^\star(x^{r+1}, y^{r+1})\rangle$$
$$\leq \|z^{r+1} - z^\star(x^r, y^r)\|^2 + 2L_z^2\|y^{r+1} - y^r\|^2 + 2L_z^2\|x^{r+1} - x^r\|^2$$
$$+ 2\langle z^{r+1} - z^\star(x^r, y^r), z^\star(x^r, y^r) - z^\star(x^{r+1}, y^{r+1})\rangle. \tag{225}$$

Then, by the mean value theorem, we have that $(\tilde{x}^{r+1}, \tilde{y}^{r+1}) = (ax^{r+1} + (1-a)x^r, ay^{r+1} + (1-a)y^r)$ where $0 < a < 1$ such that

$$\langle z^{r+1} - z^\star(x^r, y^r), z^\star(x^r, y^r) - z^\star(x^{r+1}, y^{r+1})\rangle$$
$$\leq \langle z^{r+1} - z^\star(x^r, y^r), \nabla z^\star(\tilde{x}^r, \tilde{y}^r)(\tilde{x}^{r+1} - \tilde{x}^r)\rangle \tag{226}$$
$$\overset{(a)}{\leq} \eta L_z\|z^{r+1} - z^\star(x^r, y^r)\|\|(\tilde{x}^{r+1}, \tilde{y}^{r+1}) - (\tilde{x}^r, \tilde{y}^r)\| + \langle z^{r+1} - z^\star(x^r, y^r), \eta\nabla_x z^\star(\tilde{x}^r, \tilde{y}^r)\varepsilon_x^r\rangle$$
$$+ \langle z^{r+1} - z^\star(x^r, y^r), \eta\nabla_y z^\star(\tilde{x}^r, \tilde{y}^r)\varepsilon_y^r\rangle \tag{227}$$
$$\overset{(b)}{\leq} \eta L_z\|z^{r+1} - z^\star(x^r, y^r)\|(\|x^{r+1} - x^r\| + \|y^{r+1} - y^r\|) + \langle z^{r+1} - z^\star(x^r, y^r), \eta\nabla_x z^\star(\tilde{x}^r, \tilde{y}^r)\varepsilon_x^r\rangle$$
$$+ \langle z^{r+1} - z^\star(x^r, y^r), \eta\nabla_y z^\star(\tilde{x}^r, \tilde{y}^r)\varepsilon_y^r\rangle \tag{228}$$
$$\leq \eta L_z\|z^{r+1} - z^\star(x^r, y^r)\|^2 + \frac{\eta L_z}{2}(\|x^{r+1} - x^r\|^2 + \|y^{r+1} - y^r\|^2) + \langle z^{r+1} - z^\star(x^r, y^r), \eta\nabla_x z^\star(\tilde{x}^r, \tilde{y}^r)\varepsilon_x^r\rangle$$
$$+ \langle z^{r+1} - z^\star(x^r, y^r), \eta\nabla_y z^\star(\tilde{x}^r, \tilde{y}^r)\varepsilon_y^r\rangle \tag{229}$$

where $(a)$ follows from the continuity of $z^\star$, and $(b)$ holds due to the convex combination of the two points.

Combining both terms yields

$$\|z^{r+1} - z^\star(x^{r+1}, y^{r+1})\|^2$$

$$\leq (1 + \eta L_z)\|z^{r+1} - z^\star(x^r, y^r)\|^2 + \left(2L_z^2 + \frac{\eta L_z}{2}\right)\|x^{r+1} - x^r\|^2 + \left(2L_z^2 + \frac{\eta L_z}{2}\right)\|y^{r+1} - y^r\|^2$$

$$+ \langle z^{r+1} - z^\star(x^r, y^r), \eta\nabla_x z^\star(\tilde{x}^r, \tilde{y}^r)\varepsilon_x^r\rangle + \langle z^{r+1} - z^\star(x^r, y^r), \eta\nabla_y z^\star(\tilde{x}^r, \tilde{y}^r)\varepsilon_y^r\rangle \tag{230}$$

$$\overset{(223)}{\leq} (1 + \eta L_z)\left(1 - \frac{\eta}{2}\left(\frac{1}{\gamma} - \rho\right)\right)\|z^r - z^\star(x^r, y^r)\|^2 + \left(2L_z^2 + \frac{\eta L_z}{2}\right)\|x^{r+1} - x^r\|^2 + \left(2L_z^2 + \frac{\eta L_z}{2}\right)\|y^{r+1} - y^r\|^2$$

$$+ \eta(1 + \eta L_z)\langle\varepsilon_{g_z}^r, z - z^\star(x^r, y^r)\rangle + 2\eta^2(1 + \eta L_z)\|\varepsilon_{g_z}^r\|^2$$

$$+ \langle z^{r+1} - z^\star(x^r, y^r), \eta\nabla_x z^\star(\tilde{x}^r, \tilde{y}^r)\varepsilon_x^r\rangle + \langle z^{r+1} - z^\star(x^r, y^r), \eta\nabla_y z^\star(\tilde{x}^r, \tilde{y}^r)\varepsilon_y^r\rangle. \tag{231}$$

$$\square$$

## E.2. Proof of Dual Error Bound

**Lemma 11.** *Under A1-A5, suppose that $\lambda, \bar{y}^\star(\hat{x}, \hat{y})$ are bounded and $p > L$, then the dual error bound holds, namely,*

$$\|y^\star(\hat{x}, \hat{y}; \lambda_+(\hat{x}, \hat{y})) - \bar{y}^\star(\hat{x}, \hat{y})\|^2 + \|x^\star(\hat{x}, \hat{y}; \lambda_+(\hat{x}, \hat{y})) - \bar{x}^\star(\hat{x}, \hat{y})\|^2$$
$$\leq \sigma_w\|\lambda - \lambda_+(\hat{x}, \hat{y})\|\|\lambda(\hat{x}, \hat{y}) - \lambda_+(\hat{x}, \hat{y})\| \tag{232}$$

*where*

$$\sigma_w \triangleq \frac{1 + \tau(2\ell_g + \ell_\gamma)\sigma_2}{2\tau(p - L)}, \quad and \quad \sigma_2 \triangleq \frac{p + L}{p - L}. \tag{233}$$

*Proof.* First, let

$$\lambda_+(v) = \text{Proj}_{\geq 0}\left[\lambda + \tau\nabla_\lambda K(x^\star(v; \lambda), y^\star(v; \lambda), v; \lambda)\right], \tag{234a}$$

$$\lambda(v) \in \arg\max_{\lambda \geq 0} K(\bar{x}^\star(v), \bar{y}^\star(v), v; \lambda). \tag{234b}$$

Based on the strong convexity of $K(x, \cdot, \hat{x}, \hat{y}; \lambda)$ and $K(\cdot, y, \hat{x}, \hat{y}; \lambda)$, we have

$$K(\bar{x}^\star(v), \bar{y}^\star(v), v; \lambda_+(v)) - K(x^\star(v; \lambda_+(v)), \bar{y}^\star(v), v; \lambda_+(v)) \geq \frac{p - L}{2}\|x^\star(v; \lambda_+(v)) - \bar{x}^\star(v)\|^2, \tag{235a}$$

$$K(x^\star(v; \lambda_+(v)), \bar{y}^\star(v), v; \lambda_+(v)) - K(x^\star(v; \lambda_+(v)), y^\star(v; \lambda_+(v)), v; \lambda_+(v))$$
$$\geq \frac{p - L}{2}\|y^\star(v; \lambda_+(v)) - \bar{y}^\star(v)\|^2, \tag{235b}$$

$$K(x^\star(v; \lambda_+(v)), y^\star(v; \lambda_+(v)), v; \lambda(v)) - K(x^\star(v; \lambda_+(v)), \bar{y}^\star(v), v; \lambda(v)) \geq \frac{p - L}{2}\|y^\star(v; \lambda_+(v)) - \bar{y}^\star(v)\|^2, \tag{235c}$$

$$K(x^\star(v; \lambda_+(v)), \bar{y}^\star(v), v; \lambda(v)) - K(\bar{x}^\star(v), \bar{y}^\star(v), v; \lambda(v)) \geq \frac{p - L}{2}\|x^\star(v; \lambda_+(v)) - \bar{x}^\star(v)\|^2. \tag{235d}$$

Note that $\lambda_+(v)$ is the maximizer of the following problem.

$$\max_{\tilde{\lambda} \geq 0} \tau K(x^\star(v; \lambda_+(v)), y^\star(v; \lambda_+(v)), v; \tilde{\lambda}) - \tilde{\delta}^T(v; \lambda, \lambda_+(v))\tilde{\lambda} \tag{236}$$

where

$$\tilde{\delta}(v; \lambda, \lambda_+(v)) = (\lambda_+(v) + \tau\nabla_\lambda K(x^\star(v; \lambda_+(v)), y^\star(v; \lambda_+(v)), v; \lambda_+(v)))$$
$$- (\lambda + \tau\nabla_\lambda K(x^\star(v; \lambda), y^\star(v; \lambda), v; \lambda)). \tag{237}$$

According to the Lipschitz continuity of $\nabla_\lambda K$, we have

$$
\begin{aligned}
&\|\widetilde{\delta}(v; \lambda, \lambda_+(v))\| \\
&\leq \|\lambda_+(v) - \lambda\| + \tau\|g(x^\star(v; \lambda_+(v)), y^\star(v; \lambda_+(v))) - g(x^\star(v; \lambda), y^\star(v; \lambda))\| \\
&\quad + \tau\|g_\gamma^\star(x^\star(v; \lambda_+(v)), y^\star(v; \lambda_+(v))) - g_\gamma^\star(x^\star(v; \lambda), y^\star(v; \lambda_+(v)))\|
\end{aligned}
\tag{238}
$$

$$
\overset{(a)}{\leq} (1 + \tau(2\ell_g + \ell_\gamma)\sigma_2)\|\lambda - \lambda_+(v)\|
\tag{239}
$$

where in (a) we use the primal error bounds (26), (25b), and apply the Lipschitz continuity of $g(,)$ and $g_\gamma^\star(,)$.

Based on the definition of $\lambda_+(v)$ (cf. (234a)), we have

$$
\begin{aligned}
&\tau K(x^\star(v; \lambda_+(v)), y^\star(v; \lambda_+(v)), v; \lambda(v)) - \widetilde{\delta}^T(v; \lambda, \lambda_+(v))\lambda(v) \\
&\leq \tau K(x^\star(v; \lambda_+(v)), y^\star(v; \lambda_+(v)), v; \lambda_+(v)) - \widetilde{\delta}^T(v; \lambda, \lambda_+(v))\lambda_+(v).
\end{aligned}
\tag{240}
$$

Subsequently, we can obtain

$$
\begin{aligned}
&\tau K(x^\star(v; \lambda_+(v)), y^\star(v; \lambda_+(v)), v; \lambda(v)) - \tau K(x^\star(v; \lambda_+(v)), y^\star(v; \lambda_+(v)), v; \lambda_+(v)) \\
&\leq (\lambda(v) - \lambda_+(v))^T \widetilde{\delta}(v; \lambda, \lambda_+(v))
\end{aligned}
\tag{241}
$$

$$
\overset{(239)}{\leq} \|\lambda(v) - \lambda_+(v)\|(1 + \tau(2\ell_g + \ell_\gamma)\sigma_2)\|\lambda - \lambda_+(v)\|.
\tag{242}
$$

According to the definition of $\lambda(v)$ (cf. (234b)), we have

$$
K(\bar{x}^\star(v), \bar{y}^\star(v), v; \lambda(v)) \geq K(\bar{x}^\star(v), \bar{y}^\star(v), v; \lambda_+(v)).
\tag{243}
$$

Combing (235a) to (235d), and (243) yields

$$
\begin{aligned}
&2\tau(p - L)\left(\|y^\star(v; \lambda_+(v)) - \bar{y}^\star(v)\|^2 + \|x^\star(v; \lambda_+(v)) - \bar{x}^\star(v)\|^2\right) \\
&\leq \|\lambda(v) - \lambda_+(v)\|(1 + \tau(2\ell_g + \ell_\gamma)\sigma_2)\|\lambda - \lambda_+(v)\|.
\end{aligned}
\tag{244}
$$

Therefore, we have

$$
\begin{aligned}
&\|y^\star(\widehat{x}^r, \widehat{y}^r; \lambda_+(\widehat{x}^r, \widehat{y}^r)) - \bar{y}^\star(\widehat{x}^r, \widehat{y}^r)\|^2 + \|x^\star(\widehat{x}^r, \widehat{y}^{r+1}; \lambda_+(\widehat{x}^r, \widehat{y}^r)) - \bar{x}^\star(\widehat{x}^r, \widehat{y}^r)\|^2 \\
&\leq \sigma_{\mathrm{w}}\|\lambda^{r+1} - \lambda_+(\widehat{x}^r, \widehat{y}^r)\|\|\lambda(\widehat{x}^r, \widehat{y}^r) - \lambda_+(\widehat{x}^r, \widehat{y}^r)\|
\end{aligned}
\tag{245}
$$

where $\sigma_{\mathrm{w}} \triangleq 1 + \tau(2\ell_g + \ell_\gamma)\sigma_2/(2\tau(p - L))$. $\qquad\square$

# F. More Discussion on KKT Solutions and Additional Numerical Results

## F.1. Interpretation of the Solution

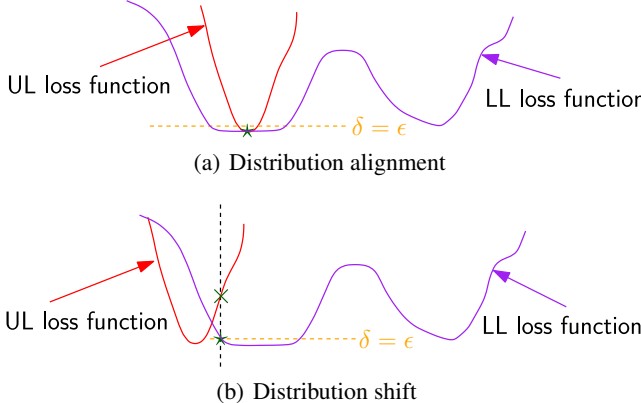

(a) Distribution alignment

(b) Distribution shift

*Figure 3.* Illustration of the Importance of Finding the KKT Solutions.

We consider a one-dimensional simple bilevel problem, formulated as $\min_x \ell_{\text{UL}}(x)$ subject to $x \in \arg\min_{x'} \ell_{\text{LL}}(x')$, as illustrated in Figure 5, to emphasize the importance of achieving KKT solutions, particularly concerning the slackness condition. Due to the nonconvexity of the LL problem, there may exist multiple stationary points. Assuming that $x$ lies within the optimal set of the LL solution, it then further needs to find the stationary (or optimal) point of the UL loss function. When there is overlap between the two loss functions, as depicted in Figure 5(a), the optimal point of the UL loss function coincides with the best solution. However, if there is a shift between the UL and LL optimal sets, the slackness condition ensures that the solution must be attained at the boundary. In contrast, penalty-based methods only guarantee convergence to some stationary points, without ensuring this level of optimality. As illustrated in Figure 5(a), the obtained solution corresponds to the minimum point of the UL given that $x$ belongs to the LL optimal set at least in this case. This underscores the importance of finding KKT points rather than stationary points in terms of generalization performance.

## F.2. Additional Numerical Results

The algorithms are further tested on representation learning for a multi-task sinusoid regression problem (Rajeswaran et al., 2019), where the UL loss is $K^{-1}\sum_{k=1}^{K}\ell(x,y_k,\mathcal{D}_{\text{val}})$ and the lower-level loss is $\ell(x,y_k,\mathcal{D}_{\text{tr}}), \forall k$. Here, $K = 10$ denotes the number of tasks, $\ell()$ is the mean square loss, $\mathcal{D}_{\text{val}}$ denotes the validation data samples (with amplitudes varying within $[0.1, 5.0]$, phase varying within $[0, \pi]$, and frequencies varying within $[0.1, 3]$), and $\mathcal{D}_{\text{tr}}$ denotes the training data samples. In the numerical experiments, the selected learning rates for all algorithms are $0.01$ for $x$, $0.05$ for $y$, and $0.06$ for $z$. The dual variable learning rate for TSP is $0.5$, selected from $\{1, 0.5, 0.1, 0.01\}$. The neural network includes 2 hidden layers of

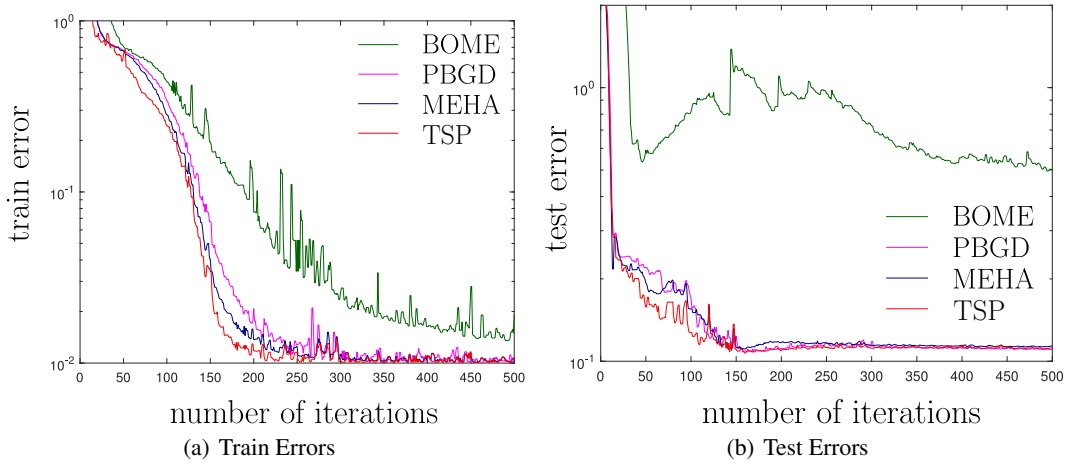

(a) Train Errors

(b) Test Errors

*Figure 4.* Training and Test Errors vs. Number of Iterations.

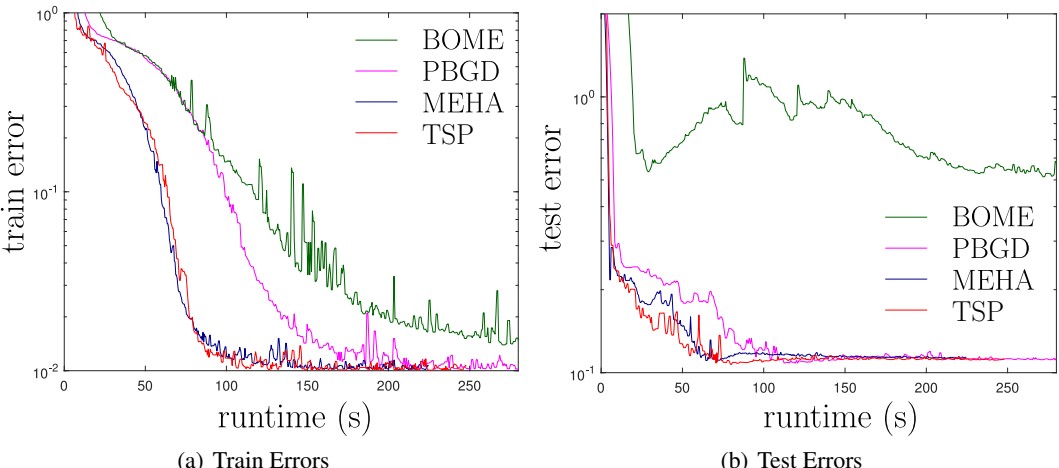

(a) Train Errors

(b) Test Errors

*Figure 5.* Training and Test Errors vs. Computational Time.

size 40 with ReLU nonlinearities (the weights are represented by $x$) and $K$ perception layers as heads for each task (each represented by $y_k$). There are a total of 10 training tasks, each with 10 data samples, and 5 test tasks. The training and test errors are averaged over these tasks. The experiments are conducted over 10 independent trials.

In Figure 5(a), it can be seen that the proposed TSP and MEHA, being single-loop algorithms, exhibit a faster convergence rate in terms of runtime compared to the double-loop algorithms BOME and PBGD. In Figure 5(b) TSP demonstrates a lower test error compared to the other algorithms. Specifically, TSP is designed to find KKT points rather than stationary points. The figure shows that the solutions achieved by TSP provide lower test errors than those obtained by the other algorithms.

