# OpenReview forum: "TSP: A Two-Sided Smoothed Primal-Dual Method for Nonconvex Bilevel Optimization"
_ICML.cc/2025/Conference — ICML 2025 poster_

### Official Review · Reviewer_fDFn · 2025-02-24

**Overall Recommendation:** 3

**Summary:**

This paper investigates a bilevel optimization problem where both the upper and lower levels are nonconvex, making it a challenging problem. The author proposes a smoothed-type single-loop algorithm and provides a theoretical complexity guarantee for convergence to a KKT-type stationary point. Numerical experiments are conducted to demonstrate the performance of the proposed algorithm.

## update after rebuttal: I appreciate the authors’ response. I agree that the existing hardness results, while closely related, do not contradict the findings presented in the paper. If the results are indeed correct, this would be a strong and impactful contribution.

**Claims And Evidence:**

Yes, it is clear.

**Essential References Not Discussed:**

Not found.

**Experimental Designs Or Analyses:**

The experimental design makes sense to me. However, it would be helpful if the authors could explain how the chosen parameters in the experiments align with the theoretical claims. Providing this clarification would strengthen the connection between the theoretical guarantees and empirical results.

**Methods And Evaluation Criteria:**

Yes, the algorithm is tested on standard machine learning tasks.

**Other Comments Or Suggestions:**

- line 195: formulations (1) and (1)
- line 215: forgotten period at the end.
- line 255 & 256: should the UL and LL get exchanged?

**Other Strengths And Weaknesses:**

If the results are correct, the problem studied in this paper could be highly significant.

**Questions For Authors:**

Please review the theoretical sections. I will definitely increase my score if I can confirm the correctness of the paper.

**Relation To Broader Scientific Literature:**

This work makes significant progress in bilevel optimization by removing the convexity assumption, which is a notable advancement in the literature.

**Theoretical Claims:**

I find the key contribution of this paper to be its ability to handle nonconvexity in the lower-level problem, which, to my knowledge, has been a significant challenge in bilevel optimization. As claimed by the authors, the proposed algorithm reaches a stationary point satisfying the KKT conditions for both the upper and lower levels under only a weakly convex assumption. This result appears quite strong. Given that lower bound results exist for related problems (such as in minimax optimization, which is a special case considered in this paper), [1] shows that such problems are PPAD-hard (though with additional constraints). Could the authors provide insights into why their approach successfully achieves this result despite these known hardness barriers? Understanding this aspect would further clarify the theoretical significance of the proposed method.

[1] Daskalakis, Constantinos, Stratis Skoulakis, and Manolis Zampetakis. "The complexity of constrained min-max optimization." Proceedings of the 53rd Annual ACM SIGACT Symposium on Theory of Computing. 2021.

---

> ### Author Rebuttal · Authors · 2025-04-01
>
> We thank reviewer fDFn for your helpful comments and questions.
>
> **Theoretical Claims:**
> *I find the key contribution of this paper to be its ability to handle nonconvexity in the lower-level problem, which, to my knowledge, has been a significant challenge in bilevel optimization. As claimed by the authors, the proposed algorithm reaches a stationary point satisfying the KKT conditions for both the upper and lower levels under only a weakly convex assumption. This result appears quite strong. Given that lower bound results exist for related problems (such as in minimax optimization, which is a special case considered in this paper), [1] shows that such problems are PPAD-hard (though with additional constraints). Could the authors provide insights into why their approach successfully achieves this result despite these known hardness barriers? Understanding this aspect would further clarify the theoretical significance of the proposed method.*
>
> > Thank you for your insightful comment. The main source of hardness in solving min-max problems (a special case of bilevel optimization) to the type of stationary points defined in [1] lies in the presence of constraints. In our setting, there are **no explicit constraints** at either the upper or lower level.
>
> > Additionally, we assume that the objective functions are **coercive**, which helps ensure that the iterates generated by our SPD algorithm remain within a bounded region without requiring additional projection. As a result, this assumption—together with the algorithm design—implicitly guarantees the boundedness of the loss values over the unconstrained domain.
>
> > These conditions align exactly with the discussion following Theorem 4.1 in [1], which argues that in such unconstrained settings with bounded loss value, approximate stationary points **do** exist and can be found in **polynomial time**. Therefore, our theoretical results do not contradict known hardness barriers; rather, they fall within a subclass of problems where efficient solutions remain tractable.
>
>
>
>
> *[1] Daskalakis, Constantinos, Stratis Skoulakis, and Manolis Zampetakis. "The complexity of constrained min-max optimization." Proceedings of the 53rd Annual ACM SIGACT Symposium on Theory of Computing. 2021.*
>
>
> **Experimental Designs Or Analyses:**
> *The experimental design makes sense to me. However, it would be helpful if the authors could explain how the chosen parameters in the experiments align with the theoretical claims. Providing this clarification would strengthen the connection between the theoretical guarantees and empirical results.*
>
> >Thank you for the helpful comment. The parameter choices (primarily the learning rates) were selected via grid search, following standard practice in the literature. For the stochastic case, we applied a learning rate decay of $ 1/\sqrt{r}$, which is guided by our theoretical results. The initial learning rates were determined through grid search.
>
> > We also apologize for the typo in line 418 of the paper: the learning rate decay was incorrectly written as $1/r$; it should be $1/\sqrt{r}$. We will correct this in the revised version.
>
>
> **Other Comments Or Suggestions:**
>
> *line 195: formulations (1) and (1)*
>
> > Typo: the second reference to equation (1) should be (3).
> >
> *line 215: forgotten period at the end.*
>
> > Thank you for your careful reading. We will add the missing period in the revised version.
>
> *line 255 & 256: should the UL and LL get exchanged?*
>
> > The terms "UL'' and "LL'' are indeed confusing in this context. We will remove them in the revised version.

---

### Official Review · Reviewer_bXuu · 2025-03-11

**Overall Recommendation:** 3

**Summary:**

This paper proposed a single-loop method for solving the stochastic bilevel optimization problem with weakly convex lower-level problem. The proposed method is proved to achieve a convergence rate of $O(\epsilon^{-4})$ in terms of a smoothed reformulation. Some experimental results on data hyper-cleaning task and representation learning task are presented to show that the proposed method outperfumes some existing methods.

**Claims And Evidence:**

This paper claims that the proposed method better solves the stochastic bilevel optimization problem in terms of both convergence rate and experimental performance. For the theoretical analysis part, my main concern is the gap between the penalty reformulation and the original bilevel problem. It seems that both of the proposed method and the convergence analysis are based on the reformulation. It is unclear that how the convergence guarantee relates to the original problems.

**Essential References Not Discussed:**

All essential references that I'm aware of are discussed.

**Experimental Designs Or Analyses:**

Regarding the baselines in the experiments, I have the following concerns/questions.

1. Regarding the data hyper-cleaning task, as described in section 4, the lower-level objective is the cross-entropy loss function, which is convex. This problem can be solved by methods designed for bilevel problems with convex lower-level problems. This application seems not suitable. The authors need to either compare SPD with the convex LL methods or choose a 'merely' weakly convex LL problem as the application.
2. All the baselines compared in the experiments are in deterministic setting, which makes the experimental results not very convincing. Moreover, it is well known that stochastic methods generalize better than deterministic methods in general. Thus, the claim 'This example further highlights the advantage of solving bilevel learning problem through the lens of equilibrium constrained optimization' in the representation learning task part may not necessarily hold.

**Methods And Evaluation Criteria:**

The main idea of the method design is to reformulate the original bilevel optimization problem into a min-max optimization problem with more feasible conditions using moreau envelope and penalty technique. This makes sense as the reformulated problem is a convex-linear min-max problem is well-studied and easier to solve.

**Other Comments Or Suggestions:**

1. In Assumption 3.1, the weak-convexity assumption A2 is unnecessary as it is implied by the smoothness assumption A1.
2. Using $G(x,y;\xi_i)$ for single sample stochastic estimator, and $\hat{g}(x,y)$ for mini batch stochastic estimator is rather confusing. It would help the readers to understand more easily if such notations are more consistent.

**Other Strengths And Weaknesses:**

I do not see other strengths and weaknesses.

**Questions For Authors:**

I do not have any other questions.

**Relation To Broader Scientific Literature:**

The main contribution of this work is that it is trying to weaken the convexity assumption on the lower-level objective of bilevel optimization, which is essential as it covers a larger family of problems.

**Theoretical Claims:**

Please see the Claims And Evidence section.

---

> ### Author Rebuttal · Authors · 2025-04-01
>
> We thank reviewer bXuu for your helpful comments and questions.
>
> **Claims And Evidence:**
> *My main concern is the gap between the penalty reformulation and the original bilevel problem. It seems that both of the proposed method and the convergence analysis are based on the reformulation. *
>
> > Correct. The proposed method addresses a Moreau envelope-based reformulation of the original bilevel optimization problem. The relationship and equivalence between the original and reformulated problems are discussed between lines 194 and 199 of the paper, summarizing the findings of existing works [Gao et al., 2023] and [Liu et al., 2024a].
> >
> > The equivalence presented in [Theorem A.2, Liu et al., 2024a] for the simplified case where the lower-level problem is unconstrained is further clarified below.
>
>
> >
> > **Original Problem:**
> > $$
> > \min_{x, y} f(x, y), \quad \text{s.t.} \quad y \in \mathcal{S}(x)
> > $$
> > where $\mathcal{S}(x) := \arg\min_y g(x, y)$.
> >
> > **Moreau Envelope-based Reformulation:**
> > $$
> > \min_{x, y} f(x, y), \quad \text{s.t.} \quad g(x, y) - g^\star_\gamma(x) \le 0
> > $$
> >
> > - When the lower-level function $g(x, y)$ is convex or satisfies the Polyak-Łojasiewicz (PL) condition, these two formulations are equivalent.
> > - When $g(x, y)$ is weakly convex and $\gamma \in (1, 1/\rho)$, the Moreau envelope-based reformulation becomes equivalent to a relaxed version of the original problem:
> > $$
> > \min_{x, y} f(x, y), \quad \text{s.t.} \quad y \in \mathcal{S}'(x)
> > $$
> > where $\mathcal{S}'(x) := \\{ y \mid \\|\nabla_y g(x, y)\\| = 0 \\}$
>
> > Given the weak convexity of $g$, it is reasonable to aim for a stationary point, as opposed to requiring a global optimum for the lower-level problem.
>
>
> **Methods And Evaluation Criteria:**
> The main idea of the method design is to reformulate the original bilevel optimization problem into a min-max optimization problem with more feasible conditions using moreau envelope and penalty technique. This makes sense as the reformulated problem is a convex-linear min-max problem is well-studied and easier to solve.
>
> > Yes, this approach is conceptually related to the primal-dual or Lagrangian methods. However, the reformulated problem is **not** a convex-linear min-max problem due to the weak convexity of both the upper- and lower-level objective functions. Therefore, standard techniques for solving convex-linear min-max problems are not directly applicable in this case.
> > The main reasons are as follows:
> > 1. The upper-level objective function $f(x, y)$ are nonconvex rather than convex.
> > 2. The lower-level function $g(x, y)$, even after applying the Moreau envelope, does not necessarily yield a convex constraint.
>
>
> **Experimental Designs Or Analyses:**
> *Regarding the data hyper-cleaning task, as described in section 4, the lower-level objective is the cross-entropy loss function, which is convex. This problem can be solved by methods designed for bilevel problems with convex lower-level problems. This application seems not suitable. *
>
> >  As noted on line 382, the lower-level objective includes a nonconvex regularization term. While the cross-entropy loss $\ell_{\text{tr}}$ is convex, the presence of the nonconvex regularizer makes the overall lower-level function **weakly convex**. Therefore, this data hyper-cleaning task remains a suitable example for evaluating methods designed for bilevel problems with weakly convex lower-level objectives.
>
>
> *All the baselines compared in the experiments are in deterministic setting, which makes the experimental results not very convincing. Moreover, it is well known that stochastic methods generalize better than deterministic methods in general. Thus, the claim 'This example further highlights ... optimization' in the representation learning task part may not necessarily hold.*
>
> > Please kindly note that the numerical experiments on the representation learning task are conducted in a **stochastic setting**. As mentioned on line 418, a batch size of 32 is used. This setting is applied to all compared algorithms, indicating that they are all implemented stochastically. We will clarify this more explicitly in the revised version to ensure it is clear that **all methods are evaluated in a stochastic manner** for a fair comparison.
>
>
> **Other Comments Or Suggestions:**
>
> *In Assumption 3.1, the weak-convexity assumption A2 is unnecessary as it is implied by the smoothness assumption A1.*
>
> > Thank you for your comment. We will remove Assumption A2 in the revised version.
>
>
> *Using $G(x,y;\xi_i)$ for single sample stochastic estimator, and $\hat{g}(x,y)$ for mini batch stochastic estimator is rather confusing. It would help the readers to understand more easily if such notations are more consistent.*
>
> > Thank you for the helpful suggestion. We will use $\nabla \hat{G}(x, y)$ to denote the mini-batch stochastic gradient estimator, so that the notation is more consistent throughout the paper.

---

> > ### Comment · Reviewer_bXuu · 2025-04-02
> >
> > Thank you for the detailed rebuttal. My concern on the gap between the penalty reformulation and the original bilevel problem is resolved. Their connections seem to be well-studied in existing works as the authors explained. The baselines in the experiments make sense now as they are implemented in stochastic manner, which makes a fair comparison.
> >
> > Regarding my other concerns:
> >
> > >Yes, this approach is conceptually related to the primal-dual or Lagrangian methods. However, the reformulated problem is not a convex-linear min-max problem due to the weak convexity of both the upper- and lower-level objective functions.
> >
> > You are right. I meant to say that the reformulated problem is a weakly-convex-linear min-max problem. But still, it is a well-studied problem, which makes the contribution of the analysis less strong.
> >
> > >As noted on line 382, the lower-level objective includes a nonconvex regularization term. While the cross-entropy loss
> >  is convex, the presence of the nonconvex regularizer makes the overall lower-level function weakly convex.
> >
> > This is still confusing to me. The lower-level problem in the data hyper-cleaning task is $\ell_{tr}(x,y')+\bar{\rho} \ \text{reg}(x)$, which is essentially just the cross-entropy loss in terms of $y'$, thus is convex in $y'$. The 'nonconvex regularization term' is independent from $y'$. What is the intuition of adding this regularization term? Is this a standard technique in data hyper-cleaning task?

---

> > > ### Author Response · Authors · 2025-04-03
> > >
> > > Dear Reviewer bXuu,
> > >
> > > We’re glad to hear that our response addressed your concerns regarding the gap between the penalty reformulation and the original bilevel problem, as well as the setup of our numerical experiments.
> > >
> > > We also appreciate your follow-up questions. Our detailed responses are provided below.
> > >
> > > >  I meant to say that the reformulated problem is a weakly-convex-linear min-max problem. But still, it is a well-studied problem, which makes the contribution of the analysis less strong.
> > >
> > > **Response**
> > >
> > > - (**Inequivalence between min-max problem and primal-dual problem**)
> > >   The key difference between a min-max problem and a primal-dual formulation lies in the boundedness of the Lagrange multiplier (i.e., the maximizer).
> > >
> > >   Please note that the reformulated problem takes the form (see Equations (4) to (6) for details):
> > >   $$
> > >   \min_{x,y} \max_{\lambda \ge 0} f(x,y) + \lambda \big(g(x,y) - g^{\star}_{\gamma}(x,y) - \delta\big)
> > >   $$
> > >   where the feasible set of the dual variable (maximizer) is **unbounded**.
> > >
> > >
> > > - (**Insufficiency of using general min-max solvers**)
> > >   While the problem is indeed weakly-convex-linear in min-max form, it involves an **unbounded** dual variable $\lambda$. Existing solvers for weakly-convex-linear min-max problems typically assume that the maximizer lies in a **compact** set (this assumption that does *not* hold in our case).
> > >
> > > - (**Pitfalls of applying general min-max solvers blindly**)
> > >   The reformulated min-max structure arises from a constrained optimization setting. The existence of KKT points depends on the structure of the constraint term
> > >   $g(x,y) - g^{\star}_{\gamma}(x,y) - \delta$.
> > >   If one were to apply a generic min-max solver without accounting for this structure, it would imply that any weakly-convex constrained optimization problem (where both objective and constraints are weakly convex) could be solved without additional regularity conditions — an implication that is clearly *incorrect*.
> > >
> > > - (**Significance of our method**)
> > >   Our work specifically addresses this class of min-max problems and establishes convergence of the iterates generated by SPD to approximate KKT points, satisfying stationarity, feasibility, and complementary slackness. We also prove that the dual variable (i.e., the maximizer $\lambda$) remains **bounded** under the SPD framework, which holds for the class of bilevel optimization-oriented constraints we consider.
> > >
> > > > The lower-level problem in the data hyper-cleaning task is
> > > $\ell_{tr}(x,y')+\bar{\rho}\textrm{reg}(x)$, which is essentially just the cross-entropy loss in terms of $y'$, thus is convex in $y'$. The 'nonconvex regularization term' is independent from $y'$. What is the intuition of adding this regularization term? Is this a standard technique in data hyper-cleaning task?
> > >
> > > **Response:**
> > >
> > > - Apologies for the typo.
> > >
> > >   - The regularization term should be
> > >   $$
> > >   \sum_{i=1}^d \frac{y'^2_i}{1 + y'^2_i}
> > >   $$
> > >   where $d$ is the dimensionality of the lower-level variable $y'$.
> > >
> > > - (**Intuition of Adding This Term**) This type of nonconvex regularization encourages sparsity, similar to $\ell_1$ regularization, but with a key difference: it doesn’t overly penalize large coefficients.
> > >   - For small values of $y'_i$,
> > >     $\frac{y'^2_i}{1 + y'^2_i} \approx y'^2_i$ — like $\ell_2$ regularization.
> > >   - For large values of $y'_i$,
> > >     $\frac{y'^2_i}{1 + y'^2_i} \approx 1$ — so the penalty saturates.
> > > This makes it a balanced choice for inducing sparsity while retaining significant features.
> > >
> > > - (**Standard Technique**) This class of penalty functions is commonly used in sparse modeling, feature selection, and robust learning—especially in neural networks. In the context of data hyper-cleaning, the lower-level variables correspond to neural network weights. Using this kind of regularization helps improve robustness by selectively suppressing unreliable or noisy components without removing important ones entirely.
> > >
> > > We sincerely thank the reviewer for these insightful questions. We will incorporate this discussion into the revised version to make the statements clearer and the contributions stronger.

---

### Official Review · Reviewer_85Pr · 2025-03-13

**Overall Recommendation:** 4

**Summary:**

This paper presents a smoothed primal-dual algorithm for solving stochastic bilevel optimization problems where the lower level problem is possibly nonconvex.
The authors first use Moreau envelope reformulation for the lower level problem and then use the smoothed primal-dual method to solve the resulting constrained optimization problem.
They establish the optimal convergence rate of the algorithm.

**Claims And Evidence:**

Yes, no major issue.

**Essential References Not Discussed:**

No.

**Experimental Designs Or Analyses:**

I briefly checked them and did not find any major issues.

**Methods And Evaluation Criteria:**

Yes

**Other Comments Or Suggestions:**

See questions below.

**Other Strengths And Weaknesses:**

Strength:
1.The algorithm can solve bilevel optimization with nonconvex lower level problem and achieve the optimal rate for stochastic problem with mild assumptions.
2. For the convergence analysis of primal-dual algorithm, they prove the boundedness of the dual variable instead of making bounded dual assumption.
Weakness:
See the questions below.

**Questions For Authors:**

Questions:
1. Is $p$ changing for different $r$? Could the authors specify the value of $p$ we should take in the algorithm and main theorem?
2. How does the algorithm in this paper compare to ``SLM: A smoothed Lagrangian method for structured nonconvex constrained optimization''? Could the authors provide details?
3. The Moreau envelope reformulation is proposed in previous papers.  Could the authors provide more details about the novelty compared to these papers?

**Relation To Broader Scientific Literature:**

Nonconvex bilevel optimization is important in machine learning, AI, engineering and economics.

**Theoretical Claims:**

I checked the proof sketch and did not find major issues.

---

> ### Author Rebuttal · Authors · 2025-04-01
>
> We thank Reviewer 85Pr for your positive feedback, thoughtful comments, and constructive questions.
>
> Questions:
> *Is $p$ changing for different? Could the authors specify the value of $p$ we should take in the algorithm and main theorem?*
>
> > In theory, $p$ is a constant and should be chosen to be on the same order as the dual variable, which is upper bounded by a constant. In practice, we can initialize it as a small value and increase it if the algorithm does not sufficiently decrease the loss, following a similar philosophy to tuning the learning rate.
>
>
> *How does the algorithm in this paper compare to ``SLM: A smoothed Lagrangian method for structured nonconvex constrained optimization''? Could the authors provide details?*
>
> > In comparison to the referenced work, the main differences are as follows:
>
> > - (**Different class of lower-level problems**) The algorithms are designed for different bilevel optimization formulations. Our work focuses on a Moreau envelope-based reformulation, where the lower-level problem is weakly convex. In contrast, the referenced work assumes that the lower-level objective satisfies the PL condition.
>
>
> > - (**Stochastic vs. deterministic setting**) Our algorithm is designed for the stochastic setting, whereas the referenced method addresses only the deterministic case. This distinction introduces significant challenges, particularly in designing the update rule for the dual variable under stochastic noise.
>
> > - (**Single-loop vs. double-loop structure**) Our algorithm adopts a single-loop structure, which is more suitable for stochastic optimization. The referenced method, on the other hand, uses a double-loop approach.
>
> > - (**New high-probability error bounds**): Our theoretical analysis explicitly quantifies stochastic errors arising from the updates of the upper-level variable, lower-level variable, and auxiliary variables. This includes handling the non-negativity constraint on the dual variable. As a result, the analysis framework is significantly different from that of the referenced work.
>
> *The Moreau envelope reformulation is proposed in previous papers. Could the authors provide more details about the novelty compared to these papers?*
>
> > The main novelties of this work, compared to previous papers that also study the Moreau envelope reformulation, are as follows:
>
> > -(**Primal-Dual Update Strategy**): To the best of our knowledge, this is the first work that applies a primal-dual algorithm to solve the Moreau envelope-based bilevel optimization problem. It is well-established in the optimization community that primal-dual methods often achieve better convergence properties than penalty-based methods. This is largely due to the dynamic update of the dual variable, whereas penalty methods require either a sufficiently large penalty parameter or a monotonically increasing schedule, which can force the learning rate to be very small. This would be particularly problematic when handling multiple lower-level problems.
>
> > -(**Stochastic Problem Formulation and Algorithm Design**) The proposed SPD method tackles a stochastic version of the Moreau envelope reformulation, which significantly broadens its applicability to real-world machine learning problems. In contrast, prior works have largely focused on deterministic settings. The stochastic nature of our formulation introduces substantial challenges in both algorithm design and theoretical analysis, particularly in achieving convergence guarantees. This further differentiates our approach from existing literature.

---

### Official Review · Reviewer_Dp7u · 2025-03-13

**Overall Recommendation:** 3

**Summary:**

This paper introduces SPD (Smoothed Primal-Dual), a first-order gradient-based primal-dual method for solving bilevel optimization problems, potentially with a nonconvex lower-level problem. SPD is based on a Moreau envelope-based reformulation of the bilevel problem and employs a proximal primal-dual Lagrangian updating framework, eliminating the need for Hessian computations. Theoretical results on iteration complexity are provided, and numerical experiments are conducted to demonstrate the efficiency of the proposed method.

**Claims And Evidence:**

This paper contains several inappropriate and incorrect claims:

At the beginning of the introduction, the authors state, "The mathematical programs with equilibrium constraints (Luo et al., 1996), also known as the bilevel optimization problem." This statement is incorrect, as mathematical programs with equilibrium constraints and bilevel optimization problems are distinct classes of problems.

On lines 69-71, the authors claim, "This property is advantageous from an optimization perspective, as the Slater condition holds automatically." This is incorrect because the Slater condition applies only to convex optimization problems, whereas (2) is clearly a nonconvex optimization problem.

In the contributions section (lines 167-169), the authors state that "this is the first time a stochastic first-order method has successfully achieved the KKT points of the bilevel optimization problem." However, in this work, they only demonstrate that their proposed method achieves an approximate KKT point, making this claim misleading.

Additionally, the reformulation (3) is incorrect.

**Essential References Not Discussed:**

no

**Experimental Designs Or Analyses:**

The representation learning task in this paper lacks a clear problem setting and sufficient discussion. While the authors mention the use of a multi-head neural network structure, where the upper-level problem optimizes a shared model parameter layer and the lower-level problem consists of multiple task-specific heads, the details of how the bilevel structure is formulated and applied remain vague. The dataset split and the specific role of SPD in optimizing meta-learning objectives are not thoroughly explained. Furthermore, the experimental discussion is limited, with only a comparison of test accuracy and generalization performance across different methods.

**Methods And Evaluation Criteria:**

The derivation of the proposed SPD algorithm in Section 2 lacks clarity. The authors claim that it is based on stochastic gradient-based updates to find the equilibrium points of $$\min_{x,y,\hat{x},\hat{y}}\max_{\lambda\geq 0}K(x,y,\hat{x},\hat{y},\lambda).$$ However, by the definition of  $K(x,y,\hat{x},\hat{y},\lambda),$ the equilibrium points of this formulation must satisfy $\hat{x}=x, \hat{y}=y$. This raises concerns about the necessity of introducing the additional variables $\hat{x},\hat{y}$, as their inclusion may not contribute meaningfully to the optimization process. A clearer explanation should be provided to justify their role and the necessity of their updates.

**Other Comments Or Suggestions:**

In line 193, on the right hand side,  "$\lambda^r+,\text{Proj}\geq 0$" should be $\lambda^r_{+},\text{Proj}_{\geq 0}$; in line 291,  “beta” should be "$\beta$"; in line 685, "w.r.t. $x,y$” should be "w.r.t. $y$".

**Other Strengths And Weaknesses:**

Strength:

the experimental results demonstrate the practical advantages of the proposed method, showing improvements in test accuracy and generalization performance over other methods

Weaknesses:
1. The notation is quite confusing. For example, the variable **h** in the dual update is not properly explained, and the same symbol is used in the primal update to represent a different concept, which can cause confusion and make it difficult to follow the derivations.
2. There are an excessive number of equations, with a total of 244 equation labels. This creates a cluttered presentation and detracts from the clarity of the paper. It would be helpful to streamline the number of equations and provide better organization or referencing to make the paper more readable.

**Questions For Authors:**

no

**Relation To Broader Scientific Literature:**

The proposed SPD is designed based on a Moreau envelope-based reformulation of the bilevel optimization problem (Gao et al., 2023; Yao
et al., 2024b; Liu et al., 2024a) and a moving average technique (Chen et al., 2023b).

**Theoretical Claims:**

I have checked parts of the proof, and they appear to be sound.

---

> ### Author Rebuttal · Authors · 2025-04-01
>
> We thank reviewer Dp7u for your helpful comments and questions.
>
> **Claims And Evidence:**
> *This statement about equilibrium constraints and bilevel optimization*
>
> > Will remove that statement.
>
> *On lines 69-71*
>
> > Will remove the statement.
>
> *contributions section (lines 167-169)*
>
> > Will add  "approximate''.
>
> *Additionally, the reformulation (3) is incorrect.*
>
> > The reformulation (3) should be:
> $$
> \min_{x,y}   f(x,y):=\mathbb{E}_{\xi\sim D\_{UL}} F(x,y;\xi)
> \\\\
> \\quad \textrm{s.t.} g(x,y)- g^{\\star}\_{\gamma} (x,y)\le 0, \quad g^{\\star}\_{\gamma} (x,y):=\min_z \mathbb{E}\_{\xi\sim D\_{LL}}   G(x,z;\xi)+\frac{1}{2\gamma}\\|z-y\\|^2.
> $$
>
> **Methods And Evaluation Criteria:**
>
> Response:
> > - Once the two quadratic proximal terms, $\frac{p}{2}\\|x - \hat{x}\\|^2$ and $\frac{p}{2}\\|y - \hat{y}\\|^2$, are introduced with auxiliary variables $\hat{x}$ and $\hat{y}$, the function $K(x, y, \hat{x}, \hat{y}, \lambda)$ becomes strongly convex. This strong convexity ensures the existence of a unique optimal solution for $x$, $y$, and $\lambda$, given any fixed $\hat{x}$ and $\hat{y}$. These optimal solutions are denoted as $x^{\star}(\hat{x}, \hat{y}; \lambda)$ or $\bar{x}^{\star}(\hat{x}, \hat{y})$, and similarly for $y$ (see equations (14) and (15) for details).
>
> > - Based on this setup, we can quantify the convergence process of the iterates $(x^r, y^r, \lambda^r)$ by measuring the distance between $x^r$ and $x^{\star}(\hat{x}^r, \hat{y}^r; \lambda^{r+1})$, and similarly between $y^r$ and $y^{\star}(\hat{x}^r, \hat{y}^r; \lambda^{r+1})$. These distances can be further upper bounded by the successive differences of the iterates, i.e., $\\|x^{r+1} - x^r\\|^2$ and $\\|y^{r+1} - y^r\\|^2$, using the standard primal error bound (Zhang & Luo, 2020).
>
> > - Therefore, the proximal terms are critical. In particular, the distances to the proximal mappings automatically shrink to zero, ensuring that the algorithm converges to the solution of the optimization problem. As such, the auxiliary variables $\hat{x}$ and $\hat{y}$ play a key role in the algorithm design and in quantifying its convergence to the KKT points of the original problem.
>
> **Experimental Designs Or Analyses:**
> *The representation learning task in this paper lacks a clear problem ... the details of how the bilevel structure is formulated and applied remain vague. *
>
> > The representation learning task can be **formulated as a bilevel optimization problem**, as shown below:
> $$
> \min_{x,\{y_i\}} \quad  f(x, \{y_i\}) := \mathbb{E}\_{\xi \sim \mathcal{D}^{\text{val}}} \left[\frac{1}{K} \sum_{i=1}^K \ell(x, y_i; \xi)\right]
> \\\\
> \text{s.t.} \quad   y_i \in \arg\min_{y'_i} \mathbb{E}\_{\xi_i \sim D^{\text{tr}}_i} \ell(x, y'_i; \xi_i), \quad \text{for } i \in [K].
> $$
>
> > - $x$ represents the **shared model parameters**, typically corresponding to the **common feature encoder** or backbone network shared across all tasks.
> > - $y_i$ denotes the **task-specific head parameters**, i.e., the final classification layer for task $i$, which is optimized using the task-specific training data $\mathcal{D}^{\text{tr}}_i$.
>
> > - This formulation captures the core idea of our approach: learning a shared feature representation ($x$) that enables effective adaptation to multiple downstream tasks through task-specific parameters ($y\_i$).
>
>
> *The dataset split and the specific role of SPD in optimizing meta-learning objectives are not thoroughly explained.*
>
> > We use a custom split of the MNIST dataset, where we create eight sub-datasets, each corresponding to a single digit class. Each sub-dataset contains 2,500 training samples and 1,500 validation samples.
> > - We use five sub-datasets for pretraining the model.
> > - The remaining three sub-datasets are used for meta-learning, which includes both meta-training and meta-testing.
>
> > The key role of SPD in this framework is to adjust the dual variables individually for each task-specific head. This flexibility helps to balance the optimization dynamics between the shared and task-specific components, which in turn improves the generalization capability of the learned representation across unseen tasks.
>
> *Furthermore, the experimental discussion is limited.*
>
> > For bilevel optimization problems, our experimental discussion is centered on evaluating the performance of the algorithms at both the upper and lower levels. We also include addtional results w.r.t training accuracy, convergence behavior, and runtime efficiency in the appendix, showing that the proposed SPD algorithm consistently achieves superior performance across these metrics.
>
> **Weakness 1**
> > - In the dual update, $h$ is an iterative variable that serves as a stochastic estimate of the gradient of the function $K(x, y, \hat{x}, \hat{y}; \lambda)$  .
> > - While the symbol $h$ also appears in the primal update, each instance is clearly distinguished by different superscripts (e.g., $f$ or $g$) and subscripts.
>
> **Weakness 2**
> > Will remove unnecessary numbers.

---

### Decision · Program_Chairs · 2025-05-01

**Decision:**

Accept (poster)

**Comment:**

This paper proposes a smoothed primal-dual algorithm (SPD) for stochastic bilevel optimization with potentially nonconvex lower-level problems. The method is based on a Moreau envelope reformulation and avoids Hessian computations. Theoretical guarantees and empirical results demonstrate its effectiveness over existing approaches.

All reviewers recommend acceptance after the rebuttal, and I agree. For the final version, the authors are encouraged to improve notation clarity, better explain the experimental setup, and ensure theoretical claims are well supported. Clarifying parameter choices, comparing more thoroughly with related methods like SLM, highlighting the novelty over prior Moreau envelope approaches, and discussing the gap between the reformulation and the original problem would further strengthen the paper.